# LANGUAGE MODEL PREFERENCE EVALUATION WITH MULTIPLE WEAK EVALUATORS

## ABSTRACT

Despite the remarkable success of Large Language Models (LLMs), evaluating their outputs' quality regarding *preference* remains a critical challenge. Existing works usually leverage a powerful LLM (e.g., GPT4) as the judge for comparing LLMs' output pairwisely, yet such model-based evaluator is vulnerable to *conflicting preference*, i.e., output A is better than B, B than C, but C than A, causing contradictory evaluation results. To improve model-based preference evaluation, we introduce GED (Preference Graph Ensemble and Denoise), a novel approach that leverages multiple model-based evaluators to construct preference graphs, and then ensemble and denoise these graphs for better, non-contradictory evaluation results. In particular, our method consists of two primary stages: aggregating evaluations into a unified graph and applying a denoising process to eliminate cyclic inconsistencies, ensuring a directed acyclic graph (DAG) structure. We provide theoretical guarantees for our framework, demonstrating its efficacy in recovering the ground truth preference structure. Extensive experiments across ten benchmark datasets show that GED outperforms baseline methods in model ranking, response selection, and model alignment tasks. Notably, GED combines weaker evaluators like Llama3-8B, Mistral-7B, and Qwen2-7B to surpass the performance of stronger evaluators like Qwen2-72B, highlighting its ability to enhance evaluation reliability and improve model performance.

## 1 INTRODUCTION

Large Language Models (LLMs) have rapidly transformed various fields within artificial intelligence, particularly natural language processing (NLP) and decision-making systems (Wu et al., 2023; Li et al., 2023a). Despite the remarkable success of LLMs, the need for effective evaluation methods becomes paramount (Liu et al., 2023; Desmond et al., 2024; Siska et al., 2024). Preference evaluation, as one of the most important assessment methods, plays an indispensable role in evaluating and optimizing model performance (Rafailov et al., 2024; Yuan et al., 2024; Dubois et al., 2024b). Existing works usually leverage a powerful LLM (e.g., GPT4 (Achiam et al., 2023)) as the judge for comparing LLMs' output pairwisely (Li et al., 2023b; Chen et al., 2023b; Wang et al., 2022).

However, while such model-based pairwise preference evaluations offer a flexible approach, they can lead to contradictory evaluations in the assessment process (Naresh et al., 2024; Zhang et al., 2024b). For example, an LLM might evaluate three responses and conclude that Response A is better than Response B (A $\succ$ B), Response B is better than Response C (B $\succ$ C), yet paradoxically also rank Response C as better than Response A (C $\succ$ A). These cyclic patterns introduce inconsistencies that undermine the reliability of the evaluation results. We model this *conflicting preference* via the preference graph. Specifically, a preference graph is constructed with each response as a node and directed edges indicating pairwise preferences—an edge from node A to node B shows that the evaluator preferred response A over B. The noise illustrated by cycles (A $\succ$ B $\succ$ C $\succ$ A) manifests as loops in the preference graph. The whole process we show in the upper part of Figure 1. Ideally, a preference graph should be structured as a directed acyclic graph (DAG) to maintain consistency. As shown in the bottom half of Figure 1, even advanced LLMs like GPT-4-o exhibit significant noise in preference evaluation, highlighting their limitations as weak evaluators (Refer to Appendix A.3 for further details).

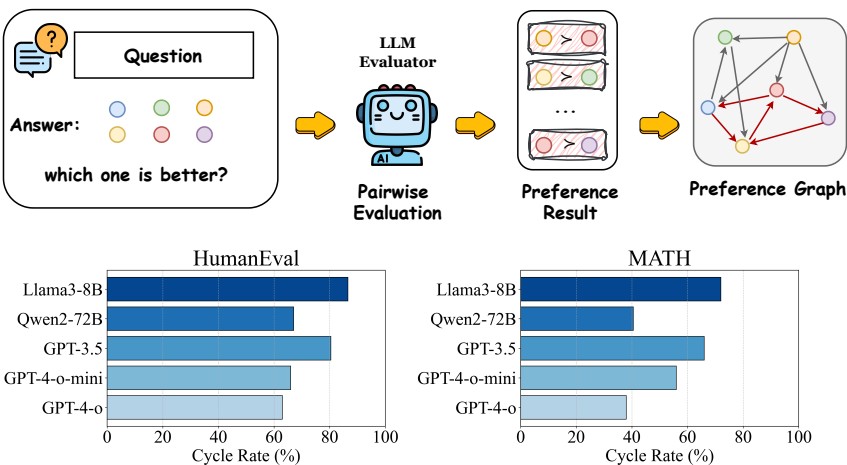

Figure 1: The figure is divided into two parts. The upper part shows the construction of a preference graph from pairwise LLM evaluations, with cycles indicating inconsistent preferences highlighted in red. The lower part presents the evaluation of ten responses for HumanEval (Chen et al., 2021) and MATH (Hendrycks et al., 2021) tasks, generated by Llama3-70B (AI@Meta, 2024). These were evaluated by GPT-4-o, GPT-4-o-mini, GPT-3.5 (Achiam et al., 2023), Qwen2-72B (Yang et al., 2024a), and Llama3-8B, revealing a significant cycle rate (details in Appendix A.5) in their evaluations. This highlights the limitations of even advanced LLMs as reliable evaluators.

To address this, we propose a novel framework, GED (Preference **G**raph **E**nsemble and **D**enoise), to address the inconsistencies in preference graphs generated through pairwise evaluations. Our method involves two key steps: (1) ensembling multiple weak evaluators to mitigate noise introduced by individual evaluators and (2) applying a denoising process to the resulting preference graph. By aggregating evaluations from multiple weak evaluators, we "average out" the noise and biases, resulting in a more robust approximation of the true preference structure. The denoising step further refines this aggregated graph by removing inconsistencies, ensuring the final preference graph is more reliable for downstream tasks. We provide a theoretical analysis demonstrating the soundness of GED, showing that by treating each individual preference graph as a random perturbation of a ground truth DAG, our ensemble and denoising framework can recover the ground truth DAG with high probability.

To validate the practical efficacy of GED, we conduct extensive experiments across model ranking, response selection, and model alignment tasks, utilizing ten widely recognized benchmark datasets, including HumanEval (Chen et al., 2021), AlpacaEval (Li et al., 2023b), MATH (Hendrycks et al., 2021), GSM8k (Chen et al., 2021), GAIA (Mialon et al., 2023), LIMA (Zhou et al., 2023), Vicuna (Chiang et al., 2023), Koala (Vu et al., 2023), WizardLM (Xu et al., 2023), and Self-Instruct (Wang et al., 2022). In these experiments, GED consistently outperformed baseline methods. For example, in the response selection task, applying GED yielded an average improvement of 4.51% compared to baseline methods across multiple benchmarks. Additionally, GED demonstrated substantial gains in scenarios where combining preference graphs from weaker evaluators surpassed the performance of even stronger individual evaluators. For instance, by using Llama3-8B, Mistral-7B, and Qwen2-7B as weaker evaluators, GED exceeded the performance of using the stronger Qwen2-72B model as an evaluator. These results highlight GED's ability to mitigate preference noise, improve consistency, and enhance model performance across diverse evaluation settings.

## 2 METHODOLOGIES

In this section, we begin by defining a preference graph, which serves as the foundation for representing pairwise preferences among candidates (Section 2.1). Building on this foundation, we introduce GED structured into three key stages (Section 2.2): (1) graph ensemble, where we aggregate individual preference graphs into a unified structure, (2) graph denoising, which removes cycles and

inconsistencies to ensure the preference graph is acyclic, and (3) graph-to-ranking, where we extract a reliable ranking of candidates from the denoised graph. Below, we provide detailed descriptions of each step.

## 2.1 PREFERENCE GRAPH

A preference graph is defined as a directed graph $G_P = (V, A, w)$, where $V = \{v_1, v_2, \ldots, v_n\}$ represents a set of $n$ alternatives or candidates, $A \subseteq V \times V$ is a set of directed arcs representing pairwise preferences between these alternatives, and $w : A \to \mathbb{R}^+$ is a weight function that assigns a positive real value to each arc, indicating the strength of the preference.

For any pair of distinct vertices $u, v \in V$, an arc $(u, v) \in A$ exists if there is a preference for $u$ over $v$, with the weight $w(u, v)$ reflecting the intensity of this preference. Formally, this can be represented as:

$$(u, v) \in A \quad \text{if and only if} \quad w(u, v) > 0 \tag{1}$$

The weight function $w(u, v)$ aggregates individual preferences or scores for the pair $(u, v)$. If multiple preference sources exist, the weight can be expressed as:

$$w(u, v) = \sum_{i=1}^{k} (s_i(u, v) - s_i(v, u)) \tag{2}$$

where $s_i(u, v)$ is the score or preference from the $i$-th source. The preference graph encapsulates the aggregate preferences among all pairs of alternatives, with the weight of each arc representing the cumulative preference strength derived from underlying data or models.

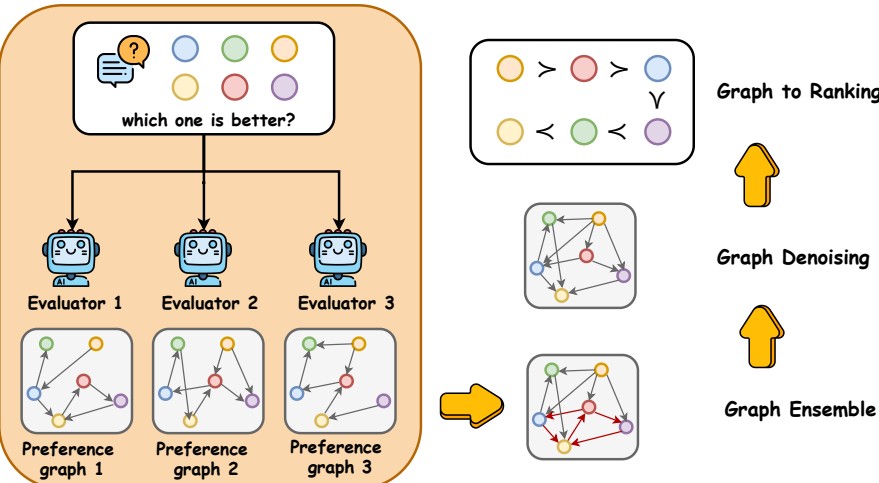

Figure 2: The GED framework mitigates noise and inconsistencies in preference evaluations by utilizing multiple weak evaluators and applying a graph denoising process. It consists of three stages: (1) Graph Ensemble, which combines individual preference graphs; (2) Graph Denoising, which removes cycles to ensure an acyclic structure; and (3) Graph-to-Ranking, which extracts a reliable ranking from the denoised graph.

## 2.2 GED: PREFERENCE GRAPH ENSEMBLE AND DENOISE

As illustrated in Figure 2, our method, GED (Preference Graph Ensemble and Denoise), begins by performing graph ensemble to aggregate a set of preference graphs. It then applies graph denoising to ensure acyclicity, followed by graph-to-ranking to derive the final node ranking. The detailed steps are as follows:

**Graph ensemble.** In the context of graph ensemble, given multiple weighted graphs $G_1 = (V, A_1, w_1), G_2 = (V, A_2, w_2), \ldots, G_k = (V, A_k, w_k)$ that share the same set of vertices $V$ but may differ in their arc sets $A_i$, the goal is to combine these graphs into a single ensemble graph $G_E = (V, A_E, w_E)$. The ensemble graph $G_E$ is formed by first defining the arc set $A_E$ as the union of the arc sets of the individual graphs and the weight function $w_E : A_E \to \mathbb{R}^+$ for the ensemble graph is then determined by summing the weights of the corresponding arcs in each graph.

**Graph denoising.** Graph denoising involves transforming the original graph $G = (V, A, w)$ into a DAG. This transformation is achieved by identifying and removing a set of arcs known as the Feedback Arc Set (FAS) (Gabow, 1995), which is a set of arcs whose removal makes the graph acyclic. The goal is to find a minimum FAS, denoted as $R^*(G)$, which is a set of arcs with the smallest total weights that needs to be removed to eliminate all cycles in $G$.

To find this minimum FAS, we can order the vertices of $G$ in a specific sequence $s = \{v_1, v_2, \ldots, v_n\}$. This vertex sequence induces a FAS $R(s)$, consisting of all arcs that point against the direction of the sequence, i.e., arcs $v_j \to v_i$ where $j > i$. The graph denoising problem is thus reframed as finding an optimal vertex sequence $s^*$ that induces the minimal FAS, such that $R(s^*) = R^*(G)$. This optimal sequence $s^*$ ensures that the total weights of arcs eliminated to achieve a DAG is minimized.

Finding a minimum FAS in general is known to be an NP-complete problem, whose computational complexity can be exponential (Karp, 2010; Bodlaender et al., 2012). Therefore, in our experiment, we apply the well-established approximation algorithm proposed in Eades et al. (1993). Details can be found in Appendix L.

**Graph to ranking.** Given a DAG graph $G = (V, A, w)$, our goal is to derive a ranking based on the structure of $G$. For each vertex $v \in V$, we compute the *descendant count* $\operatorname{desc}(v)$, defined as the number of vertices that are reachable from $v$ through directed arcs:

$$\operatorname{desc}(v) = |\{u \in V : v \rightsquigarrow u\}|, \tag{3}$$

where $v \rightsquigarrow u$ denotes that there is a directed path from $v$ to $u$. Vertices are then ranked based on their descendant counts, with higher descendant counts indicating a higher position in the ranking. Formally, the ranking $\mathcal{R}$ is represented as a sequence of subsets:

$$\mathcal{R} = \{v_1, v_2, \ldots, v_n\}, \tag{4}$$

where each $v_i$ represents a set of vertices with the $i$-th highest descendant count. The final ranking is then:

$$v_1 \succ v_2 \succ \cdots \succ v_n. \tag{5}$$

This structured approach ensures that the ranking reflects not only the individual preferences captured in the graph but also the relative strength of these preferences as represented through their descendant connections.

## 2.3 APPLICATIONS

We apply GED to three tasks: Response Selection, selecting the best response from LLM-generated candidates; Model Ranking, ranking models based on task performance; and Model Alignment, identifying the best instruction-response pairs for training. The steps are as follows:

**Response selection.** In the response selection task, a model $\mathcal{M}$ generates $n$ candidate answers $\{ans_1, \ldots, ans_n\}$ for each question $q \in Q$, with the objective of identifying the optimal answer $ans_q^*$ for each query. To achieve this, we employ multiple evaluators $\mathcal{A} = \{a_1, \ldots, a_k\}$, who provide pairwise preferences among the candidate answers. For each question $q$, we construct a set of preference graphs $\{G_a : a \in \mathcal{A}\}$, where each graph $G_a = (V_q, A_a, w_a)$ encapsulates the preferences of evaluator $a$. The vertex set $V_q = \{v_1, \ldots, v_n\}$ corresponds to the candidate answers, while the directed arcs $A_a$ indicate the pairwise preferences among these responses. Each arc is weighted by $w_a$, reflecting the strength of the preference indicated by the evaluator. The construction of preference graphs involves evaluating each pair of candidate answers $ans_i$ and $ans_j$. Evaluators assess the quality of these answers, assigning a preference that is denoted by a directed arc $(v_i, v_j)$ in $A_a$, with a corresponding weight $w_a(v_i, v_j)$ based on the strength of preference. This process is detailed in Appendix D. After collecting the preference graphs $\{G_a : a \in \mathcal{A}\}$ for a question $q$, we apply GED to

aggregate these graphs. The aggregation begins with merging all preference graphs into a unified graph $G_q = (V_q, A_q, w_q)$, which is subsequently processed to remove cycles, resulting in a DAG. From this DAG, we derive the ranking $\mathcal{R}_q = \{v_1 \succ v_2 \succ \ldots \succ v_n\}$ of the candidate answers, where the highest-ranked answer is selected as $ans_q^*$. This process is repeated for each question $q \in Q$, yielding a final set of selected answers $ans^* = \{ans_1^*, \ldots, ans_t^*\}$. This set reflects a consensus from multiple evaluators in $\mathcal{A}$, ensuring the chosen answers represent the highest quality responses based on rigorous preference evaluation.

**Model Ranking.** In model ranking task, the goal is to rank a set of models $M = \{\mathcal{M}_1, \ldots, \mathcal{M}_n\}$ based on their responses to a series of questions $Q = \{q_1, \ldots, q_t\}$. A group of evaluators, denoted as $\mathcal{A} = \{a_1, a_2, \ldots, a_k\}$, assesses the model outputs by providing preferences for pairs of responses for each question. For each question $q \in Q$, the evaluators generate preference graphs $\{G_a : a \in \mathcal{A}\}$, where each graph $G_a = (V_q, A_a, w_a)$ encapsulates the preferences of evaluator $a$ over the models. The vertex set $V_q = \{v_1, \ldots, v_n\}$ corresponds to the models, while the directed arcs $A_a$ indicate pairwise preferences, with weights $w_a : A_a \to \mathbb{R}^+$ reflecting the strength of these preferences. The preference graph for a given question $q$ is constructed by evaluating each pair of models $\mathcal{M}_i$ and $\mathcal{M}_j$, represented by nodes $v_i$ and $v_j$. For their respective answers $ans_i = \mathcal{M}_i(q)$ and $ans_j = \mathcal{M}_j(q)$, evaluators provide a preference indicating which answer is favored. This preference is represented by a directed arc $(v_i, v_j)$ in $A_a$, assigned a weight $w_a(v_i, v_j)$ based on preference strength. The detailed procedure is outlined in Appendix D. Once the preference graphs $\{G_a : a \in \mathcal{A}\}$ are collected for a question $q$, we employ GED to aggregate these graphs. The method begins by merging all preference graphs into a single graph $G_q = (V_q, A_q, w_q)$, which is then transformed into a DAG by removing cycles. The final ranking $\mathcal{R}_q = \{v_1 \succ v_2 \succ \ldots \succ v_n\}$ is derived from this DAG. This process is repeated for each question $q \in Q$, yielding a set of rankings $\{\mathcal{R}_q : q \in Q\}$. To compute the overall ranking $\mathcal{R}^*$ of the models across all questions, we conduct a ranking ensemble on the set $\{\mathcal{R}_q : q \in Q\}$, as detailed in Appendix M. This approach culminates in a final ranking that reflects the models' performance as assessed by multiple evaluators in $\mathcal{A}$.

**Model Alignment.** In the model alignment task, we have multiple data pairs of the form $(x, y_1), (x, y_2), \ldots, (x, y_n)$ for each instruction $x$. The objective is to identify the best response $y^*$ corresponding to each instruction $x$. We utilize multiple evaluators $\mathcal{A} = \{a_1, \ldots, a_k\}$ to provide pairwise preferences among the candidate responses. For each instruction $x$, we construct a set of preference graphs $\{G_a : a \in \mathcal{A}\}$, where each graph $G_a = (V_x, A_a, w_a)$ represents the preferences of evaluator $a$. The vertices $V_x$ correspond to the candidate responses $\{y_1, \ldots, y_n\}$, and the directed arcs in $A_a$ indicate preferences, weighted by $w_a$. After constructing the preference graphs, we apply GED to aggregate them into a single graph $G_x$. This graph undergoes a denoising process to remove cycles, allowing us to derive a ranking $\mathcal{R}_x$ of the responses. The highest-ranked response in $\mathcal{R}_x$ is selected as $y^*$ for that instruction $x$. This process is repeated for all $t$ instructions, resulting in the final training set $\{(x_1, y_1^*), \ldots, (x_t, y_t^*)\}$, which reflects a consensus across the evaluators in $\mathcal{A}$.

## 3 THEORETICAL ANALYSIS

In this section, we provide a theoretical foundation for our method, showing that by modeling preference graphs as random perturbations of a ground truth DAG, GED can reliably recover the true structure through graph ensemble and denoising with high probability, demonstrating its robustness in handling noisy evaluations.

Theoretically, we treat each of our preference graph as a random perturbation of some ground truth DAG $G = (V, A)$. Specifically, we consider a random graph generator $\mathcal{G}(G, \delta_1, \delta_2)$ with parameters $\delta_1, \delta_2 \in [0, 1]$ such that $G_i = (V_i, A_i) \sim \mathcal{G}(G, \delta_1, \delta_2)$ satisfies $V_i = V$. Furthermore, for each $u, v \in V$ with $u \neq v$,

1) If $(u \to v) \in A$, then
$$\mathbb{P}((u \to v) \in A_i) = 1 - \delta_1 \quad \text{and} \quad \mathbb{P}((v \to u) \in A_i) = \delta_1;$$

2) If $(u \to v), (v \to u) \notin A$, then
$$\mathbb{P}((u \to v), (v \to u) \notin A_i) = 1 - \delta_2, \quad \mathbb{P}((u \to v) \in A_i) = \frac{\delta_2}{2} \quad \text{and} \quad \mathbb{P}((v \to u) \in A_i) = \frac{\delta_2}{2}.$$

That is, each edge in $E$ has probability $\delta_1$ of being flipped and each pair of unconnected nodes has probability $\delta_2$ of being connected with a random direction.

Now, given that $G_1, \ldots, G_N \overset{i.i.d.}{\sim} \mathcal{G}(G, \delta_1, \delta_2)$, we will show that to some extent our combination of graph ensemble and graph denoising can indeed provably recover the ground truth DAG $G$. For simplicity, all edges in $G_1, \ldots, G_N$ and $G$ are considered equal weighted. Meanwhile, we use MAS($\cdot$) to denote the graph obtained by denoising, which stands for the maximum acyclic subgraph (MAS). Then, we have the following theorem.

**Theorem 1** *Suppose $G_1, \ldots, G_N \overset{i.i.d.}{\sim} \mathcal{G}(G, \delta_1, \delta_2)$ for some ground truth $G = (V, A)$. Let $\widehat{G}$ be the graph ensembled from $G_1, \ldots, G_N$ by operations defined in Section 2.2. Then, as long as $\delta_1 = 0.5 - \epsilon$. For some $\epsilon > 0$, we have*

$$\mathbb{P}\left(G \subseteq MAS(\widehat{G})\right) \geq 1 - 2|A|\exp\left(-\frac{N\epsilon^2}{2}\right) - 2U\exp\left(-\frac{N\epsilon^2}{6U^2\delta_2 + 2U\epsilon}\right),$$

*where $G \subseteq MAS(\widehat{G})$ represents that $G$ is a subgraph of $MAS(\widehat{G})$ and $U = \frac{|V|(|V|-1)}{2} - |A|$ is the number of pairs of unconnected nodes in $G$.*

The full proof is given in Appendix E. From the theorem, we can see that the probability of failure decreases exponentially as the number of samples $N$ increases. Meanwhile, this guarantee only requires $\delta_1 < 0.5$ and does not place restrictions on $\delta_2$, which are very mild conditions.

## 4 EXPERIMENTS ON RESPONSE SELECTION

Table 1: Performance comparison of response selection methods across five benchmarks. GED consistently outperforms baseline methods, demonstrating the effectiveness of graph denoising and the aggregation of weaker evaluators.

| Method | | HumanEval | AlpacaEval | MATH | GSM8k | GAIA | Avg |
|---|---|---|---|---|---|---|---|
| **Single model** | Llama3-8B | 43.90 | 27.29 | 22.08 | 56.67 | 6.78 | 31.34 |
| | Mistral-7B | 23.17 | 11.80 | 23.25 | 39.83 | 7.03 | 21.01 |
| | Qwen2-7B | 48.58 | 25.71 | 59.92 | 76.75 | 7.70 | 43.73 |
| | Qwen2-72B | 57.93 | 29.58 | 72.75 | 84.67 | 11.52 | 51.29 |
| | ContraSolver | 65.42 | 31.12 | 74.95 | 86.84 | 12.22 | 54.11 |
| | ListPreference | 61.52 | 31.67 | 71.75 | 85.0 | 10.90 | 52.16 |
| | Self-consistency | 60.98 | 29.33 | 73.58 | 84.91 | 8.86 | 51.53 |
| **Single evaluator** | Llama3-8B | 62.19 | 29.31 | 74.27 | 83.16 | 11.31 | 52.04 |
| | with graph denoising | 64.02 | 30.18 | 74.73 | 86.00 | 11.72 | 53.33 |
| | Mistral-7B | 67.24 | 27.70 | 74.41 | 83.83 | 10.50 | 52.73 |
| | with graph denoising | 68.73 | 29.93 | 74.77 | 83.91 | 10.74 | 53.61 |
| | Qwen2-7B | 61.58 | 28.69 | 74.50 | 85.41 | 11.11 | 52.25 |
| | with graph denoising | 65.85 | 29.44 | 74.79 | 86.38 | 11.25 | 53.54 |
| | Qwen2-72B | 60.97 | 31.04 | 74.73 | 86.47 | 12.14 | 53.07 |
| | with graph denoising | 68.90 | 31.17 | 75.33 | 87.45 | 12.26 | 55.02 |
| **Multiple evaluator** | Multi-MV | 66.18 | 29.57 | 74.77 | 86.42 | 11.72 | 53.73 |
| | GED (w/o denoising) | 69.25 | 30.98 | 74.29 | 87.17 | 12.68 | 54.87 |
| | GED | **70.73** | **32.44** | **75.58** | **88.18** | **13.33** | **56.05** |

**Experiment Setup.** In this section, we evaluate the performance of GED on five benchmarks: HumanEval (Chen et al., 2021), AlpacaEval (Li et al., 2023b), MATH (Hendrycks et al., 2021), GSM8k (Chen et al., 2021), and GAIA (Mialon et al., 2023). The Qwen2-72B (Yang et al., 2024a) model ($\mathcal{M}$) generates ten candidate responses per question, and we assess the effectiveness of different methods in selecting the best response. For further implementation details, see Appendix A.

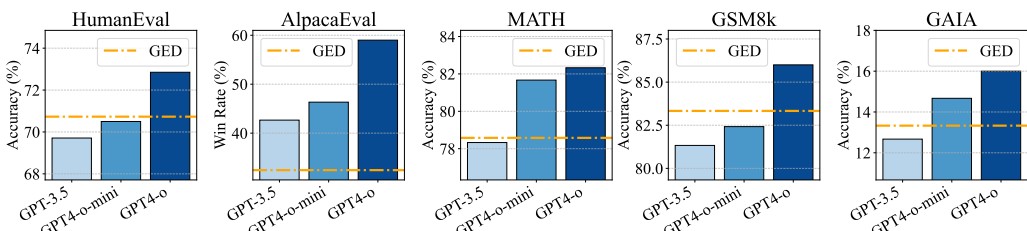

Figure 3: Comparison of GED with GPT-3.5, GPT-4-o-mini, and GPT-4-o on 100 randomly selected tasks. GED consistently outperforms GPT-3.5 across all tasks and surpasses GPT-4-o-mini on challenging tasks like HumanEval and GSM8k, showcasing the effectiveness of weak evaluator aggregation with graph denoising.

We evaluate performance using three setups. First, in the *single model* setting, the baselines include ContraSolver(Zhang et al., 2024b), Self-consistency(Wang et al., 2022), and direct evaluation with models (Llama3-8B, Mistral-7B, Qwen2-7B and Qwen2-72B). Additionally, we include a baseline called ListPreference, where instead of pairwise comparisons, all candidate responses are input into Qwen2-72B for selecting the most appropriate response. Then, in the *single evaluator* setting, individual evaluators (Llama3-8B, Mistral-7B, Qwen2-7B, Qwen2-72B) select the best response from $\mathcal{M}$'s outputs, with and without applying GED's graph denoising. Finally, in the *multiple weak evaluators* setup, we combine three weaker evaluators (Llama3-8B, Qwen2-7B, Mistral-7B) to select responses from Qwen2-72B with GED. We also include a new baseline referred to as Multi-MV, which just takes the majority for each pairwise comparison as the preferred answer. We present the results of GED and its variant (w/o denoising), which ensembles the preference graphs without the denoising step.

**Main results.** Table 1 presents the results of the response selection task across five benchmarks. GED consistently outperforms baseline methods, including both single model evaluations (*single model*) and direct response selection by individual models (*single evaluator*). This demonstrates the strength of aggregating weak evaluators with GED, particularly when coupled with graph denoising, which enhances response quality by filtering out noise and biases. This highlights the effectiveness of aggregating weak evaluators and applying graph denoising to improve response quality. Furthermore, by combining preference graphs from weaker models (Llama3-8B, Mistral-7B, Qwen2-7B), GED surpasses the performance of a much stronger evaluator (Qwen2-72B). This underscores the value of ensemble methods in mitigating the limitations of individual evaluators. Then, the denoising process proves to be crucial for improving consistency and overall response quality. The substantial performance gains observed when using GED with denoising, compared to both the single evaluator setup and the ensemble without denoising, highlight its importance in refining response selection. We also evaluated Multi-MV, which aggregates pairwise comparisons using majority voting across evaluators. While Multi-MV offers improvements over individual evaluators, it underperforms compared to GED, highlighting GED's ability to capture more nuanced evaluation signals and reduce inconsistencies. Additionally, we observed that the ListPreference baseline performed worse than Qwen2-72B as single evaluator, likely due to LLM limitations in handling long-text. Lastly, to further evaluate GED, we compared its performance with GPT-3.5, GPT-4-o-mini, and GPT-4-o. Due to computational and API cost constraints, we limited the evaluation to 100 data points for each task. As shown in Figure 3, GED consistently outperformed GPT-3.5 across all tasks and surpassed GPT-4-o-mini on challenging benchmarks like HumanEval and GSM8k. These results highlight the superiority of GED, particularly in leveraging multi-weak evaluators and graph denoising to outperform individual state-of-the-art models.

**Ablation study.** We evaluate the impact of removing the ensembling step in GED, referred to as the *(w/o ensemble)* variant. In this case, individual evaluators' preference graphs are denoised and converted to rankings, which are then aggregated using methods such as *Weight Score*, *Kemeny*, *Weighted Kemeny*, *Pairwise Majority*, and *Weighted Pairwise Majority* (detailed in Appendix M). For simplicity of presentation, we use *Weight Score* to represent *GED (w/o ensemble) (Weight Score)*. As shown in Figure 4, all *(w/o ensemble)* methods consistently underperform compared to GED. This performance gap arises because converting graphs to ranks before aggregation leads to information

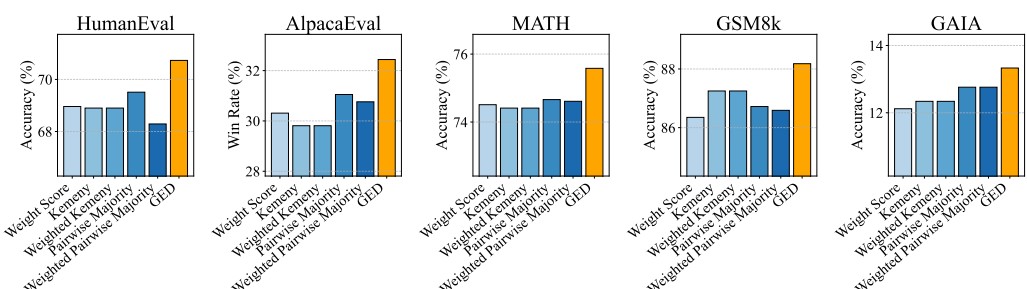

Figure 4: Comparison of GED and *(w/o ensemble)* variants. GED outperforms due to preserving more information by directly ensembling preference graphs, while rank aggregation in the *(w/o ensemble)* methods leads to performance loss.

Table 2: Results of the model ranking task, evaluated using Ranking Correction. Higher correlation values indicate a stronger alignment with the ground truth rankings.

| Model | | Weight Score | Kemeny | Weighted Kemeny | Pairwise Majority | Weighted Pairwise Majority | Avg. |
|---|---|---|---|---|---|---|---|
| **Single evaluator** | Llama3-70B | 50.88 | 60.80 | 60.80 | 62.23 | 61.85 | 59.31 |
| | with graph denoising | 52.44 | 62.54 | 62.54 | 63.92 | 62.18 | 60.72 |
| | Qwen2-72B | 65.34 | 59.87 | 67.39 | 66.05 | 66.59 | 65.04 |
| | with graph denoising | 66.05 | 70.43 | 70.43 | 72.32 | 72.41 | 70.32 |
| | Qwen1.5-72B | 63.64 | 60.72 | 60.72 | 62.65 | 63.28 | 62.20 |
| | with graph denoising | 64.81 | 61.77 | 61.77 | 64.36 | 64.76 | 63.49 |
| | Mistral-8×7B | 64.90 | 68.74 | 68.74 | 73.06 | 72.87 | 69.66 |
| | with graph denoising | 65.47 | 70.06 | 69.92 | 73.39 | 73.21 | 70.41 |
| **Multiple evaluator** | GED (w/o ensemble) | 62.82 | 68.44 | 68.44 | 69.34 | 67.34 | 67.27 |
| | GED (w/o denoising) | 64.84 | 69.23 | 69.81 | 75.35 | 74.37 | 70.72 |
| | GED | **66.59** | **71.14** | **71.14** | **77.17** | **76.46** | **72.50** |

loss. In contrast, GED ensembles the graphs directly, preserving more detailed preference information and resulting in better final rankings.

## 5 EXPERIMENTS ON MODEL RANKING

**Experiment Setup.** In this section, we evaluate the effectiveness of GED in the model ranking task within a human preference setting, using the AlpacaEval benchmark (Li et al., 2023b). We employ 30 widely used models from the AlpacaEval dataset as our model set $\mathcal{M}$, while the benchmark's questions form the question set $Q$. The rankings provided by the AlpacaEval benchmark serve as ground truth for evaluating the accuracy of various ranking methods. This is justified by AlpacaEval's strong correlation with Chatbot Arena rankings, making it a reasonable proxy for human judgments (Dubois et al., 2024a). We adopt Ranking Correction, measured by the Spearman rank correlation coefficient, to evaluate the similarity. To generate rankings, we utilize outputs from the open-source models Llama3-70B, Qwen2-72B, Mistral-8×7B, and Qwen1.5-72B as our evaluators, denoted as set $\mathcal{A}$. For further implementation details, see Appendix A. We investigate two variants of GED: (w/o ensemble) denoises the preference graphs from different evaluators for the same question, converts each into a ranking, and then ensembles these rankings to produce the final output, while (w/o denoising) directly ensembles the preference graphs to obtain the final ranking without denoising.

**Main results.** The results, presented in Table 2, show that GED outperforms all single-model baselines, highlighting the significant improvement in ranking accuracy achieved by leveraging preference information from multiple evaluators. Moreover, GED surpasses the (w/o ensemble) variant, indicating that generating rankings through graph ensemble first prevents information loss compared to converting individual graphs into rankings. When the ensemble graph is not denoised (w/o denoising), residual noise can adversely affect the final ranking quality. Additionally, our denoising method also enhances results in single-model settings.

# 6 EXPERIMENTS ON INSTRUCT TUNING

**Experiment Setup.** In this section, we explore the effects of various data selection methods for model alignment on Llama-2-7B (Touvron et al., 2023) and Mistral-7B (Jiang et al., 2023) through instruct tuning. Specifically, we randomly sampled 5000 data points from UltraFeedback (Cui et al., 2023), and utilized Qwen1.5-14B (Yang et al., 2024a) to generate 8 responses for each data point as instruct data. We then applied four different methods—Random, Longest (Zhao et al., 2024), ContraSolver (Zhang et al., 2024b), and our proposed GED—to select a subset of these responses for model alignment training. The Origin refers to the performance of the base model without alignment. The resulting models were evaluated on the HH-RLHF (Bai et al., 2022) benchmark, which consists of four sub-sets: Harmless (base), Helpful (base), Helpful (online), and Helpful (rejection). For evaluation, we adopted the same Reward model, following prior work (Song et al., 2024; Yu et al., 2023), to measure human preference levels gained by the models. These results are summarized in Table 3. To ensure a comprehensive evaluation, we also tested the models, using the Llama-2-7B backbone, across additional benchmarks such as LIMA (Zhou et al., 2023), Vicuna (Chiang et al., 2023), Koala (Vu et al., 2023), WizardLM (Xu et al., 2023) and Self-Instruct (Wang et al., 2022), in line with recent works (Chen et al., 2023b; Zhang et al., 2024a; Hu et al., 2024).

Table 3: Performance comparison of different methods (Random, Longest, ContraSolver, and GED) on model alignment task across the HH-RLHF benchmark. The results demonstrate the superiority of GED in consistently selecting high-quality responses, leading to improved model performance compared to baseline methods.

| BaseModel | | Harmless (base) | Helpful (base) | Helpful (online) | Helpful (rejection) | Avg. |
|---|---|---|---|---|---|---|
| **Llama-2-7B** | Origin | 69.67 | 61.12 | 65.41 | 64.06 | 65.07 |
| | Random | 69.38 | 62.87 | 66.75 | 65.57 | 66.14 |
| | Longest | 69.65 | 63.54 | 66.99 | 66.43 | 66.65 |
| | ContraSolver | 69.57 | 63.61 | 66.87 | 66.59 | 66.66 |
| | GED | **69.71** | **64.10** | **67.87** | **67.01** | **67.17** |
| **Mistral-7B** | Origin | 61.59 | 59.51 | 65.21 | 63.17 | 62.37 |
| | Random | 59.15 | 59.61 | 64.06 | 62.38 | 61.30 |
| | Longest | 61.81 | 60.53 | 64.52 | 63.22 | 62.52 |
| | ContraSolver | 61.48 | 59.85 | 64.66 | 63.41 | 62.85 |
| | GED | **61.96** | **60.71** | **65.49** | **63.82** | **63.50** |

**Main results.** From Table 3, we observe that GED consistently outperforms all baseline methods, demonstrating its effectiveness in selecting high-quality responses when multiple answers are available for a given instruction. When faced with multiple responses $y_1, y_2, \ldots, y_n$ for a given instruction $x$, the Random selection method can have a detrimental impact, especially when the quality of the responses is inconsistent. This effect is most evident with the Mistral-7B, where Random selection actually performs worse than the Origin, indicating that randomly chosen data points can introduce noise and degrade the model's performance. Moreover, we find that simply selecting the longest response does not always lead to the best outcomes. While longer responses may provide more detailed answers, they are not necessarily better in terms of quality, particularly when both high-quality and low-quality answers exist for the same question. This is reflected in the results where the Longest method underperforms compared to both ContraSolver and GED, emphasizing that response length alone is not always a reliable criterion. From Figure 5, we can draw similar conclusions as Table 3. Specifically, we observe that GED consistently outperforms all baselines, demonstrating its effectiveness across all datasets. Particularly in AlpacaEval and Self-Instruct, the Random baseline performs worse than the Origin model, indicating that when response quality varies significantly, poor selection can lead to negative performance. In contrast, GED excels by aggregating preference graphs and applying denoising, effectively filtering out low-quality responses. This ensures robust performance, especially in cases where response quality is inconsistent. The denoising step is crucial for removing noisy evaluations, leading to improved model alignment. The denoising process in GED proves essential, particularly in settings with inconsistent responses, as it removes evaluation noise and leads to more robust performance.

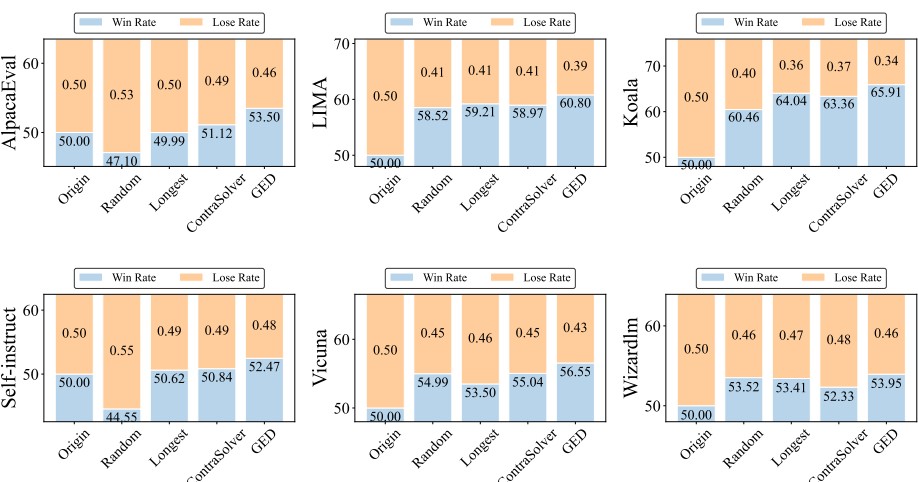

Figure 5: Performance comparison of different methods (Random, Longest, ContraSolver, and GED) across multiple benchmarks, including LIMA, Vicuna, Koala, WizardLM and Self-Instruct. The results show GED effectively filters low-quality responses, improving performance and model alignment over baselines.

## 7 RELATED WORK

**Preference evaluation of LLMs.** Reference-free evaluation metrics have a long history (Louis & Nenkova, 2013), which evaluates the generated text based on intrinsic properties and coherence with the context. Although they achieve high accuracy on matching inner-evaluator, the achievement suffers from spurious correlations such as perplexity and length (Durmus et al., 2022). Recently, people have started using a strong model (e.g., GPT-4) as an evaluator to perform a zero-shot reference-free evaluation on the weak models (Shen et al., 2023; Dubois et al., 2024b; Chen et al., 2023b). However, using LLM-based preference evaluations can introduce inconsistencies in preference graphs, often resulting in cyclic preferences or contradictions when comparing multiple outputs.

**Weak supervision.** The concept of weak-to-strong supervision originates from the need to leverage noisy or partial labels in machine learning tasks, enabling the development of more robust models from imperfect data (Ratner et al., 2016; Zhang et al., 2023b; 2022). In LLMs, weak-to-strong supervision aids AI alignment by allowing weaker models to improve strong ones, enhancing performance without extensive data and supporting scalable oversight (Zheng et al., 2024a; Guo & Yang, 2024; Tong et al., 2024). Similarly, in task-oriented LLMs, weak-to-strong learning improves LLM's ability by enabling strong models to refine their data autonomously, boosting performance without extensive high-quality input (Zhang et al., 2023a; Yang et al., 2024b). Through weak-to-strong supervision, LLM performance can be significantly improved by iteratively transforming low-quality labels into more reliable ones, leading to more effective model training and robust outputs (Zakershahrak & Ghodratnama, 2024; Lang et al., 2024).

## 8 CONCLUSION

In this paper, we presented GED, a framework designed to address inconsistencies in pairwise preference evaluations by LLMs. By employing graph ensemble techniques and denoising, GED reduces cyclic patterns and enhances the reliability of evaluation outcomes. Our theoretical analysis shows that GED can recover the ground truth DAG under reasonable conditions, improving consistency in preference rankings. Extensive experiments across response selection, model ranking, and instruct tuning demonstrate the efficacy of our method. GED consistently outperformed baseline methods in both single-evaluator and multi-evaluator settings, particularly in scenarios where combining weak evaluators led to superior results over stronger individual evaluators. Future work will explore extending GED to broader evaluation frameworks and applying its principles to more complex decision-making tasks, including multi-agent systems and human-AI interaction.

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

# A IMPLEMENTATION DETAILS

## A.1 EXPERIMENTAL SETUP

All experimental procedures were conducted on a machine equipped with an AMD EPYC 7543 32-Core Processor, 512GB memory, 128 CPUs, and four 80GB NVIDIA A800 GPUs. The code is available at `https://anonymous.4open.science/r/kkk-B9F2`. The references to Llama-2-7B, Llama3-70B, Llama3-8B, Mistral-7B, Mistral-8×7B, Qwen2-7B, and Qwen2-72B in the main text refer to the specific models: Llama-2-7b-chat-hf, Meta-Llama-3-70B-Instruct, Meta-Llama-3-8B-Instruct, Mixtral-7B-Instruct-v0.3, Mixtral-8x7B-Instruct-v0.1, Qwen2-7B-Instruct, and Qwen2-72B-Instruct. We utilized the reward model oasst-rm-2-pythia-6.9b-epoch-1 following prior works (Song et al., 2024; Yu et al., 2023). Each experiment was repeated three times, and the average performance was reported as the final result. Our training script was adapted from the example provided in LlamaFactory (Zheng et al., 2024b), specifically `https://github.com/hiyouga/LLaMA-Factory/blob/main/examples/train_full/llama3_full_sft_ds3.yaml`. The training was configured with a batch size of 1 per device, gradient accumulation steps of 4, a learning rate of 1e-5, and he model was trained for 3 epochs, with warmup over 20 steps and a cosine learning rate scheduler. For generating diverse responses from LLMs, we followed the configuration in Yuan et al. (2024), setting $T = 0.7$ and $p = 0.9$. For tasks such as AlpacaEval (Dubois et al., 2024b), we used GPT-4-o unless stated otherwise.

## A.2 DETAILS OF EVALUATOR SELECTION ACROSS DIFFERENT TASKS

In this subsection, we provide more detailed information about the selection of evaluators across different tasks.

**Response Selection.** In the response selection task, we evaluated both single models and ensembles of evaluators using the GED method. For single model evaluation, we assessed the standalone performance of Llama3-8B, Mistral-7B, Qwen2-7B, and Qwen2-72B on benchmarks such as HumanEval, AlpacaEval, MATH, GSM8k, and GAIA. This served two purposes: first, to establish the baseline performance of smaller models (Llama3-8B, Mistral-7B, Qwen2-7B) that are later combined using GED, and second, to compare against Qwen2-72B, where we tested a baseline approach by randomly selecting one response from the 10 it generated for each question. For single evaluator evaluation, each model (Llama3-8B, Mistral-7B, Qwen2-7B, and Qwen2-72B) was also used as an evaluator to rank responses generated by Qwen2-72B. Notably, Qwen2-72B acted as a self-evaluator, selecting the best response from its own generated outputs. Finally, for multiple evaluator evaluation, our GED method combined the evaluations of Llama3-8B, Mistral-7B, and Qwen2-7B to rank responses.

**Model Ranking.** For the model ranking task, we selected larger models as evaluators to ensure alignment with rankings produced by GPT-4. Specifically, we used Llama3-70B, Qwen2-72B, Qwen1.5-72B, and Mistral-8×7B as single evaluators. This choice was guided by two factors: performance considerations and practical feasibility. Larger models generally produce more reliable rankings, closely aligning with GPT-4, and the AlpacaEval benchmark, containing 805 tasks, makes the computational cost of using larger models acceptable. In the multiple evaluator setting, our GED method aggregated the evaluations from these four larger models to produce robust and consistent rankings. The combination of these high-capacity models ensures that our approach yields rankings that are both accurate and consistent across tasks.

**Instruction Tuning.** In the instruction tuning task, the objective was to perform data selection for model training. For this task, we employed Llama3-8B, Mistral-7B, and Qwen2-7B as evaluators. These models were selected because they balance computational efficiency and evaluation performance, making them suitable for iterative instruction tuning processes. The evaluators' pairwise preferences were aggregated using GED to identify the most appropriate responses for training. This approach ensured that the selected data pairs reflected a consensus among the evaluators while keeping computational costs manageable.

## A.3 DEFINITION OF WEAK EVALUATORS

In this work, "weak evaluators" are defined by their tendency to produce noisy or inconsistent pairwise preferences, not by their overall model capacity. Even advanced models like GPT-4-o can generate preference graphs with cycles, indicating that preference inconsistency is a universal challenge, regardless of model scale. Evaluators such as Llama3-8B, Mistral-7B, Qwen2-7B, and even GPT-4-o are considered weak if their pairwise evaluations exhibit significant noise or conflicts.

## A.4 EVALUATION SETTINGS: SINGLE MODEL VS. SINGLE EVALUATOR

The "single model" setting evaluates each model's (Llama3-8B, Mistral-7B, Qwen2-7B, Qwen2-72B) outputs directly on benchmarks like HumanEval, AlpacaEval, MATH, GSM8k, and GAIA, without selection or modification. In contrast, the "single evaluator" setting uses these models to select the best response from ten candidates generated by Qwen2-72B, assessing their evaluation capability. The key difference is that the single model setting focuses on generation quality, while the single evaluator setting assesses evaluation ability.

## A.5 DEFINITION OF CYCLE RATE

The Cycle Rate is the percentage of preference graphs with at least one cycle, indicating inconsistency in pairwise comparisons. For example, if an evaluator produces cycles in 100 out of 164 graphs for the HumanEval dataset, the Cycle Rate is $\frac{100}{164} \times 100 = 60.97\%$. A lower Cycle Rate indicates greater consistency, while a higher rate suggests evaluator biases or difficulties with ambiguous comparisons. This metric helps assess the reliability of evaluators, such as GPT-4-o and GPT-4-o-mini, across different datasets.

## A.6 DATASET

In this appendix, we provide detailed information about the datasets used in main text.

- **UltraFeedback (Cui et al., 2023):** UltraFeedback is a large-scale, fine-grained, diverse preference dataset, used for training powerful reward models and critic models. We collect about 64k prompts from diverse resources (including UltraChat, ShareGPT, Evol-Instruct, TruthfulQA, FalseQA, and FLAN). We then use these prompts to query multiple LLMs (see Table for model lists) and generate 4 different responses for each prompt, resulting in a total of 256k samples.

- **HH-RLHF (Bai et al., 2022):** The HH-RLHF dataset contains human preference data for training language models to be helpful and harmless, as well as red teaming data to identify harmful model outputs. The preference data includes pairs of chosen and rejected responses, while the red teaming data includes transcripts of adversarial interactions with AI assistants, rated for harmfulness. We strictly follow prior works (Song et al., 2024; Yu et al., 2023) and used the code from this repository [1] for testing.

- **MATH (Hendrycks et al., 2021):** The MATH dataset consists of 12,500 challenging competition-level math problems, each with a detailed step-by-step solution. It is designed to teach models to generate answer derivations and explanations, aiding in mathematical reasoning. Despite progress in improving accuracy, the dataset highlights the limitations of large Transformer models in solving complex math problems without new algorithmic advancements. Due to the high resource cost of using the full test set, we randomly sampled 400 problems from the test set for evaluation.

- **GSM8k (Chen et al., 2021):** GSM8K (Grade School Math 8K) is a collection of 8.5K high-quality math word problems designed for grade school students. It supports the task of multi-step reasoning and question answering in basic math. The problems require 2 to 8 steps, focusing on elementary arithmetic operations (addition, subtraction, multiplication, and division). The solutions are provided in natural language, making it accessible for evaluation of language models' internal reasoning. GSM8K has been widely used to test logic and mathematical capabilities in language models, especially for benchmarks like

---

[1]https://github.com/AlibabaResearch/DAMO-ConvAI/tree/main/PRO

the LLM Leaderboard. Due to the high computational cost of using the entire test set, we randomly sampled 400 data points from the test set for our evaluation.

- GAIA (Mialon et al., 2023): The GAIA dataset is a benchmark designed to evaluate next-generation LLMs with augmented capabilities like tooling and search access. It consists of over 450 complex questions with unambiguous answers, requiring various levels of autonomy and tooling. The dataset is divided into three levels, each increasing in difficulty, with a public dev set for validation and a private test set for evaluation. We used the entire test set for our evaluation.

- HumanEval (Chen et al., 2021): The OpenAI HumanEval dataset contains 164 programming problems, each with a function signature, docstring, body, and unit tests. These problems are handwritten to ensure they were not included in the training sets of code generation models. The dataset is designed to evaluate the performance of models in Python code generation. We used the entire test set for evaluation.

- AlpacaEval. (Dubois et al., 2024b): AlpacaEval consists of 805 instructions, including 252 from the self-instruct test set (Wang et al., 2022), 188 from the Open Assistant (OASST) test set, 129 from Anthropic's helpful test set (Zhou et al., 2023), 80 from the Vicuna test set (Chiang et al., 2023), and 156 from the Koala test set (Vu et al., 2023).

- LIMA. (Zhou et al., 2023): LIMA collects a training dataset of 1000 prompts and responses, curated to ensure stylistic consistency while accommodating diverse input types. It also includes an open-source test set of 300 prompts and a development set of 50. The data is primarily sourced from community-driven Q&A platforms like Stack Exchange, wikiHow, and the Pushshift Reddit Dataset (Baumgartner et al., 2020), along with manually curated examples. The inclusion of human-authored examples further increases dataset diversity. In our experiments, we use the LIMA test set to evaluate our models.

- Vicuna. (Chiang et al., 2023): Vicuna organizes its 80 test instructions into eight distinct categories: Fermi problems, commonsense, roleplaying, coding/math/writing tasks, counter-factuals, knowledge, and general questions. This categorization aims to comprehensively assess different facets of chatbot performance. Prior work suggests that Vicuna's instructions are generally of lower complexity and difficulty (Xu et al., 2023). We utilize the Vicuna test set to assess the performance of large language models across these varied categories of instructions.

- Self-Instruct. (Wang et al., 2022): Self-Instruct contains 252 human-authored test instructions, each paired with a well-constructed output. This dataset is curated to simulate real-world use cases of instruction-following models, spanning various domains such as email composition, social media, productivity tools, and coding tasks. The instructions differ in format and complexity, featuring diverse task lengths and output types such as bullet points, tables, code snippets, and mathematical expressions. In our research, we utilized the Self-Instruct test set to rigorously evaluate our model's ability to follow detailed instructions across multiple domains.

- Wizardlm. (Xu et al., 2023): Wizardlm consists of a training set of 70k examples derived from 52k instructions initially provided by Alpaca. The test set includes 218 instructions sourced from various open-source projects and online communities, covering 29 distinct skills derived from real-world tasks. These skills range from Code Generation & Debugging to Reasoning, Mathematics, Writing, Complex Format Handling, and Mastery of Extensive Domains. In our study, we employed the Wizardlm test set to evaluate the model's ability to adhere to detailed instructions comprehensively.

- Koala. (Vu et al., 2023): Koala comprises 180 real-world user queries sourced from the web, spanning diverse topics and typically reflecting a conversational tone. These queries are especially relevant for evaluating models intended for chat-based applications. To ensure no overlap with training data, any query yielding a BLEU score above 20% compared to examples from our training set is excluded. Additionally, queries involving programming or non-English languages are omitted, as our evaluation team, composed of crowd-sourced raters, lacks the expertise to assess such content effectively. We exclusively used the Koala test set to gauge our model's proficiency in handling authentic conversational queries.

## A.7 Aggregation Process in GED Across Different Tasks

The GED implementation differs between response selection and model ranking tasks due to their objectives. In response selection, GED aggregates preference graphs from multiple evaluators for each question into a single DAG. From this DAG, a final ranking is derived, and the top-ranked answer is selected as the output. Notably, aggregation is performed only at the per-question level, without a rank aggregation across questions. In model ranking, GED involves two steps: first, aggregating the evaluators' preference graphs into a DAG for each question to rank the models, and second, employing a rank ensemble method to aggregate these per-question rankings into a final overall model ranking across all questions. For details on the aggregation modeling, see Section 2.3, covering Response Selection, Model Ranking, and Model Alignment.

## B Alternative Evaluator Configurations for Model Ranking

To address concerns about the computational cost and fairness of using 70B-level models as weak evaluators in the model ranking task, we conducted additional experiments using smaller, more comparable models. Table 4 summarizes the performance of these models, both individually and when combined using GED. The results show that GED outperforms individual evaluators even when using smaller 7B-scale models, achieving an average score of 62.70 compared to the best individual performance of 57.92 (Qwen2-7B with graph denoising). This demonstrates that the aggregation of smaller models through GED effectively enhances performance while reducing computational costs. These findings validate the versatility of GED, showing that it can provide robust and accurate rankings without relying solely on large-scale models.

Table 4: Performance comparison in the model ranking task using 7B-scale models as evaluators on the AlpacaEval dataset. Higher values indicate better performance. GED achieves robust performance even with smaller evaluators.

| Model | | Weight Score | Kemeny | Weighted Kemeny | Pairwise Majority | Weighted Pairwise Majority | Avg. |
|---|---|---|---|---|---|---|---|
| | Llama3-8B | 35.88 | 45.80 | 45.80 | 47.23 | 46.85 | 44.31 |
| | with graph denoising | 37.44 | 47.54 | 47.54 | 48.92 | 48.18 | 45.92 |
| | Qwen2-7B | 55.34 | 52.87 | 52.87 | 56.05 | 56.59 | 54.74 |
| Single evaluator | with graph denoising | 56.05 | 57.43 | 57.43 | 59.32 | 59.41 | 57.92 |
| | Mistral-7B | 49.90 | 53.74 | 53.74 | 58.06 | 57.87 | 54.66 |
| | with graph denoising | 50.47 | 55.06 | 54.92 | 61.39 | 61.21 | 56.61 |
| Multiple evaluator | GED | **57.59** | **61.14** | **61.14** | **67.17** | **66.46** | **62.70** |

## C Impact of Evaluator Quantity on Denoising Quality

We investigated how the number of evaluators affects the denoising quality in GED by conducting experiments using different numbers of evaluators in the response selection setting. Specifically, we evaluated the performance of GED when aggregating preferences from two, three, and four evaluators. The evaluators used were Llama3-8B, Mistral-7B, Qwen2-7B, and Gemma-9B[2].

Table 5: Performance comparison of GED with varying numbers of evaluators across five benchmarks. Increasing the number of evaluators enhances denoising quality, reflected in improved performance metrics.

| Evaluators Set | HumanEval | AlpacaEval | MATH | GSM8k | GAIA | Avg. |
|---|---|---|---|---|---|---|
| Llama3-8B, Mistral-7B | 69.21 | 31.87 | 74.97 | 86.92 | 12.51 | 55.10 |
| Llama3-8B, Mistral-7B, Qwen2-7B | 70.73 | 32.44 | 75.58 | 88.18 | 13.33 | 56.05 |
| Llama3-8B, Mistral-7B, Qwen2-7B, Gemma-9B | 70.98 | 32.87 | 75.91 | 88.75 | 13.46 | 56.39 |

---

[2]`https://huggingface.co/google/gemma-2-9b-it`

As shown in Table 5, increasing the number of evaluators consistently improves the denoising quality of GED, as reflected by higher performance across all benchmarks. For instance, the average performance improves from 55.10% with two evaluators to 56.39% with four evaluators. This enhancement can be attributed to the diverse perspectives and complementary strengths of multiple evaluators, contributing to a more robust and accurate aggregation of preferences. These results highlight the importance of selecting a diverse and capable set of evaluators to enhance GED's effectiveness. Incorporating more evaluators allows the denoising process to better identify and mitigate inconsistencies in the preference graphs, leading to improved overall performance in response selection tasks.

## D    CONSTRUCTION OF THE PREFERENCE GRAPH

In this section, we provide a detailed explanation of the process used to construct the preference graph sets for both the model ranking and response selection tasks, as outlined in Algorithms 1 and 2. These algorithms form the backbone of our method, enabling the representation of pairwise preferences as directed graphs, which are essential for downstream aggregation and ranking.

Algorithm 1 describes the construction process for generating a set of preference graphs for the model ranking task. The procedure is as follows: *Initialization*: For each question $q \in Q$, we begin by initializing a vertex set $V_q$, where each vertex $v_i$ corresponds to a model $\mathcal{M}_i$ in the set of models $M$. We also initialize an empty set of edges $A_a$ and a weight function $w_a$, which will be used to track the strength of the preferences between model pairs. *Pairwise Comparisons*: For each pair of models $\mathcal{M}_i$ and $\mathcal{M}_j$, the assigned evaluator $a \in \mathcal{A}$ assesses their responses to the given question $q$. If evaluator $a$ prefers $\mathcal{M}_i$ over $\mathcal{M}_j$, a directed edge $(v_i \to v_j)$ is added to the edge set $A_a$, and its corresponding weight is incremented. Conversely, if $\mathcal{M}_j$ is preferred, the edge $(v_j \to v_i)$ is added or its weight is updated. *Graph Storage*: Once all pairwise comparisons have been processed for a given evaluator, the resulting graph $G_a = (V_q, A_a, w_a)$ is stored for that evaluator. This process is repeated for all evaluators in $\mathcal{A}$ and for all questions in $Q$, generating a set of preference graphs for each evaluator and question.

---

**Algorithm 1** Construction of the Preference Graph for Model Ranking

---

**Require:** Set of models $M = \{\mathcal{M}_1, \mathcal{M}_2, \ldots, \mathcal{M}_n\}$, set of questions $Q = \{q_1, q_2, \ldots, q_t\}$, set of evaluators
   $\mathcal{A} = \{a_1, a_2, \ldots, a_k\}$
**Ensure:** Set of preference graph sets $\{G_a : a \in \mathcal{A}\}$ for each question $q \in Q$
 1: **for** each question $q \in Q$ **do**
 2:    **for** each evaluator $a \in \mathcal{A}$ **do**
 3:       Initialize vertex set $V_q = \{v_1, v_2, \ldots, v_n\}$, where each $v_i$ corresponds to model $\mathcal{M}_i$
 4:       Initialize edge set $A_a = \emptyset$, and weight function $w_a : A_a \to \mathbb{R}^+$
 5:       **for** each pair of models $(\mathcal{M}_i, \mathcal{M}_j)$ with $i \neq j$ **do**
 6:          Let $ans_i = \mathcal{M}_i(q)$ and $ans_j = \mathcal{M}_j(q)$
 7:          **if** $a(ans_i, ans_j) > 0$ **then**                    ▷ evaluator $a$ prefers $\mathcal{M}_i$ over $\mathcal{M}_j$
 8:             **if** $(v_i, v_j) \notin A_a$ **then**
 9:                Add directed edge $(v_i \to v_j)$ to $A_a$
10:                Set $w_a(v_i, v_j) = 1$
11:             **else**
12:                Increment $w_a(v_i, v_j)$ by 1
13:             **end if**
14:          **else**
15:             **if** $(v_j, v_i) \notin A_a$ **then**
16:                Add directed edge $(v_j \to v_i)$ to $A_a$
17:                Set $w_a(v_j, v_i) = 1$
18:             **else**
19:                Increment $w_a(v_j, v_i)$ by 1
20:             **end if**
21:          **end if**
22:       **end for**
23:       Store the preference graph $G_a = (V_q, A_a, w_a)$ for evaluator $a$
24:    **end for**
25: **end for**

---

Algorithm 2 follows a similar structure but applies to the response selection task, where the objective is to rank a set of candidate answers for each question: *Initialization*: For each question $q \in Q$, we initialize a vertex set $V_q$, where each vertex corresponds to a candidate answer $ans_i$. As in the

model ranking task, we also initialize an edge set $A_a$ and a weight function $w_a$ for each evaluator $a \in \mathcal{A}$. *Pairwise Comparisons*: Evaluators compare the quality of pairs of candidate answers $ans_i$ and $ans_j$ for each question. A directed edge is added based on the evaluator's preference, with the weight reflecting the strength of preference. As before, if evaluator $a$ prefers $ans_i$ over $ans_j$, an edge $(v_i \rightarrow v_j)$ is added or its weight incremented, and vice versa. *Graph Storage*: After all pairwise comparisons are complete, the preference graph $G_a = (V_q, A_a, w_a)$ for evaluator $a$ is stored. This procedure is repeated for all evaluators and questions, resulting in a set of preference graphs for each evaluator and each question.

Both algorithms ensure that the preference graphs are constructed in a consistent manner, forming the basis for the aggregation and denoising processes used later in our framework. These graphs encapsulate the evaluators' preferences and provide a structured representation of pairwise comparisons, facilitating further analysis.

---

**Algorithm 2** Construction of the Preference Graph for Response Selection

---

**Require:** Set of candidate answers $\{ans_1, ans_2, \ldots, ans_n\}$ for each question $q \in Q$, set of evaluators $\mathcal{A} = \{a_1, a_2, \ldots, a_k\}$
**Ensure:** Set of preference graph sets $\{G_a : a \in \mathcal{A}\}$ for each question $q \in Q$
1: **for** each question $q \in Q$ **do**
2:     **for** each evaluator $a \in \mathcal{A}$ **do**
3:         Initialize vertex set $V_q = \{v_1, v_2, \ldots, v_n\}$, where each $v_i$ corresponds to $ans_i$
4:         Initialize edge set $A_a = \emptyset$, and weight function $w_a : A_a \rightarrow \mathbb{R}^+$
5:         **for** each pair of answers $(ans_i, ans_j)$ with $i \neq j$ **do**
6:             **if** $a(ans_i, ans_j) > 0$ **then**
7:                 **if** $(v_i, v_j) \notin A_a$ **then**
8:                     Add directed edge $(v_i \rightarrow v_j)$ to $A_a$, set $w_a(v_i, v_j) = 1$
9:                 **else**
10:                     Increment $w_a(v_i, v_j)$ by 1
11:                 **end if**
12:             **else**
13:                 **if** $(v_j, v_i) \notin A_a$ **then**
14:                     Add directed edge $(v_j \rightarrow v_i)$ to $A_a$, set $w_a(v_j, v_i) = 1$
15:                 **else**
16:                     Increment $w_a(v_j, v_i)$ by 1
17:                 **end if**
18:             **end if**
19:         **end for**
20:         Store the preference graph $G_a = (V_q, A_a, w_a)$ for evaluator $a$
21:     **end for**
22: **end for**

---

# E   PROOF OF THEOREM 1

**Theorem 1** *Suppose* $G_1, \ldots, G_N \overset{i.i.d.}{\sim} \mathcal{G}(G, \delta_1, \delta_2)$ *for some ground truth* $G = (V, A)$. *Let* $\widehat{G}$ *be the graph ensembled from* $G_1, \ldots, G_N$ *by operations defined in Section 2.2. Then, as long as* $\delta_1 = 0.5 - \epsilon$. *For some* $\epsilon > 0$, *we have*

$$\mathbb{P}\left(G \subseteq MAS(\widehat{G})\right) \geq 1 - 2|A|\exp\left(-\frac{N\epsilon^2}{2}\right) - 2U\exp\left(-\frac{N\epsilon^2}{6U^2\delta_2 + 2U\epsilon}\right),$$

*where* $G \subseteq MAS(\widehat{G})$ *represents that* $G$ *is a subgraph of* $MAS(\widehat{G})$ *and* $U = \frac{|V|(|V|-1)}{2} - |A|$ *is the number of pairs of unconnected nodes in* $G$.

**Proof E.1** *For brevity, we consider all edges in* $G$ *have weights equal to 1 and all weights in* $\widehat{G}$ *are divided by* $N$. *By construction, we can notice that for each* $(u \rightarrow v) \in A$, *the weight* $w_{\widehat{G}}(v \rightarrow u)$ *can be viewed as an empirical estimate of* $\delta_1$. *Then, we claim that the following two events can imply* $G \subseteq MAS(\widehat{G})$:

- ($\mathcal{E}_1$) *For any* $(u \rightarrow v) \in A$, *it holds* $\left|w_{\widehat{G}}(v \rightarrow u) - \delta_1\right| \leq \frac{\epsilon}{2}$.

- ($\mathcal{E}_2$) *For any pair of nodes* $(u, v)$ *such that* $(u \rightarrow v), (v \rightarrow u) \notin A$, *it holds* $\left|w_{\widehat{G}}(u \rightarrow v) - w_{\widehat{G}}(v \rightarrow u)\right| \leq \frac{\epsilon}{U}$.

*To see this, first by Lemma 1, we know that for any pair of nodes $(u, v)$, $\mathrm{MAS}(\widehat{G})$ will contain exactly one of $(u \to v)$ and $(v \to u)$.[3] Therefore, for any $(u \to v) \in A$, $\mathrm{MAS}(\widehat{G})$ will contain exactly one of $(u \to v)$ and $(v \to u)$. Then, since $\mathcal{E}_1$ holds, $\delta_1 = 0.5 - \epsilon < 0.5$ and $w_{\widehat{G}}(u \to v) + w_{\widehat{G}}(v \to u) = 1$, we have $w_{\widehat{G}}(u \to v) - w_{\widehat{G}}(v \to u) \geq \epsilon$. Furthermore, since $\mathcal{E}_2$ holds, for $(u, v)$ such that $(u \to v), (v \to u) \notin A$, arbitrary way of edge removing among these nodes can influence the total edge weights by at most $\epsilon$. Therefore, when applying the denoising operation to $\widehat{G}$, for any $(u \to v) \in A$, only $(u \to v)$ will be kept in $\mathrm{MAS}(\widehat{G})$, which makes $G \subseteq \mathrm{MAS}(\widehat{G})$. As a result, we have $\mathbb{P}(\mathcal{E}_1 \cap \mathcal{E}_2) \leq \mathbb{P}\left(G \subseteq \mathrm{MAS}(\widehat{G})\right)$.*

*Then, we can now bound the probability of $\mathcal{E}_1 \cap \mathcal{E}_2$. In particular, for fixed $(u \to v) \in A$, since $w_{\widehat{G}}(v \to u)$ is an empirical mean estimate of $\delta_1$, by Hoeffding's inequality, we have*

$$\mathbb{P}\left(\left|w_{\widehat{G}}(v \to u) - \delta_1\right| \leq \frac{\epsilon}{2}\right) \geq 1 - 2\exp\left(-N\epsilon^2/2\right)$$
$$\implies \mathbb{P}(\mathcal{E}_1) \geq 1 - 2|A|\exp\left(-N\epsilon^2/2\right),$$

*where the second inequality comes from the union bound over all edges in $A$. Similarly, for fixed node pair $(u, v)$ that is unconnected in $G$, $w_{\widehat{G}}(u \to v) - w_{\widehat{G}}(v \to u)$ can be viewed as $\frac{1}{N}\sum_{i=1}^{N} X_i$, where $X_i$'s are i.i.d. and*

$$X_i = \begin{cases} 1, & \text{with probability } \frac{\delta_2}{2} \\ -1, & \text{with probability } \frac{\delta_2}{2} \\ 0, & \text{with probability } 1 - \delta_2 \end{cases}.$$

*Therefore, by Bernstein's inequality, we have*

$$\mathbb{P}\left(\left|w_{\widehat{G}}(u \to v) - w_{\widehat{G}}(v \to u)\right| \leq \frac{\epsilon}{U}\right) \geq 1 - 2\exp\left(-\frac{N\epsilon^2}{6U^2\delta_2 + 2U\epsilon}\right)$$
$$\implies \mathbb{P}(\mathcal{E}_2) \geq 1 - 2U\exp\left(-\frac{N\epsilon^2}{6U^2\delta_2 + 2U\epsilon}\right),$$

*where the second inequality is an union bound over all unconnected node pairs in $G$. As a result, we eventually have*

$$\mathbb{P}\left(G \subseteq \mathrm{MAS}(\widehat{G})\right) \geq \mathbb{P}\left(\mathcal{E}_1 \cap \mathcal{E}_2\right) \geq 1 - 2|A|\exp\left(-\frac{N\epsilon^2}{2}\right) - 2U\exp\left(-\frac{N\epsilon^2}{6U^2\delta_2 + 2U\epsilon}\right).$$

**Lemma 1** *For a weighted directed graph $G = (V, A, w)$, if $(u \to v), (v \to u) \in A$, then $\mathrm{MAS}(G)$ contains exactly one of $(u \to v)$ and $(v \to u)$.*

**Proof E.2** *Recall that $\mathrm{MAS}(G)$ gives an acyclic subgraph of $G$ with the maximum weight. Since it has to be acyclic, it is obvious that $\mathrm{MAS}(G)$ cannot contain both $(u \to v)$ and $(v \to u)$.*

*We will then use contradiction to show it is impossible for $\mathrm{MAS}(G)$ to contain neither $(u \to v)$ nor $(v \to u)$. Suppose this is true instead. Then, since $\mathrm{MAS}(G)$ is a maximum acyclic subgraph, adding either $(u \to v)$ or $(v \to u)$ to $\mathrm{MAS}(G)$ will make it cyclic. That is, $\mathrm{MAS}(G)$ contains a path that goes from $v$ to $u$; meanwhile, it also contains a path from $u$ to $v$. As a result, it contains a cycle that goes from $v$ to $u$ and then from $u$ to $v$, which contradicts with the fact that $\mathrm{MAS}(G)$ is a maximum acyclic subgraph. Therefore, $\mathrm{MAS}(G)$ must contain exactly one of $(u \to v)$ and $(v \to u)$.*

# F  IMPACT OF EVALUATOR WEIGHTING ON GED PERFORMANCE

In our theoretical analysis, GED assumes equal weighting of edges in the preference graphs. However, in practical scenarios, evaluators may have varying levels of reliability or expertise. Incorporating evaluator-specific confidence scores or performance metrics could enhance the effectiveness of GED. To investigate this, we conducted experiments using a weighted version of GED, referred to as

---

[3]There is non-zero probability that some edges in $\widehat{G}$ will have zero weight, but we treat them as existing for the ease of argument. That is, we allow only $\widehat{G}$ to contain zero-weight edges.

Table 6: Performance comparison of GED and WeightGED using 7B-scale evaluators across five benchmarks. WeightGED marginally outperforms GED, demonstrating the potential benefits of incorporating evaluator-specific weights.

| Method | | HumanEval | AlpacaEval | MATH | GSM8k | GAIA | Avg. |
|---|---|---|---|---|---|---|---|
| **7B Evaluators** | GED | 70.73 | 32.44 | 75.58 | 88.18 | 13.33 | 56.05 |
| | WeightGED | **70.97** | **32.67** | **75.71** | **88.24** | **13.56** | **56.23** |
| **GPT Evaluators** | GED | 73.21 | 59.87 | 82.49 | 86.43 | 16.27 | 63.65 |
| | WeightGED | **74.52** | **61.71** | **83.93** | **87.84** | **17.32** | **65.06** |

WeightGED, under the Response Selection setting. In WeightGED, we assigned weights to evaluators based on their individual performance on specific datasets. For example, on the GSM8k dataset, the response selection accuracies for Llama3-8B, Mistral-7B, and Qwen2-7B were 62.19%, 67.24%, and 61.58%, respectively. These accuracies were normalized to compute evaluator weights:

$$\text{weight(Llama3-8B)} = \frac{62.19}{62.19 + 67.24 + 61.58} = 0.326,$$

$$\text{weight(Mistral-7B)} = \frac{67.24}{62.19 + 67.24 + 61.58} = 0.352,$$

$$\text{weight(Qwen2-7B)} = \frac{61.58}{62.19 + 67.24 + 61.58} = 0.322.$$

These weights were used to scale the contributions of each evaluator's preferences during graph construction. We compared the performance of GED and WeightGED across multiple benchmarks, as presented in Table 6.

As shown in Table 6, WeightGED achieves marginal but consistent improvements over GED across all benchmarks. For instance, using 7B-scale evaluators, WeightGED improves the average performance from 56.05% to 56.23%. Similarly, with GPT evaluators, the average performance increases from 63.65% to 65.06%. These results suggest that incorporating evaluator-specific weights based on performance metrics can enhance the effectiveness of GED. Furthermore, we conducted additional experiments using stronger evaluators such as GPT-3.5, GPT-4-o-mini, and GPT-4-o. The weights were computed based on their respective performance accuracies, following the same normalization procedure. The improvements observed with these evaluators reinforce the potential of weighting schemes in enhancing GED. In summary, our preliminary findings indicate that integrating evaluator-specific confidence scores or performance metrics is a promising direction for future work. Systematically designing and optimizing these weighting schemes could lead to more robust and accurate evaluation frameworks.

## G EVALUATING GED ON DIVERSE METRICS

To address the need for more comprehensive assessments beyond accuracy-based metrics, we expanded our evaluation of GED to include nuanced quality aspects of LLM-generated outputs, such as factuality, relevance, coherence, informativeness, helpfulness, and validity. This evaluation was motivated by the understanding that tasks involving LLMs often require subtle judgments beyond simple accuracy, and GED's adaptability to these scenarios is crucial. Building upon this foundation, we conducted experiments following (Chen et al., 2023a). Specifically, we used Llama3-70B to generate ten candidate responses for each query, and GED was applied in the response selection setting with Llama3-8B, Mistral-7B, and Qwen2-7B serving as evaluators. Each metric—factuality, relevance, coherence, informativeness, helpfulness, and validity—was assessed to measure GED's ability to enhance the overall quality of LLM outputs. The results, shown in Table 7, highlight GED's effectiveness. GED consistently outperformed individual evaluators and random selection across all metrics. For instance, GED improved factuality by approximately 5 percentage points over the best individual evaluator. Similarly, it enhanced relevance and coherence, indicating better alignment with the query and logically consistent responses. GED also demonstrated notable improvements

in informativeness and helpfulness, suggesting that the responses selected by GED provide more valuable and user-centric information. These findings confirm GED's adaptability to tasks requiring nuanced quality assessments. By aggregating preferences from multiple evaluators, GED captures subtle qualities of generated content, enhancing not only accuracy but also the overall reliability and utility of LLM outputs. This adaptability reinforces GED's value in scenarios demanding comprehensive evaluations of language models.

Table 7: Performance comparison on nuanced quality metrics (in %). GED outperforms individual evaluators and random selection across Factuality, Relevance, Coherence, Inform., Helpful. and Validity metrics.

| Method | Factuality | Relevance | Coherence | Inform. | Helpful. | Validity | Avg. |
|--------|-----------|-----------|-----------|---------|----------|----------|------|
| Random | 86.73 | 87.91 | 92.47 | 77.62 | 17.48 | 48.92 | 68.52 |
| Llama3-8B | 88.59 | 89.91 | 94.41 | 79.77 | 18.48 | 50.92 | 70.35 |
| Mistral-7B | 89.10 | 90.29 | 94.85 | 79.95 | 18.55 | 51.13 | 70.65 |
| Qwen2-7B | 89.25 | 90.44 | 95.03 | 80.09 | 18.58 | 51.21 | 70.77 |
| GED | **94.73** | **95.91** | **97.36** | **86.62** | **19.48** | **55.92** | **75.00** |

## H  EXPANDED RELATED WORK ON PREFERENCE DENOISING FOR LLMS

This work situates itself within the broader field of denoising preference evaluations for LLMs, addressing inconsistencies and noise in preference graphs. We acknowledge that existing literature has explored two primary approaches to this challenge: within-model preference modeling and modular pre-processing. Below, we provide a detailed comparison of GED with representative methods from these approaches.

Within-model approaches, such as robust DPO (rDPO) (Chowdhury et al., 2024) and conservative DPO (cDPO) (Mitchell, 2023), focus on integrating denoising mechanisms directly within the preference modeling process. These methods incorporate regularization techniques to mitigate the effects of noisy or adversarial preference data during model alignment. While these approaches are effective in refining the preference modeling pipeline, they are tightly coupled with specific models and tasks, limiting their versatility. In contrast, GED is a modular framework that operates as a pre-processing step, denoising preference graphs before downstream tasks, making it adaptable to a wide range of applications and models. Modular approaches like CURATRON (Naresh et al., 2024) address noise and missing comparisons in preference datasets using techniques such as low-rank matrix completion. While CURATRON effectively mitigates certain types of noise, it does not explicitly target cyclic inconsistencies (e.g., $A \succ B, B \succ C, C \succ A$), which are a critical focus of GED. By leveraging a graph-based framework, GED detects and removes such cycles through its denoising process, ensuring that the resulting preference graph is acyclic and thus more consistent and reliable for downstream use. Additionally, GED distinguishes itself by providing theoretical guarantees for recovering the ground truth preference DAG under certain conditions. Furthermore, the ensemble mechanism in GED demonstrates the novel insight that combining weaker evaluators can surpass the performance of a single stronger evaluator, a feature not emphasized in the aforementioned methods.

## I  IMPACT OF EVALUATOR SELECTION ON GED PERFORMANCE

The selection of weak evaluators is a critical factor influencing the performance of GED. While our initial experiments demonstrated the benefits of combining weaker evaluators, further analysis is necessary to understand how their diversity and capabilities affect GED's effectiveness. To address this, we conducted additional experiments evaluating three different configurations of evaluator sets: the original set comprising Llama3-8B, Mistral-7B, and Qwen2-7B; a diverse set that adds Gemma-9B[4] to introduce more variation in model architecture and training data; and a higher-capability set that replaces smaller models with larger ones, specifically Llama3-70B, Mistral-8×7B, and Qwen2-72B. The results, shown in Table 8, reveal two key insights. First, incorporating diversity

---

[4] https://huggingface.co/google/gemma-2-9b-it

Table 8: Performance comparison with different evaluator sets across five benchmarks. The results highlight the impact of evaluator diversity and capability on GED's effectiveness.

| Evaluators Set | HumanEval | AlpacaEval | MATH | GSM8k | GAIA | Avg |
|---|---|---|---|---|---|---|
| Original | 70.73 | 32.44 | 75.58 | 88.18 | 13.33 | 56.05 |
| Diverse | 70.98 | 32.87 | 75.91 | 88.75 | 13.46 | 56.39 |
| Higher Capability | 74.73 | 47.92 | 75.58 | 90.04 | 16.21 | 60.89 |

Table 9: Performance comparison of ActiveGED under different budget constraints. ActiveGED achieves competitive performance with significantly fewer pairwise evaluations.

| Evaluators Set of GED | HumanEval | AlpacaEval | MATH | GSM8k | GAIA | Avg. |
|---|---|---|---|---|---|---|
| Random | 57.93 | 29.58 | 72.75 | 84.67 | 11.52 | 51.29 |
| ActiveGED (30%) | 67.28 | 30.96 | 74.65 | 85.73 | 11.39 | 54.00 |
| ActiveGED (50%) | 68.62 | 31.91 | 74.87 | 87.06 | 12.08 | 54.91 |
| GED (Full Budget) | **70.73** | **32.44** | **75.58** | **88.18** | **13.33** | **56.05** |

by adding Gemma-9B improves GED's performance slightly, increasing the average score from 56.05% to 56.39%. This suggests that models with diverse training data and architectures can contribute to more robust aggregated evaluations. Second, replacing smaller evaluators with higher-capability models yields a more substantial improvement, with the average score rising to 60.89%. Notably, benchmarks like AlpacaEval and GAIA benefit the most from the advanced reasoning and language understanding capabilities of larger models. These findings demonstrate the importance of both diversity and capability in evaluator selection. Diversity provides marginal gains by bringing varied perspectives, while higher-capacity models contribute to more significant improvements by enhancing the overall quality of evaluations. This suggests that, when computational resources allow, incorporating advanced models into the evaluator set can meaningfully boost GED's performance.

## J COST CONSIDERATIONS AND ACTIVE LEARNING WITH ACTIVEGED

Using multiple evaluators and aggregating their preferences can be computationally expensive, especially when constructing dense preference graphs. This limitation becomes more pronounced in scenarios where preferences across all pairs of evaluators are required, which scales quadratically with the number of responses or models being compared. To address this, we clarify the computational trade-offs and propose an active learning-based approach to reduce the number of required pairwise evaluations. To further reduce the number of pairwise evaluations needed, we developed an active learning algorithm called **ActiveGED**. This algorithm strategically selects the most informative pairs for evaluation, effectively lowering the overall computational cost while maintaining high performance. ActiveGED combines random sampling with uncertainty-based selection to maximize information gain from each additional pairwise evaluation. We evaluated ActiveGED under budget constraints of 30% and 50% of the total possible pairwise comparisons, where we set the random budget ratio as 0.5. The results, presented in Table 9, demonstrate that ActiveGED achieves competitive performance with significantly fewer evaluations. For example, under a 50% budget, ActiveGED retains most of the performance benefits of full GED while cutting the number of pairwise comparisons in half.

The algorithm behind ActiveGED is outlined in Algorithm 3. It begins by randomly selecting a portion of the budget to initialize the preference graph and then iteratively selects the most informative edges based on uncertainty, as estimated using PageRank scores. This approach balances exploration (random sampling) and exploitation (uncertainty-based selection) to efficiently construct an accurate preference graph.

ActiveGED demonstrates that by carefully selecting the most informative pairs, it is possible to achieve competitive performance with significantly fewer evaluations. This makes GED more scalable and practical for real-world applications where computational resources are limited.

---

**Algorithm 3** ActiveGED

---

**Require:** Candidate set $V = \{v_1, v_2, \ldots, v_n\}$; Initial preference graph $G = (V, A, w)$; Evaluator
     set $E = \{\text{ev}_1, \ldots, \text{ev}_k\}$; Total budget $B$; Random budget ratio $\alpha \in (0, 1)$
**Ensure:** Updated preference graph $G^* = (V, A^*, w^*)$
  1: Randomly select $m_{\text{rand}} = \alpha B$ edges from $(V \times V) \setminus A$ to form $A_{\text{rand}}$
  2: **for** each edge $(u, v) \in A_{\text{rand}}$ **do**
  3:      **for** each evaluator $\text{ev}_i \in E$ **do**
  4:          Obtain preference weight $w_i(u, v)$
  5:      **end for**
  6:      Aggregate weights $w(u, v) = \frac{1}{k} \sum_{i=1}^{k} w_i(u, v)$
  7: **end for**
  8: Update $A^* \leftarrow A \cup A_{\text{rand}}$
  9: Set current budget $b \leftarrow m_{\text{rand}}$
10: Compute PageRank $PR(v)$ for all $v \in V$ using $G^* = (V, A^*, w^*)$
11: **while** $b < B$ **do**
12:      **for** each unevaluated edge $(u, v) \in (V \times V) \setminus A^*$ **do**
13:          Estimate uncertainty $U(u, v)$ based on current PageRank scores
14:      **end for**
15:      Select edge $(u^*, v^*)$ with highest $U(u, v)$
16:      **for** each evaluator $\text{ev}_i \in E$ **do**
17:          Obtain preference weight $w_i(u^*, v^*)$
18:      **end for**
19:      Aggregate weights $w(u^*, v^*) = \frac{1}{k} \sum_{i=1}^{k} w_i(u^*, v^*)$
20:      Update $A^* \leftarrow A^* \cup \{(u^*, v^*)\}$
21:      Increment $b \leftarrow b + 1$
22:      Recompute PageRank $PR(v)$ for all $v \in V$ using the updated $G^*$
23: **end while**
24: **return** $G^* = (V, A^*, w^*)$

---

## K   MITIGATING EVALUATOR BIASES IN GED

Evaluator biases, such as position bias and token bias, can significantly impact the accuracy and fairness of the preference graphs used in GED. Addressing these biases is crucial for ensuring reliable evaluations and robust performance. In this section, we describe the strategies employed in our framework to mitigate these biases and discuss potential areas for future improvement. Position bias arises when evaluators exhibit a preference for a particular position in a pairwise comparison, such as consistently favoring the first or second option regardless of content. To counter this, we explicitly include both orderings of each question and its candidate answers in the response selection setting. Specifically, for a question $Q$ with two candidate answers $A_1$ and $A_2$, we evaluate both configurations:

- **Order 1:**

    ```
    Question: [Question Text]

    Answer 1: [Answer Text A1]

    Answer 2: [Answer Text A2]

    Which one is better?
    ```

- **Order 2:**

    ```
    Question: [Question Text]

    Answer 1: [Answer Text A2]

    Answer 2: [Answer Text A1]
    ```

Which one is better?

By testing both (Q, A1, A2) and (Q, A2, A1), we ensure that any positional preferences of the evaluators are balanced out when constructing the preference graph. This approach minimizes the impact of position bias on the overall rankings. Token bias occurs when an evaluator exhibits a latent preference for a specific option or label (e.g., consistently favoring "Option A" over "Option B"). Our framework implicitly mitigates token bias through the aforementioned position-swapping strategy, which prevents evaluators from associating a fixed label with a specific position. By averaging the results across both orderings, any systematic preference for a particular label is effectively neutralized.

## L  DENOISING OF PREFERENCE GRAPHS FOR GED

In this section, we describe the denoising procedure used in the GED framework, specifically for transforming the aggregated preference graph into a DAG. Let $G = (V, A)$ be a simple connected directed graph, where $n = |V|$ and $m = |A|$, with $n - 1 \leq m \leq \binom{n}{2}$. A *feedback arc set* (FAS) of $G$, denoted as $R(G)$, is a set of arcs whose removal transforms $G$ into a DAG. The *minimum feedback arc set* $R^*(G)$ is the FAS of minimum cardinality, and finding $R^*(G)$ is the well-known *FAS problem*.

Consider a scenario where the vertices of $G$ are arranged sequentially along a horizontal line and labeled as $v_1, v_2, \ldots, v_n$ from left to right. This arrangement is referred to as a *vertex sequence* and is denoted by $s = v_1 v_2 \ldots v_n$. Each vertex sequence $s$ induces a feedback arc set $R(s)$, consisting of all leftward arcs $v_j \rightarrow v_i$ for $j > i$. The FAS problem can therefore be reformulated as finding a vertex sequence $s^*$ such that $R(s^*) = R^*(G)$. Our proposed algorithm, GED, computes a vertex sequence $s$ that corresponds to a minimal feedback arc set $R(s)$. The algorithm iteratively removes vertices (and their incident arcs) from $G$, focusing on sinks, sources, and vertices that maximize a specific property. For any vertex $u \in V$, let $d(u)$ denote its degree, $d^+(u)$ its outdegree, and $d^-(u)$ its indegree, such that $d(u) = d^+(u) + d^-(u)$. At each step, after removing sinks and sources, the algorithm selects a vertex $u$ for which $\delta(u) = d^+(u) - d^-(u)$ is maximized. If the removed vertex $u$ is a sink, it is concatenated with a vertex sequence $s_2$; otherwise, it is concatenated with $s_1$. Once $G$ is reduced to an empty graph, the final vertex sequence $s$ is obtained by concatenating $s_1$ and $s_2$. The detailed steps of the algorithm are shown in Algorithm 4. In our GED framework, this denoising step is essential for ensuring that the aggregated preference graph becomes acyclic, thus enabling a reliable ranking to be extracted. By iteratively removing vertices based on their structural properties, we minimize the feedback arc set and ensure that the remaining graph is a DAG, which can be directly used to generate rankings in subsequent steps.

## M  RANK ENSEMBLE METHOD

**Weight score (Adler et al., 2002)**   : The Weight Score method assigns a score to each vertex $v$ based on its position in each ranking $\mathcal{R}_i$. For a vertex $v$ in ranking $\mathcal{R}_i$, the score is given by:

$$S_i(v) = l_i - r_i(v) + 1 \tag{6}$$

where $l_i$ is the length of ranking $\mathcal{R}_i$ and $r_i(v)$ is the rank of vertex $v$ in $\mathcal{R}_i$. If $v$ is not present in $\mathcal{R}_i$, $S_i(v) = 0$. The total score for each vertex across all rankings is:

$$T(v) = \sum_{i=1}^{k} S_i(v) \tag{7}$$

The final consensus ranking $\mathcal{R}^*$ is obtained by sorting the vertices $v$ in descending order of $T(v)$.

**Kemeny and weighted Kemeny (Kemeny, 1959)**   : The Kemeny method seeks a consensus ranking $\mathcal{R}^*$ that minimizes the total pairwise disagreements between $\mathcal{R}^*$ and the input rankings, measured using the Kendall $\tau$-distance:

---

**Algorithm 4** Preference Graphs Denoising for GED

---

**Require:** $G$: Directed graph, **var** $s$: Vertex sequence
1:  $s_1 \leftarrow \emptyset$
2:  $s_2 \leftarrow \emptyset$
3: **while** $G \neq \emptyset$ **do**
4:     **while** $G$ contains a sink **do**
5:         Choose a sink $u$
6:         $s_2 \leftarrow u \cdot s_2$
7:         $G \leftarrow G - u$
8:     **end while**
9:     **while** $G$ contains a source **do**
10:        Choose a source $u$
11:       $s_1 \leftarrow s_1 \cdot u$
12:       $G \leftarrow G - u$
13:    **end while**
14:    Choose a vertex $u$ with maximum $\delta(u)$
15:    $s_1 \leftarrow s_1 \cdot u$
16:    $G \leftarrow G - u$
17: **end while**
18: $s \leftarrow s_1 \cdot s_2$

---

$$\mathcal{R}^* = \arg\min_{\mathcal{R}} \sum_{i=1}^{k} \tau(\mathcal{R}, \mathcal{R}_i) \tag{8}$$

The Weighted Kemeny method introduces a weight $\alpha_i$ for each ranking $\mathcal{R}_i$, reflecting its importance or reliability:

$$\mathcal{R}^* = \arg\min_{\mathcal{R}} \sum_{i=1}^{k} \alpha_i \cdot \tau(\mathcal{R}, \mathcal{R}_i) \tag{9}$$

Here, the goal is to minimize the weighted Kendall tau distance, emphasizing rankings with higher weights.

**Pairwise majority and weighted pairwise majority (Caragiannis et al., 2016)**   : The Pairwise Majority (PM) method determines a consensus ranking $\mathcal{R}^*$ by maximizing the number of pairwise agreements with the input rankings. For each pair of vertices $(v_i, v_j)$, the goal is to ensure that the majority of rankings agree with their relative order in $\mathcal{R}^*$:

$$\mathcal{R}^* = \arg\max_{\mathcal{R}} \sum_{i<j} \left( \sum_{p=1}^{k} \mathbb{1}(\mathcal{R}_p(v_i) < \mathcal{R}_p(v_j)) \right) \cdot \mathbb{1}(\mathcal{R}(v_i) < \mathcal{R}(v_j)) \tag{10}$$

The Weighted Pairwise Majority method incorporates weights $\alpha_p$ to account for the reliability of each ranking $\mathcal{R}_p$:

$$\mathcal{R}^* = \arg\max_{\mathcal{R}} \sum_{i<j} \left( \sum_{p=1}^{k} \alpha_p \cdot \mathbb{1}(\mathcal{R}_p(v_i) < \mathcal{R}_p(v_j)) \right) \cdot \mathbb{1}(\mathcal{R}(v_i) < \mathcal{R}(v_j)) \tag{11}$$

In both methods, the objective is to maximize the (weighted) pairwise agreement between the consensus ranking and the input rankings.

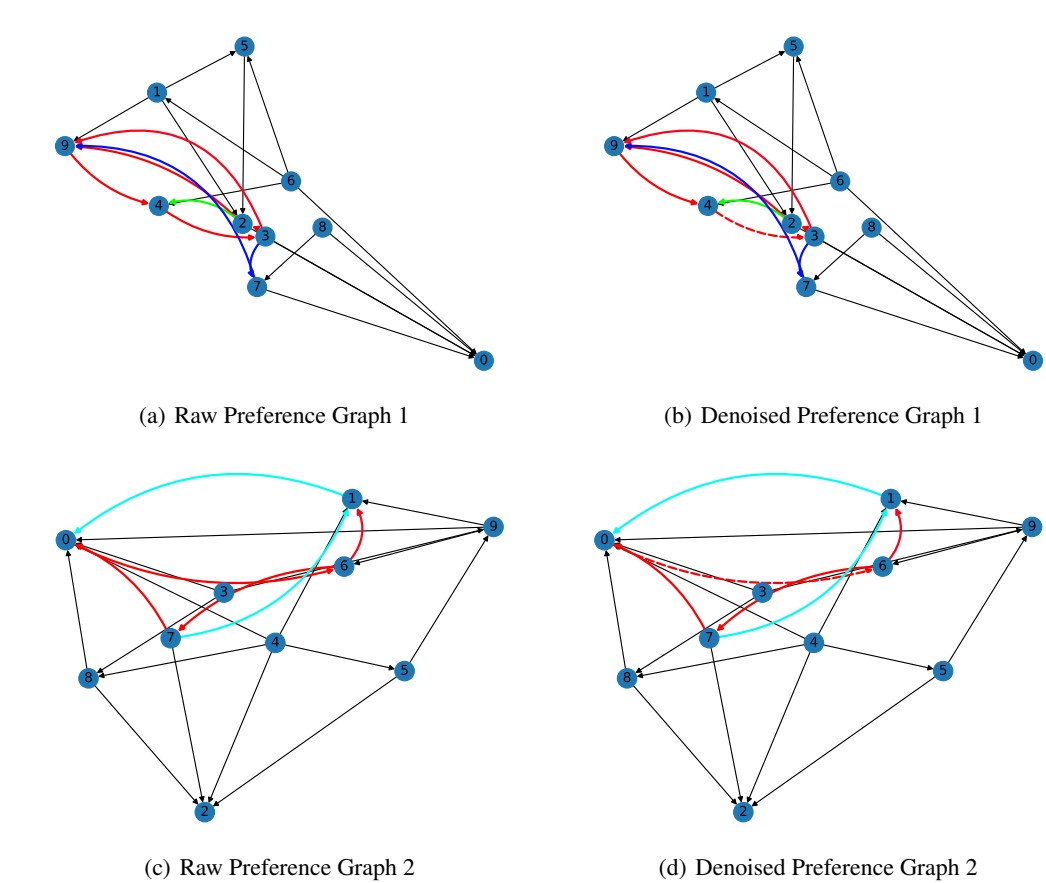

(a) Raw Preference Graph 1

(b) Denoised Preference Graph 1

(c) Raw Preference Graph 2

(d) Denoised Preference Graph 2

Figure 6: Case studies showcasing the raw and denoised preference graphs. In both Case 1 and Case 2, the raw preference graphs (a, c) contain cyclic inconsistencies, which are resolved by GED into directed acyclic graphs (b, d). The dashed lines in the denoised graphs represent the edges that were removed by GED to eliminate cycles. The nodes labeled 0-9 correspond to the ten generated responses by Qwen2-72B. These examples illustrate the effectiveness of GED in eliminating noise and restoring consistency in preference evaluations.

# N VISUALIZATION

In this section, we present two case studies that demonstrate the effectiveness of our proposed GED method in denoising preference graphs. Figure 6 illustrates the raw preference graphs, which are generated by multiple evaluators, i.e., Llama3-8B, Mistral-7B, and Qwen2-7B, through the responses produced by Qwen2-72B on the HumanEval benchmark. The nodes labeled 0-9 in the graphs correspond to the ten generated responses. The comparison between the raw (left) and denoised (right) graphs shows how our method successfully resolves cyclic inconsistencies, transforming noisy graphs into DAGs. In the denoised graphs, the dashed lines represent the edges that were removed by GED to eliminate cycles.

**Case Study 1.** In the first case study, we showcase the impact of our denoising approach. The raw graph (Figure 6 (a)) contains multiple cyclic inconsistencies, such as $9 \succ 4 \succ 3 \succ 9$, $9 \succ 4 \succ 3 \succ 2 \succ 9$, and $9 \succ 4 \succ 3 \succ 7 \succ 9$. By applying GED, we identify that removing a single edge ($4 \succ 3$) eliminates all cycles, converting the noisy preference graph into a consistent DAG, as shown in Figure 6 (b).

**Case Study 2.** The second case study (Figure 6 (c)) presents another scenario with conflicting preferences, where cycles like $7 \succ 0 \succ 6 \succ 7$ and $7 \succ 1 \succ 0 \succ 6 \succ 7$ indicate noisy judgments.

Table 10: Prompt template for evaluating programming solutions on the HumanEval dataset.

> **Prompt for HumanEval**
>
> **System Prompt:**
> You are an expert programmer and code reviewer. Your task is to evaluate code solutions for programming problems. Assess each solution based on its correctness, efficiency, readability, and adherence to best coding practices.
>
> **User Prompt:**
> Please compare the following two code solutions to the given programming problem. For each solution, evaluate whether it produces correct outputs for all edge cases, whether it is efficient in terms of time and space complexity, and whether the code is clean, well-documented, and follows best practices. Identify any errors or areas for improvement.
> **Programming Problem:** [Problem Description]
> **Solution A:** [Candidate Solution A]
> **Solution B:** [Candidate Solution B]
> **Question:** Which solution is better and why? Provide a detailed comparison focusing on correctness, efficiency, readability, and coding standards.

Here, removing just one edge ($0 \succ 6$) using GED is sufficient to eliminate all cycles and convert the graph into a DAG, as depicted in Figure 6 (d).

**Conclusion from Case Studies.** In both case studies, removing any other edge would not fully resolve all cyclic inconsistencies without requiring additional deletions, which would result in more information loss. GED effectively minimizes edge removals while maintaining the integrity of the original preference graph, making it a highly efficient solution for improving the consistency of preference evaluations.

## O PROMPT TEMPLATE

**Prompt for Response Selection.** In this section, we provide detailed prompt templates used for response selection across five datasets: HumanEval (Chen et al., 2021), AlpacaEval (Dubois et al., 2024b), MATH (Hendrycks et al., 2021), GSM8k (Chen et al., 2021), and GAIA (Mialon et al., 2023). These prompts include both a system prompt to establish the evaluator's context and a user prompt tailored to the specific task requirements. Each prompt is designed to guide the evaluators in comparing two candidate responses based on task-specific criteria such as correctness, clarity, efficiency, relevance, and completeness. The templates are shown in Table 10, Table 11, Table 12, Table 13, and Table 14.

**Prompt for Model Ranking.** This section presents the prompt template used in the Model Ranking task. The template is designed to evaluate and compare responses generated by different models for a given instruction, based on criteria such as accuracy, clarity, completeness, and helpfulness. The evaluation process involves analyzing two candidate responses and identifying which one better fulfills the requirements of the instruction. The detailed prompt for the Model Ranking is provided in Table 15.

**Prompt for Instruct Tuning.** In this section, we provide the prompt template used for data selection in the Instruct Tuning task. The goal is to select the most appropriate response for each instruction from multiple candidates, ensuring that the selected responses are helpful, harmless, and relevant. We use the HH-RLHF dataset (Bai et al., 2022), which contains human preference data aimed at training language models to be both helpful and harmless. The detailed prompt used by evaluators to assess and compare candidate responses is presented in Table 16.

Table 11: Prompt template for assessing instruction-following responses on the AlpacaEval dataset.

> **Prompt for AlpacaEval**
>
> **System Prompt:**
> You are an AI assistant trained to assess and compare responses to user instructions. Your evaluations should be based on accuracy, clarity, completeness, and helpfulness.
>
> **User Prompt:**
> Please compare the following two responses to the given instruction. Analyze each response for how well it follows the instruction, the accuracy of the information provided, the clarity of the explanation, and the overall helpfulness to the user. Point out any errors, omissions, or areas where the response could be improved.
> **Instruction:** [Instruction Text]
> **Response A:** [Candidate Response A]
> **Response B:** [Candidate Response B]
> **Question:** Which response better addresses the instruction and why? Provide a detailed comparison focusing on the criteria mentioned above.

Table 12: Prompt template for evaluating mathematical solutions on the MATH dataset.

> **Prompt for MATH**
>
> **System Prompt:**
> You are a mathematician and educator skilled at evaluating mathematical solutions. Assess the correctness, completeness, and clarity of the following solutions to the math problem. Pay attention to the logical reasoning steps, the mathematical accuracy, and the clarity of explanations.
>
> **User Prompt:**
> Please evaluate the following two solutions to the given math problem. For each solution, analyze whether the reasoning is correct, if all necessary steps are included, and if the explanations are clear and easy to understand. Identify any errors or misconceptions.
> **Math Problem:** [Problem Description]
> **Solution A:** [Candidate Solution A]
> **Solution B:** [Candidate Solution B]
> **Question:** Which solution is better and why? Provide a detailed comparison focusing on correctness, completeness, and clarity.

Table 13: Prompt template for assessing multi-step reasoning answers on the GSM8k dataset.

**Prompt for GSM8k**

**System Prompt:**
You are a teacher specializing in elementary mathematics. Evaluate student answers to math word problems for correctness and quality of reasoning. Consider whether the student has correctly understood the problem, applied appropriate mathematical operations, and provided clear explanations for each step.

**User Prompt:**
Please compare the following two answers to the given math word problem. For each answer, assess the accuracy of the solution, the appropriateness of the reasoning steps, and the clarity of the explanations. Highlight any mistakes or areas for improvement.
**Math Word Problem:** [Problem Description]
**Answer A:** [Candidate Answer A]
**Answer B:** [Candidate Answer B]
**Question:** Which answer is more accurate and better explained, and why? Provide a detailed comparison focusing on the criteria mentioned above.

Table 14: Prompt template for evaluating complex question answers on the GAIA dataset.

**Prompt for GAIA**

**System Prompt:**
You are an expert in complex problem-solving and knowledge retrieval. Assess the following answers for accuracy, relevance, depth, and comprehensiveness in response to the complex question. Consider whether the answers provide correct information, cover all aspects of the question, and are well-articulated.

**User Prompt:**
Please evaluate the following two answers to the given question. For each answer, analyze the correctness of the information provided, the relevance to the question asked, the depth of the explanation, and the overall quality of the response. Note any inaccuracies, omissions, or areas where the answer could be improved.
**Question:** [Complex Question]
**Answer A:** [Candidate Answer A]
**Answer B:** [Candidate Answer B]
**Question:** Which answer provides a better response to the question and why? Provide a detailed comparison focusing on the criteria mentioned above.

Table 15: Prompt template in the Model Ranking.

**Prompt for Model Ranking**

**System Prompt:**
You are an AI assistant trained to assess and compare responses to user instructions. Your evaluations should be based on accuracy, clarity, completeness, and helpfulness.

**User Prompt:**
Please compare the following two responses to the given instruction. Analyze each response for how well it follows the instruction, the accuracy of the information provided, the clarity of the explanation, and the overall helpfulness to the user. Point out any errors, omissions, or areas where the response could be improved.
**Instruction:** [Instruction Text]
**Response A:** [Candidate Response A]
**Response B:** [Candidate Response B]
**Question:** Which response better addresses the instruction and why? Provide a detailed comparison focusing on the criteria mentioned above.

Table 16: Prompt template for evaluating responses in the Instruct Tuning.

**Prompt for Instruct Tuning**

**System Prompt:**
You are a highly skilled AI assistant trained to evaluate and compare responses to user instructions. Your evaluations should focus on helpfulness, harmlessness, and relevance.

**User Prompt:**
Please compare the following two responses to the given instruction. For each response, assess the following aspects:
**Helpfulness:** Does the response effectively address the instruction and provide useful, accurate information?
**Harmlessness:** Does the response avoid any harmful, offensive, or inappropriate content?
**Relevance:** Is the response directly related to the instruction without unnecessary or irrelevant information?
Provide your analysis for each aspect, noting any issues or areas for improvement.
**Instruction:** [Instruction Text]
**Response A:** [Candidate Response A]
**Response B:** [Candidate Response B]
**Question:** Which response better satisfies the criteria above and why? Provide a detailed explanation supporting your choice, focusing on helpfulness, harmlessness, and relevance.

