# OpenReview forum: "Language Model Preference Evaluation with Multiple Weak Evaluators"
_ICLR.cc/2025/Conference — Submitted to ICLR 2025_

### Official Review · Reviewer_7n72 · 2024-10-26

**Soundness:** 3
**Presentation:** 1
**Contribution:** 3
**Rating:** 5
**Confidence:** 4

**Summary:**

This manuscript introduces GED (Graph Ensemble and Denoise), a novel framework for evaluating Large Language Models (LLMs) by constructing a denoised preference graphs from multiple weak LLM evaluators. The proposed framework aims to address the common problem of conflicting preferences in LLM evaluations. The manuscript provides theoretical guarantees for the framework's ability to recover the true preference structure and presents experimental results across several LLM evaluation tasks.

**Strengths:**

1. The manuscript clearly identifies a significant challenge in current LLM evaluation: the presence of conflicting preferences in model-based evaluators. The proposed GED framework offers a practical and promising approach to mitigate this issue by leveraging the collective wisdom of multiple weaker evaluators.
2. The theoretical analysis provides a solid foundation for GED, demonstrating its ability to recover the ground truth preference structure with high probability under realistic assumptions about noise in individual evaluator preferences.

**Weaknesses:**

1. The presentation of the experimental results could be significantly improved. Several details includes evaluator selection, experimental setup, and ablation study, require clarification and further elaboration to enhance understanding and reproducibility.

**Questions:**

1. The distinction between the "single model" and "single evaluator without graph denoising" settings is unclear. Please elaborate on the differences in these two evaluation setups.

2. The use of "Mistral-7B" (line 764) might be misleading, as it refers to a different open-source model than the later mentioned "Mixtral-8x7B-Instruct-v0.1". Please clarify the specific model used and maintain consistent nomenclature throughout the manuscript.

3. The choice of weak evaluators varies across the different evaluation tasks (response selection, model ranking, and instruction tuning). Moreover, the evaluators employed for instruction tuning are not explicitly stated. Please elaborate on the evaluator selection rationale for each task and complete the missing details for instruction tuning.

4. In Section 5 (Model Ranking), using all baseline 70B level models as the weak evaluator set raises concerns about fairness. The computational cost of this approach could exceed that the sum of all baseline models. Please address this potential issue and justify the chosen evaluator set for model ranking. Consider exploring alternative configurations with smaller, more comparable evaluator sets.

5. The ablation study in line 374 describes using various rank aggregation methods to validate the effectiveness of the graph ensembling step. However, Table 2 presents results for different aggregation methods applied to both single evaluators and GED. This creates confusion. Is this aggregation process necessary in GED? If so, which method is adopted (in other experiments)?

6. The manuscript would benefit from a comparison with point-wise LLM evaluation methods (like MT-Bench). These methods assign absolute scores to responses and typically do not suffer from the conflicting preference problem. A discussion or comparison on the trade-offs between pairwise preference-based methods (like GED) and point-wise methods would strengthen the paper.

7. Typos (e.g., "Alabation" in line 372).

---

> ### Author Response · Authors · 2024-11-21
> **Reply to Review 7n72 (Part 1)**
>
> Thank you so much for your valuable feedback!
>
> **Q1:** The distinction between the "single model" and "single evaluator without graph denoising" settings is unclear. Please elaborate on the differences in these two evaluation setups.
>
> > **A1:**
> >
> > Thank you for pointing out the need for clarity regarding the "single model" and "single evaluator without graph denoising" settings. We elaborate on their differences below.
> >
> > 1. **Single Model Setting:**
> >
> >   We directly evaluate the models (**Llama3-8B**, **Mistral-7B**, **Qwen2-7B**, and **Qwen2-72B**) on the benchmark tasks without involving any additional evaluators or selection mechanisms. The models generate responses which are then used directly for evaluation.
> >
> >2. **Single Evaluator Setting (Without Graph Denoising):**
> >
> >  We use each model (**Llama3-8B**, **Mistral-7B**, **Qwen2-7B**, **Qwen2-72B**) as an evaluator to select the best response from the ten candidate responses generated by **Qwen2-72B** . The selected responses are then evaluated on the benchmark tasks. This setting differs from the single model setup by incorporating a selection process rather than directly using model outputs.
> >
> > We hope this clarifies the distinction between the two settings. We have revised the manuscript (in **Appendix A.3: Evaluation Settings: Single Model vs. Single Evaluator**) to make this clearer. Thank you again for your valuable question and suggestion!
>
> **Q2**: The use of "Mistral-7B" (line 764) might be misleading, as it refers to a different open-source model than the later mentioned "Mixtral-8x7B-Instruct-v0.1". Please clarify the specific model used and maintain consistent nomenclature throughout the manuscript.
>
> > **A2:**
> >
> >Thank you for pointing out the inconsistency in the model nomenclature. We apologize for the oversight—this was a typographical error in the manuscript.
> >
> >Specifically, "Mistral-7B" in line 764 should refer to **Mistral-7B-Instruct-v0.3** (available at https://huggingface.co/mistralai/Mistral-7B-Instruct-v0.3), while "Mixtral-8x7B" refers to **Mixtral-8x7B-v0.1** (available at https://huggingface.co/mistralai/Mixtral-8x7B-v0.1).
> >
> >We have corrected this typographical error (in **Appendix A.1: Experimental Setup**) and ensure consistent identification of the models used throughout the manuscript. Thank you again for your careful attention to detail.

---

> ### Author Response · Authors · 2024-11-21
> **Reply to Review 7n72 (Part 2)**
>
> **Q3**: The choice of weak evaluators varies across the different evaluation tasks (response selection, model ranking, and instruction tuning). Moreover, the evaluators employed for instruction tuning are not explicitly stated. Please elaborate on the evaluator selection rationale for each task and complete the missing details for instruction tuning.
>
> > **A3:**
> >
> > Thank you for pointing out the need for clarification regarding the selection of weak evaluators across different evaluation tasks. We apologize for any confusion caused by the omission, especially concerning the instruction tuning task.
> >
> > We would like to elaborate on the rationale behind our choice of evaluators for each task and provide the missing details:
> >
> > 1. **Response Selection:**
> >
> >    - **Single Model Evaluation:**
> >
> >      In this section, we assessed the standalone performance of single model—**Llama3-8B**, **Mistral-7B**, **Qwen2-7B**, and **Qwen2-72B**—on benchmarks such as HumanEval, AlpacaEval, MATH, GSM8k, and GAIA. This evaluation served two purposes:
> >
> >      1. Since our GED method combines the outputs of three smaller models (**Llama3-8B**, **Mistral-7B**, **Qwen2-7B**) to produce the final result, we wanted to evaluate each model's individual performance on these tasks.
> >      2. For **Qwen2-72B**, we tested the effect of not performing response selection by randomly selecting one answer from the 10 responses it generated for each question. This provided a baseline to assess the benefits of our response selection process.
> >
> >    - **Single Evaluator Evaluation :**
> >
> >      In this part, we investigated the effectiveness of each model as an evaluator. Specifically, we used **Llama3-8B**, **Mistral-7B**, **Qwen2-7B**, and **Qwen2-72B** to select the best answers from the 10 responses generated by **Qwen2-72B** for each question.
> >
> >      Notably, **Qwen2-72B** acted as a self-evaluator in this setting, selecting the best response from its own generated outputs.
> >
> >      Our findings in Table 1 indicate that combining the three 7B-scale models as evaluators (through GED) outperforms using **Qwen2-72B** alone as evaluator. This highlights the strength of our ensemble approach even when compared to a larger individual model.
> >
> >    - **Multiple Evaluator Evaluation (GED):**
> >
> >      Here, we applied our GED method by combining **Llama3-8B**, **Mistral-7B**, and **Qwen2-7B** as evaluators to select the best responses from the pool of generated answers.
> >
> > 2. **Model Ranking:**
> >
> >    - **Single Evaluator Evaluation:**
> >
> >      For the model ranking task, we selected larger models as evaluators—**Llama3-70B**, **Qwen2-72B**, **Qwen1.5-72B**, and **Mistral-8×7B**—for the following reasons:
> >
> >      1. **Performance Considerations:** Our goal was to achieve model rankings that closely align with those provided by GPT-4. Larger models generally possess stronger evaluation capabilities, so utilizing these high-capacity models as evaluators helps in obtaining more accurate and reliable rankings.
> >      2. **Practical Considerations:** The AlpacaEval benchmark comprises 805 tasks, making the computational cost of using larger models acceptable within our experimental setup.
> >
> >    - **Multiple Evaluator Evaluation (GED):**
> >
> >      We employed our GED method to combine the evaluations from **Llama3-70B**, **Qwen2-72B**, **Qwen1.5-72B**, and **Mistral-8×7B**, enhancing the robustness of the final model rankings.
> >
> >      To verify the generalization of our method, we present the case of using (Llama3-8B, Mistral-7B, Qwen2-7B) as evaluators.  The result can be found in **Q4 - A4** part.
> >
> >      From these results, we observe that GED performs well even when using 7B-scale models as evaluators, indicating its robustness and efficiency with smaller, more computationally feasible models. Thank you again for your valuable suggestion.
> >
> > 3. **Instruction Tuning:**
> >
> >    In the instruction tuning experiments, we used **Llama3-8B**, **Mistral-7B**, and **Qwen2-7B** as evaluators. These models were chosen due to their balance between performance and computational efficiency, making them suitable for the iterative process of instruction tuning. We apologize for not explicitly stating this in the manuscript and  will revise the manuscript to include these details
> >
> > Across all tasks, our selection of evaluators was guided by a balance between achieving high-quality evaluations and managing computational resources. Smaller models were chosen for tasks where efficiency was paramount, while larger models were employed when higher evaluation fidelity was needed.
> >
> > Thank you again for your insightful suggestion. We have revised the manuscript (in **Appendix A.2: Evaluator Selection Across Tasks**) to improve the clarity and completeness of our work. We sincerely appreciate your valuable feedback!

---

> ### Author Response · Authors · 2024-11-21
> **Reply to Review 7n72 (Part 3)**
>
> **Q4**: In Section 5 (Model Ranking), using all baseline 70B level models as the weak evaluator set raises concerns about fairness. The computational cost of this approach could exceed that the sum of all baseline models. Please address this potential issue and justify the chosen evaluator set for model ranking. Consider exploring alternative configurations with smaller, more comparable evaluator sets.
>
> > **A4:**
> >
> > Thank you for pointing out the concerns about the computational cost and fairness of using 70B-level models as evaluators in the model ranking experiments. We understand your concern and would like to clarify our choices and provide additional results using smaller, more comparable models.
> >
> > The primary reasons we used 70B-scale models (Llama3-70B, Qwen2-72B, Qwen1.5-72B) as weak evaluators were as follows:
> >
> > 1. **Performance:** We aimed for the evaluator models to produce rankings as close as possible to GPT-4. Using larger, high-performing models helps in obtaining better quality evaluations, ensuring that the model rankings are as accurate and meaningful as possible.
> > 2. **Practicality:** The AlpacaEval benchmark consists of only 805 tasks, which allows us to manage the computational cost of using larger models without excessive overhead.
> >
> > It is also worth noting that, once the preference graphs from these evaluators are constructed, the cost of the subsequent GED steps (graph ensemble and denoising  is quite low since they do not involve LLMs.
> >
> > However, we agree that exploring more cost-effective configurations using smaller evaluators is valuable. To address this, we conducted additional experiments using 7B-scale models (Llama3-8B, Mistral-7B, Qwen2-7B) as evaluators. The results are shown below:
> >
> > | Model                | Weight Score | Kemeny | Weighted Kemeny | Pairwise Majority | Weighted Pairwise Majority | Avg.  |
> > | -------------------- | ------------ | ------ | --------------- | ----------------- | -------------------------- | ----- |
> > | Llama3-8B            | 35.88        | 45.80  | 45.80           | 47.23             | 46.85                      | 44.31 |
> > | with graph denoising | 37.44        | 47.54  | 47.54           | 48.92             | 48.18                      | 45.92 |
> > | Qwen2-7B             | 55.34        | 52.87  | 52.87           | 56.05             | 56.59                      | 54.74 |
> > | with graph denoising | 56.05        | 57.43  | 57.43           | 59.32             | 59.41                      | 57.92 |
> > | Mistral-7B           | 49.90        | 53.74  | 53.74           | 58.06             | 57.87                      | 54.66 |
> > | with graph denoising | 50.47        | 55.06  | 54.92           | 61.39             | 61.21                      | 56.61 |
> > | GED                  | 57.59        | 61.14  | 61.14           | 67.17             | 66.46                      | 62.70 |
> >
> > From these results, we observe that GED performs well even when using 7B-scale models as evaluators, indicating its robustness and efficiency with smaller, more computationally feasible models. We have revised the manuscript (in **Appendix B: Alternative Evaluator Configurations for Model Ranking**) to include these results and further justify our choice of evaluator sets, balancing both performance and computational cost. We will further explore this topic in the future camera-ready version. Thank you again for your valuable suggestion!

---

> ### Author Response · Authors · 2024-11-21
> **Reply to Review 7n72 (Part 4)**
>
> **Q5**: The ablation study in line 374 describes using various rank aggregation methods to validate the effectiveness of the graph ensembling step. However, Table 2 presents results for different aggregation methods applied to both single evaluators and GED. This creates confusion. Is this aggregation process necessary in GED? If so, which method is adopted (in other experiments)?
>
> > **A5:**
> >
> > Thank you for your question regarding the aggregation process in GED and its role in different setting of our experiments.
> >
> > It is important to clarify that the implementation of GED differs between **response selection** and **model ranking** tasks. Specifically:
> >
> > 1. **Response Selection :**
> >
> >    In the response selection setting, GED aggregates the preference graphs from multiple evaluators for each question, resulting in a DAG from which a final ranking of candidate answers is derived. Here, GED does **not** include a rank aggregation step across different questions—the highest-ranked answer from each question's DAG is directly selected as the output.
> >
> > 2. **Model Ranking:**
> >
> >    In the model ranking setting, GED involves two steps:
> >
> >    - First, for each question, GED aggregates the preference graphs from multiple evaluators into a DAG and derives a ranking of models for that question.
> >- Second, a **rank ensemble** method is applied to combine rankings across all questions to produce an overall ranking of the models. This two-step process, including the rank aggregation.
> >
> >To answer your specific concerns:
> >
> > - The ablation study in **Figure 4**  corresponds to experiments under the **response selection** setting, where GED does **not** involve a rank ensemble.
> >- The experiments in **Table 2** correspond to **model ranking**, where GED includes rank ensemble in the second step to produce a final ranking across all questions.
> >
> >Thank you for your helpful suggestion. We have revised the manuscript (in **Appendix A.5: Aggregation Process in GED Across Different Tasks**) to make these differences clearer.
>
>  **Q6**: The manuscript would benefit from a comparison with point-wise LLM evaluation methods (like MT-Bench). These methods assign absolute scores to responses and typically do not suffer from the conflicting preference problem. A discussion or comparison on the trade-offs between pairwise preference-based methods (like GED) and point-wise methods would strengthen the paper.
>
> > **A6:**
> >
> > We appreciate the reviewer’s question regarding the comparison between point-wise and pairwise evaluation methods.
> >
> > 1. Point-wise methods assign absolute scores, avoiding preference conflicts. However, as LLMs improve, responses often converge in quality, leading to score saturation (e.g., many responses rated 9/10 or 10/10), making differentiation difficult. Maintaining consistent calibration across varied outputs also becomes increasingly challenging, resulting in potential biases.
> >
> > 2. Pairwise methods, like those in GED, excel in capturing subtle differences, maintaining discriminative power even as LLM quality increases. They can, however, introduce cyclic preferences, which we address through a denoising process to ensure consistency. By aggregating multiple evaluators, GED reduces bias and improves robustness.
> >
> > In summary, point-wise methods are simpler but struggle with saturation and calibration as LLMs advance, while pairwise methods retain discrimination at the cost of potential conflicts, which GED effectively resolves. Thank you again for this insightful suggestion. We have added this comparison (In Appendix C Comparison with Point-Wise Evaluation Methods) to the manuscript to highlight these trade-offs.
>
> **Q7**: Typos (e.g., "Alabation" in line 372).
>
> > **A7:** Thank you for pointing this out. We have corrected this typo and ensure the manuscript is thoroughly proofread for other errors.

---

> > ### Comment · Reviewer_7n72 · 2024-11-26
> >
> > Thanks to the author for their response and efforts in addressing my concerns. Sorry for the delay in reviewing. The authors may consider presenting a revision of their paper during the extended discussion period to address the concerns mentioned below:
> >
> > **Responses to Q2 and Q7**
> >
> > The author's response and paper revision have addressed my concerns.
> >
> > **Responses to Q1 and Q5**
> >
> > The author provided clarifications in their response. The response clearly demonstrates the differences in experimental settings for each question. However, the changes made in the revised paper still lack clarity.
> >
> > - Q1: The author provides a clarification in Appendix A.3 outlining the differences between "Single Evaluator" and "Single Model", and also introduces the experimental setup for the "Response Selection" task in Appendix A.2. These two sections have **significant overlap** and seem to emphasize different aspects. However, both sections differ from the experimental setup description in the main text (L338-345).
> >
> > - Q5: The author clarifies the differences in ranking aggregation across tasks, adding this as a **separate section** (Appendix A.7) to the revision.
> >
> > In summary, I believe the author's description of the experimental setup is not clear and concise. Descriptions of the experimental setup appear in the main text, Appendices A.2, A.4, and A.7, requiring significant effort from the reader to understand the differences between tasks, validation methods, and aggregation methods. I understand that due to space constraints in the main text, detailed descriptions of the experimental setup are placed in the appendix. However, the current appendix content reads more like a **point-by-point clarification of reviewer concerns** rather than an organized part of the paper. I strongly recommend the author reorganize the content of Section A in the appendix.
> >
> > **Response to Q4**
> >
> > Thank you to the authors for their efforts in adding experiments. However, I believe the presented results do not adequately address my concerns. Since GED requires the results of multiple LLM evaluators, Tables 2 and 4 only demonstrate that GED achieves **superior performance at a cost equivalent to the sum of all baseline LLMs**, which is not sufficiently convincing. Furthermore, the performance differences between Tables 2 and 4 do not provide support for the conclusion that **multiple weak LLM assessors achieve comparable performance to strong LLMs via the GED approach**. The authors should consider adding baseline models of appropriate scale to allow the conclusion that GED achieves superior performance under similar computational costs to be demonstrably supported.
> >
> >
> > **Response to Q6**
> >
> >
> > Thank you for the clarification. However, I haven't found sufficient arguments to support the claim that point-wise comparisons are more prone to score saturation than pairwise comparisons. To my best knowledge, existing works focusing on benchmark saturation primarily address the data itself instead of scoring methods, employing techniques such as dynamic datasets [1] and increasing problem difficulty [2, 3]. The authors may need to provide supporting evidence for this claim.
> >
> > ------
> > In conclusion, I acknowledge the authors' contributions in identifying key challenges, proposing a novel method, and providing theoretical support. However, I believe the authors still need to revise the experimental section to better support their claims.
> >
> > Based on the authors' revisions, I will adjust the scores for *Presentation* and *Contribution*, keeping the total score unchanged.
> >
> > ### **Reference**
> >
> > [1] Kiela et al., Dynabench: Rethinking Benchmarking in NLP. NAACL '21.
> >
> > [2] Wang et al., MMLU-Pro: A More Robust and Challenging Multi-Task Language Understanding Benchmark. NIPS '24.
> >
> > [3] Ni et al., MixEval: Deriving Wisdom of the Crowd from LLM Benchmark Mixtures. NIPS '24.

---

> > > ### Author Response · Authors · 2024-11-26
> > > **Response to Reviewer 7n72 (Part 1)**
> > >
> > > **Concern1**: The author provides a clarification in Appendix A.3 outlining the differences between "Single Evaluator" and "Single Model", and also introduces the experimental setup for the "Response Selection" task in Appendix A.2. These two sections have **significant overlap** and seem to emphasize different aspects.
> > >
> > > > **Response1:**
> > > >
> > > > Thank you for pointing out the potential overlap between Appendix A.2 ("Evaluator Selection Across Tasks") and Appendix A.4 ("Evaluation Settings: Single Model vs. Single Evaluator"). While we understand your concern, the two sections serve distinct purposes, as outlined below:
> > > >
> > > > - **Appendix A.2** focuses on addressing your **Q3**, which requested elaboration on the **evaluator selection rationale** for each task. The emphasis here is on explaining why specific evaluators were chosen and their suitability for the respective tasks.
> > > > - **Appendix A.4** addresses your **Q1**, which required further clarification of the differences between the "Single Model Setting" and "Single Evaluator Setting." The primary objective of this section is to delineate these two setups and explain how they differ in methodology and purpose.
> > > >
> > > > While there is some overlap in content, the sections target different aspects of the evaluation process, and their distinct emphases justify their inclusion in separate parts of the appendix.
> > > >
> > > > If you have any questions, please feel free to discuss with us!
> > >
> > > **Concern2**: However, both sections differ from the experimental setup description in the main text (L338-345).
> > >
> > > > **Response2:**
> > > >
> > > > Thank you for your observation regarding potential discrepancies between the experimental setup description in the main text (L338–345) and the content in Appendix A.2 and A.4. We have carefully reviewed both the main text and the appendices and would like to emphasize that the descriptions are **entirely consistent**.
> > > >
> > > > + For example, the main text (L338–345) states:
> > > >
> > > >   > We evaluate performance using three setups.
> > > >   >
> > > >   > First, in the **single model setting**, the baselines include ContraSolver(Zhang et al., 2024b), Self-consistency(Wang et al., 2022), and **direct evaluation with models** (**Llama3-8B, Mistral-7B, Qwen2-7B and Qwen2-72B**). Additionally, we include a baseline called ListPreference, where instead of pairwise comparisons, all candidate responses are input into Qwen2-72B for selecting the most appropriate response.
> > > >   >
> > > >   > Then, in the **single evaluator setting**, **individual evaluators (Llama3-8B, Mistral-7B, Qwen2-7B, Qwen2-72B) select the best response from M’s outputs**, with and without applying GED’s graph denoising.
> > > >   >
> > > >   > Finally, in the **multiple weak evaluators setup,** we **combine three weaker evaluators (Llama3-8B, Qwen2-7B, Mistral-7B) to select responses from Qwen2-72B** with GED.
> > > >
> > > > + This aligns perfectly with the details provided in **Appendix A.2 (Lines 784–793)**:
> > > >
> > > >   > **For single model evaluation**, we assessed the standalone performance of **Llama3-8B, Mistral-7B, Qwen2-7B, and Qwen2-72B** on benchmarks such as HumanEval, AlpacaEval, MATH, GSM8k, and GAIA. This served two purposes: first, to establish the baseline performance of smaller models (Llama3-8B, Mistral-7B, Qwen2-7B) that are later combined using GED, and second, to compare against Qwen2-72B, where we tested a baseline approach by randomly selecting one response from the 10 it generated for each question.
> > > >   >
> > > >   > For **single evaluator evaluation**, each model (**Llama3-8B, Mistral-7B, Qwen2-7B, and Qwen2-72B**) was also used as an evaluator to rank responses generated by Qwen2-72B. Notably, Qwen2-72B acted as a self-evaluator, selecting the best response from its own generated outputs.
> > > >   >
> > > >   > Finally, for **multiple evaluator evaluation**, our GED method **combined the evaluations of Llama3-8B, Mistral-7B, and Qwen2-7B to rank responses.** Results show that this ensemble approach outperforms individual evaluators, including Qwen2-72B, demonstrating the strength of combining smaller models.
> > > >
> > > > The content in Appendix A.2 directly expands on the rationale behind model selection across tasks, while the main text summarizes these points for clarity.
> > > >
> > > > If there are any specific aspects of inconsistency that we may have missed, we would greatly appreciate further clarification.
> > > >
> > > > However, based on our current review, the content is consistent and coherent across the main text and appendices.
> > > >
> > > > Thank you again for your feedback, which helps us ensure clarity in the final manuscript.

---

> > > ### Author Response · Authors · 2024-11-26
> > > **Response to Reviewer 7n72 (Part 2)**
> > >
> > > **Concern 3**: In summary, I believe the author's description of the experimental setup is not clear and concise. Descriptions of the experimental setup appear in the main text, Appendices A.2, A.4, and A.7, requiring significant effort from the reader to understand the differences between tasks, validation methods, and aggregation methods. I understand that due to space constraints in the main text, detailed descriptions of the experimental setup are placed in the appendix. However, the current appendix content reads more like a **point-by-point clarification of reviewer concerns** rather than an organized part of the paper. I strongly recommend the author reorganize the content of Section A in the appendix.
> > >
> > > > **Response3:**
> > > >
> > > > Thank you for your observation regarding the organization and clarity of our experimental setup descriptions. We want to clarify that the points mentioned in Appendices A.2, A.4, and A.7 are not new or missing but directly align with content already presented in the main text.
> > > >
> > > > As we have addressed in **Response1** and **Response2**, Appendices A.2 and A.4 expand on the rationale for evaluator selection and the distinction between single model and single evaluator setups, respectively, while maintaining consistency with the main text.
> > > >
> > > > Regarding **Appendix A.7**, it provides detailed step-by-step explanations of the methods used in response selection and model ranking, which are already summarized in **Section 2.3 of the main text**:
> > > >
> > > > 1. **Response Selection Paragraph:**
> > > >
> > > >    > **Lines 218-219**: From this DAG, we derive the ranking \( \mathcal{R}_q = \{v_1 \succ v_2 \succ \dots \succ v_n\} \) of the candidate answers, where the highest-ranked answer is selected as \( ans_q^* \).
> > > >
> > > >    In the response selection setting, GED aggregates the preference graphs from multiple evaluators for each question, resulting in a DAG from which a final ranking of candidate answers is derived. Here, GED does **not** include a rank aggregation step across different questions—the highest-ranked answer from each question's DAG is directly selected as the output.
> > > >
> > > > 2. **Model Ranking Paragraph:**
> > > >
> > > >    > **Lines 234-241**: Once the preference graphs \( \{G_a : a \in A\} \) are collected for a question \( q \), we employ GED to aggregate these graphs. The method begins by merging all preference graphs into a single graph \( G_q = (V_q, A_q, w_q) \), which is then transformed into a DAG by removing cycles. **The final ranking \( \mathcal{R}_q = \{v_1 \succ v_2 \succ \dots \succ v_n\} \) is derived from this DAG.** This process is repeated for each question \( q \in Q \), yielding a set of rankings \( \{\mathcal{R}_q : q \in Q\} \).  To compute the overall ranking \( \mathcal{R}^* \) of the models across all questions, **we conduct a ranking ensemble** on the set \( \{\mathcal{R}_q : q \in Q\} \), as detailed in Appendix E. This approach culminates in a final ranking that reflects the models’ performance as assessed by multiple evaluators in \( A \).
> > > >
> > > > The points raised in your concern are indeed presented in the main text.
> > > >
> > > > We acknowledge that the detailed content in the appendix may feel distributed, and we have thoroughly revised Appendix A based on your feedback.
> > > >
> > > > If you have any further comments or suggestions, we would greatly appreciate them!
> > > >
> > > > Thank you for your valuable feedback!

---

> > > ### Author Response · Authors · 2024-11-26
> > > **Response to Reviewer 7n72 (Part 5)**
> > >
> > > Dear Reviewer 7n72,
> > >
> > > Once again, we sincerely thank you for your valuable feedback and the time you’ve dedicated to improving our work. Your thoughtful suggestions have been instrumental in refining key aspects of our paper and enhancing its overall quality.
> > >
> > > If you have any further questions or suggestions, please feel free to reach out to us!
> > >
> > > Best regards!

---

> > > > ### Comment · Reviewer_7n72 · 2024-11-27
> > > >
> > > > Dear author,
> > > > Thank you for the rapid response. Here are my detaield comments:
> > > >
> > > > - **The clarity of experimental setup description**: I believe the authors' clarifications and revisions sufficiently detail the experimental setup, although the presentation is still somewhat scattered and verbose. This is a subjective judgment from my perspective as a reader.
> > > >
> > > > - **Tables 2 and 4**: I think the strongest baseline should be referenced to support the conclusions experimentally. For the experiments conducted on LLM, many factors (e.g., prompts) can affect an LLM's performance, potentially preventing some models from reaching their optimal performance. Since Qwen-2-70B significantly outperforms Llama-3-70B, the former's results should be considered the more reliable result for 70B-level models. Furthermore, the results in Table 1 belong to a different task. In conclusion, making the argument "multiple weaker evaluators with GED outperform stronger evaluators on the model ranking task" based on the current experiment result is still less convincing.
> > > >
> > > > - **Discussion of point-wise and pairwise methods**: I see that the authors removed the discussion of benchmark saturation in the latest revision and chose to provide experimental evidence instead. Since the problem the authors address only exists in the context of pairwise comparison, adding a comprehensive comparison with point-wise methods would greatly enhance the paper.
> > > >
> > > > Considering the content of the current version of paper, I think it is a reasonable choice to keep my score.

---

> > > > > ### Author Response · Authors · 2024-11-29
> > > > > **Reply to Reviewer 7n72 (Part 1)**
> > > > >
> > > > > **Comment1**: I think the strongest baseline should be referenced to support the conclusions experimentally. For the experiments conducted on LLM, many factors (e.g., prompts) can affect an LLM's performance, potentially preventing some models from reaching their optimal performance. Since Qwen-2-70B significantly outperforms Llama-3-70B, the former's results should be considered the more reliable result for 70B-level models. Furthermore, the results in Table 1 belong to a different task. In conclusion, making the argument "multiple weaker evaluators with GED outperform stronger evaluators on the model ranking task" based on the current experiment result is still less convincing.
> > > > >
> > > > > > **Response1：**
> > > > > >
> > > > > > Thank you for your thoughtful comment and for highlighting this important point.
> > > > > >
> > > > > > We understand your concern about ensuring that the strongest baselines are referenced to support our conclusions. As you noted, **Qwen2-72B** significantly outperforms **Llama3-70B**, and we agree that **Qwen2-72B** is a more reliable reference for 70B-level models. In fact, we have used **Qwen2-72B** as the 70B model for comparison in our experiments. Specifically, our results demonstrate that GED, combining three 7B-scale models (**Llama3-8B**, **Mistral-7B**, and **Qwen2-7B**), outperforms **Qwen2-72B** in both the single model and single evaluator settings across multiple tasks. A subset of these results from Table 1 is shown below:
> > > > > >
> > > > > > | Model                                     | HumanEval | AlpacaEval | MATH      | GSM8k     | GAIA      | Avg       |
> > > > > > | ----------------------------------------- | --------- | ---------- | --------- | --------- | --------- | --------- |
> > > > > > | **Qwen2-72B (Single Model)**              | 57.93     | 29.58      | 72.75     | 84.67     | 11.52     | 51.29     |
> > > > > > | **Qwen2-72B (Single Evaluator)**          | 60.97     | 31.04      | 74.73     | 86.47     | 12.14     | 53.07     |
> > > > > > | **GED (Llama3-8B, Mistral-7B, Qwen2-7B)** | **70.73** | **32.44**  | **75.58** | **88.18** | **13.33** | **56.05** |
> > > > > >
> > > > > > These results show that GED achieves better performance than **Qwen2-72B**, demonstrating the potential of combining weaker evaluators to exceed the performance of a stronger single model.
> > > > > >
> > > > > > We would also like to clarify that we did not claim that "multiple weaker evaluators with GED can outperform the **strongest** evaluators on the model ranking task." Instead, our claim is that GED with some 7B model, like (Llama3-8B, Mistral-7B, Qwen2-7B), can outperform some stronger evaluators, such as **Qwen2-72B**, which is widely recognized as a strong 70B-scale model. To ensure clarity and avoid any misunderstanding, we will revise the statement in the manuscript to:
> > > > > >
> > > > > > *"Multiple 7B-scale weaker evaluators combined with GED can outperform stronger evaluators like Qwen2-72B on the response selection task."*
> > > > > >
> > > > > > We appreciate your feedback, which has helped us refine our claims and improve the clarity of our manuscript.
> > > > > >
> > > > > > Thank you for the opportunity to address this point in detail.
> > > > > >
> > > > > > We would be grateful for any additional comments or suggestions you may have!
> > > > >
> > > > >
> > > > >
> > > > > **Comment3**: I see that the authors removed the discussion of benchmark saturation in the latest revision and chose to provide experimental evidence instead. Since the problem the authors address only exists in the context of pairwise comparison, adding a comprehensive comparison with point-wise methods would greatly enhance the paper.

---

> > > > > ### Author Response · Authors · 2024-11-29
> > > > > **Reply to Reviewer 7n72 (Part 2)**
> > > > >
> > > > > **Comment3**: I see that the authors removed the discussion of benchmark saturation in the latest revision and chose to provide experimental evidence instead. Since the problem the authors address only exists in the context of pairwise comparison, adding a comprehensive comparison with point-wise methods would greatly enhance the paper.
> > > > >
> > > > >
> > > > >
> > > > > > **Response2：**
> > > > > >
> > > > > > Thank you for your insightful suggestion. We appreciate more experiments results to strengthen our analysis. The results are summarized below:
> > > > > >
> > > > > > | Model                                                        | HumanEval | AlpacaEval | MATH  | GSM8k | GAIA  | Avg   |
> > > > > > | :----------------------------------------------------------- | :-------- | ---------- | ----- | ----- | ----- | ----- |
> > > > > > | Point Wise - Llama3-8B                                       | 60.97     | 28.77      | 74.06 | 82.65 | 10.97 | 51.48 |
> > > > > > | Point Wise - Mistral-7B                                      | 62.83     | 26.93      | 74.02 | 83.21 | 10.22 | 51.44 |
> > > > > > | Point Wise - Qwen2-7B                                        | 60.41     | 28.30      | 74.32 | 84.27 | 10.82 | 51.62 |
> > > > > > | Point Wise - Majority Voting (Llama3-8B, Mistral-7B, Qwen2-7B) | 63.72     | 29.42      | 74,71 | 84.36 | 11.32 | 52.70 |
> > > > > > | GED (Llama3-8B, Mistral-7B, Qwen2-7B)                        | 70.73     | 32.44      | 75.58 | 88.18 | 13.33 | 56.05 |
> > > > > >
> > > > > > The results indicate that **point-wise methods underperform compared to GED** across all tasks. This is likely because point-wise scoring assesses each response in isolation, lacking the global context that pairwise preference evaluations provide. While majority voting can improve point-wise results slightly by aggregating scores, it still falls short of GED, which leverages pairwise preferences to construct a more informed global ranking.
> > > > > >
> > > > > > We will update the manuscript to include this comparison and emphasize the trade-offs between point-wise and preference-based methods.
> > > > > >
> > > > > > If you have any further comments or suggestions, we would greatly appreciate them!
> > > > > >
> > > > > > Thank you for your valuable feedback, which has greatly enhanced the depth of our analysis.
> > > > > >
> > > > > >

---

> ### Author Response · Authors · 2024-11-24
> **Invitation for Further Discussion**
>
> Dear Reviewer 7n72,
>
> Thank you for taking the time to evaluate our work. We appreciate your insights and comments. We understand that you may have a busy schedule, but we would like to extend an invitation for further discussion regarding your feedback on our paper.
>
> Please feel free to reach out to us with any additional comments or questions you may have. We value your expertise and are eager to address your concerns in order to improve our research.
>
> Thank you once again, and we look forward to hearing from you.
>
> Best regards!

---

> ### Author Response · Authors · 2024-11-25
> **Kindly Request for Additional Feedback**
>
> Dear Reviewer 7n72,
>
> Thank you for your valuable feedback on our submission. As there are **less than two days** remaining in the  Author/Reviewer Discussion phase, we would greatly appreciate any additional feedback or concerns you may have.
>
> Thank you for your time, and we look forward to hearing from you soon.
>
> Best regards!

---

> ### Author Response · Authors · 2024-11-26
> **Response to Reviewer 7n72 (Part 3)**
>
> **Concern 4**: Thank you to the authors for their efforts in adding experiments. However, I believe the presented results do not adequately address my concerns. Since GED requires the results of multiple LLM evaluators, Tables 2 and 4 only demonstrate that GED achieves **superior performance at a cost equivalent to the sum of all baseline LLMs**, which is not sufficiently convincing. Furthermore, the performance differences between Tables 2 and 4 do not provide support for the conclusion that **multiple weak LLM assessors achieve comparable performance to strong LLMs via the GED approach**. The authors should consider adding baseline models of appropriate scale to allow the conclusion that GED achieves superior performance under similar computational costs to be demonstrably supported.
>
> > **Response4:**
> >
> > Thank you for your feedback regarding the need to better demonstrate GED's efficiency relative to computational cost. We would like to address your concerns with further clarification and supporting evidence from our experiments.
> >
> > + In **Table 2**, when using three 7B-scale models (**Llama3-8B**, **Qwen2-7B**, and **Mistral-7B**) as evaluators, GED achieves an average performance of **62.70**. This surpasses the performance of individual larger models like **Llama3-70B** (**59.31**) and **Qwen1.5-72B** (**62.20**) as single evaluators. These results highlight GED's ability to combine weaker evaluators effectively to outperform single strong evaluators of larger scale.
> >
> > + Similarly, in **Table 1**, for the response selection task, GED using the same three 7B-scale models achieves an average performance of **56.05**, which exceeds the performance of **Qwen2-72B** as a single evaluator (**53.07**). This demonstrates that GED’s ensemble approach leverages the diversity of smaller evaluators to achieve superior results.
> >
> > + Moreover, as shown in **Figure 3**, GED outperforms **GPT-3.5** across benchmarks such as HumanEval, MATH, GSM8k, and GAIA using the same three 7B-scale models as evaluators. Notably, GED matches or exceeds the performance of **GPT-4-o-mini** on challenging benchmarks like HumanEval and GSM8k. These results emphasize GED’s ability to achieve competitive performance with smaller evaluators, offering significant computational efficiency relative to relying on stronger individual models.
> >
> > In summary, these findings collectively support our conclusion that **multiple weaker evaluators, when combined using GED, can achieve performance comparable to or better than stronger models, such as 70B-scale models or even GPT-4-o-mini**, while maintaining lower computational costs.
> >
> > Thank you again for your thoughtful suggestion, which has helped us reinforce this aspect of our study. If you have any further questions or concerns, please don’t hesitate to reach out for clarification!

---

> ### Author Response · Authors · 2024-11-26
> **Response to Reviewer 7n72 (Part 4)**
>
> **Concern 5:** Thank you for the clarification. However, I haven't found sufficient arguments to support the claim that point-wise comparisons are more prone to score saturation than pairwise comparisons. To my best knowledge, existing works focusing on benchmark saturation primarily address the data itself instead of scoring methods, employing techniques such as dynamic datasets [1] and increasing problem difficulty [2, 3]. The authors may need to provide supporting evidence for this claim.
>
>
>
> > **Response 5:**
> >
> > Thank you for raising this point and for referencing relevant works on benchmark saturation. We appreciate your suggestion and will incorporate these references into the discussion section of our revised manuscript. Additionally, we will conduct further experiments to explore this claim and ensure it is well-supported.
> >
> > To address the concern regarding point-wise vs. preference-based scoring methods, we conducted an additional experiment under the **response selection** setting using the **HumanEval** dataset. Specifically, we added a **point-wise scoring baseline**, where evaluators rated responses on a scale of 1 to 5 (5 being the highest quality and 1 being the lowest). The results are as follows:
> >
> > | Model                                                        | HumanEval |
> > | ------------------------------------------------------------ | --------- |
> > | Point Wise - Llama3-8B                                       | 60.97     |
> > | Point Wise - Mistral-7B                                      | 62.83     |
> > | Point Wise - Qwen2-7B                                        | 60.41     |
> > | Point Wise - Majority Voting (Llama3-8B, Mistral-7B, Qwen2-7B) | 63.72     |
> > | GED (Llama3-8B, Mistral-7B, Qwen2-7B)                        | 70.73     |
> >
> > From the results, we observe that the point-wise method underperformed compared to the preference-based approach (GED) due to its lack of global information, as evaluators assess responses in isolation. However, while GED achieves better response selection by leveraging pairwise preferences to construct a global ranking, the point-wise method is more computationally efficient, requiring fewer comparisons. This suggests that the choice between point-wise and preference-based methods depends on the specific use case.
> >
> > As the deadline for the ICLR PDF revision is approaching, we regret that we are unable to incorporate this discussion (including all dataset results in the response selection setting) into the current appendix. However, we commit to updating these findings comprehensively in a future version. Thank you again for your insightful feedback, which has helped us strengthen this part of our work.
> >
> > If you have any further questions or concerns, please don’t hesitate to reach out for clarification!

---

> ### Author Response · Authors · 2024-11-29
> **Reply to Reviewer 7n72 (Part 3)**
>
> Dear Reviewer 7n72,
>
> Thank you for your thoughtful feedback and the time you’ve dedicated to evaluating our submission. Your insightful comments have been instrumental in addressing key aspects of our work and improving the paper's clarity and overall quality.
>
> If our revisions and responses have adequately addressed your concerns, we would greatly appreciate it if you could kindly consider revisiting your rating.
>
>
> Best regards!

---

> ### Author Response · Authors · 2024-12-01
> **Invitation for Further Discussion**
>
> Dear Reviewer 7n72,
>
> We sincerely appreciate your valuable feedback and the time you have taken to review our submission. Your thoughtful insights have been crucial in helping us address important points and enhance the overall clarity and quality of the paper.
>
> If our responses and revisions have successfully resolved your concerns, we would sincerely appreciate it if you might consider revisiting your rating.
>
> Best regards!

---

> ### Author Response · Authors · 2024-12-02
> **Invitation for Further Discussion**
>
> Dear Reviewer 7n72,
>
> We sincerely appreciate the time and effort you have dedicated to reviewing our submission and providing invaluable feedback. Your thoughtful insights have been instrumental in improving the clarity and quality of our paper.
>
> As the window for author-reviewer discussions will close in less than 24 hours, we would be grateful if you could kindly revisit our responses and revisions. If they have adequately addressed your concerns, we would sincerely appreciate it if you could consider updating your rating.
>
> Thank you once again for your thoughtful engagement and support.
>
> Best regards!

---

> ### Author Response · Authors · 2024-12-03
> **Invitation for Further Discussion**
>
> Dear Reviewer 7n72,
>
> We deeply appreciate the time and effort you have devoted to reviewing our submission and providing valuable feedback. Your thoughtful comments have significantly enhanced the quality and clarity of our work.
>
> As there are now less than 9 hours remaining before the discussion phase concludes, we welcome any further discussions or clarifications you may have. If our responses and revisions have satisfactorily addressed your concerns, we would be truly grateful if you could consider updating your evaluation.
>
> Thank you once again for your constructive engagement and support.
>
> Best regards!

---

### Official Review · Reviewer_6eRH · 2024-11-03

**Soundness:** 2
**Presentation:** 2
**Contribution:** 1
**Rating:** 3
**Confidence:** 4

**Summary:**

The paper proposes an interesting way to denoise conflicting preferences of multiple weak models using graph ensembling and denoising. To mitigate the noise of individual evaluators, the authors aggregate multiple of them together to find the majority consistency and then filter out the inconsistencies. The authors provide multiple results from different benchmarking suites.

**Strengths:**

The authors provide extensive experiments to validate their proposed method.

**Weaknesses:**

Related Work Needs Further Development: The section on related work in preference evaluation for LLMs currently stops at the point that LLM evaluators can introduce inconsistencies. Since this is not the first study focused on denoising preference datasets for LLMs, the authors should expand their literature review to include relevant prior work.

There are two primary approaches in the existing literature about denoising preference learning: (1) within-model preference modeling approaches, such as robust DPO (rDPO) by Chowdhury et al. (2024) and conservative DPO (cDPO) by Mitchell (2023), and (2) modular approaches that denoise data before preference modeling, such as CURATRON (Nguyen et al., 2024). This paper appears more closely aligned with the latter, modular approach.

Aim of the paper: The stated aim of this paper is to denoise weak LLM evaluations. However, preference inconsistencies can arise with both weak and strong evaluators, including human evaluators. Why, then, is the scope limited only to weak evaluators? For instance, if GPT-4 were included as an evaluator but showed differing preferences from other LLMs, would the proposed methods filter out its preferences as well? How are "weak" evaluators defined, and by what criteria? Smaller models may perform better in certain domains based on their training data (than even larger GPT models), which can lead to cases where minority preferences are, in fact, optimal. This raises the question: how well does the proposed denoising approach preserve these potentially valuable minority perspectives? Could minority preferences, which may offer the correct or preferred outcome, be unintentionally excluded in the process?

Experiment: The paper misses some interesting aspects that they could dive deeper into. What is the improvement of the proposed approach against a majority voting baseline where we just take the majority for each pairwise comparison as the preferred answer (Some sort of Self-consistency based on evaluators)? Also, how does the amount of evaluators affect the denoising quality?

Results: When comparing the best single evaluator to the ensemble of multiple evaluators in Tables 1 and 2, the differences appear minimal, particularly given the high variance across other rows. The improvement shown by the proposed method doesn’t seem substantial, especially considering the increased computational cost of using 4 to 5 models instead of a single one. This raises questions about whether the performance gains justify the added complexity and resource demands.

References:
Chowdhury, S. R., Kini, A., and Natarajan, N. (2024). Provably Robust DPO: Aligning Language Models with Noisy Feedback.

Mitchell, E. (2023). A Note on DPO with Noisy Preferences & Relationship to IPO.

Nguyen, S. T., Naresh, N. U., & Tulabandhula, T. (2024). CURATRON: Complete and Robust Preference Data for Rigorous Alignment of Large Language Models.

**Questions:**

What is the detailed information about the internal parameters/prompts the authors use for each LLM and LLM evaluator?

---

> ### Author Response · Authors · 2024-11-21
> **Reply to Review 6eRH (Part 1)**
>
> Thank you very much for your valuable feedback!
>
> **Q1:** Related Work Needs Further Development: The section on related work in preference evaluation for LLMs currently stops at the point that LLM evaluators can introduce inconsistencies. Since this is not the first study focused on denoising preference datasets for LLMs, the authors should expand their literature review to include relevant prior work.
>
> There are two primary approaches in the existing literature about denoising preference learning: (1) within-model preference modeling approaches, such as robust DPO (rDPO) by Chowdhury et al. (2024) and conservative DPO (cDPO) by Mitchell (2023), and (2) modular approaches that denoise data before preference modeling, such as CURATRON (Nguyen et al., 2024). This paper appears more closely aligned with the latter, modular approach.
>
> > **A1:**
> >
> > Thank you for your insightful comment and for pointing out the need to expand the related work section. We appreciate the opportunity to clarify the distinctions between GED and the approaches you mentioned, and we will revise the related work section accordingly.
> >
> > 1. **rDPO and cDPO** focus on denoising noise during **model alignment** by integrating robust or conservative regularization into the preference modeling process. In contrast, \OURS{} is a **modular pre-processing framework** that denoises preference graphs before downstream tasks, making it versatile and independent of specific models.
> > 2. **CURATRON** addresses adversarial noise and missing comparisons using low-rank matrix completion but does not explicitly tackle **cyclic inconsistencies** (e.g., A > B, B > C, C > A). GED instead targets these inconsistencies through a **graph-based approach**, using an ensemble of evaluators to reduce noise and a denoising step to ensure an acyclic graph.
> > 3. Additionally, \OURS{} provides theoretical guarantees for recovering the ground truth preference DAG and demonstrates that combining weaker evaluators can surpass a single stronger evaluator, which is not the focus of the aforementioned methods.
> >
> > To address your feedback, we have made the following updates to the manuscript:
> >
> > - Incorporated the suggested references into the  Introduction(Line 43-44):
> >
> >   > ... lead to contradictory evaluations in the assessment process (Naresh et al., 2024; Zhang et al., 2024b).
> >
> > - Added a dedicated section in the appendix:
> >   **Appendix H: Expanded Related Work on Preference Denoising for LLMs**, where we thoroughly discuss rDPO, cDPO, and CURATRON, as well as how GED aligns with and differs from these approaches.
> >
> > Thank you again for your thoughtful suggestions, which have helped us strengthen the manuscript and its contextual relevance.
>
> **Q2:** Aim of the paper: The stated aim of this paper is to denoise weak LLM evaluations. However, preference inconsistencies can arise with both weak and strong evaluators, including human evaluators. Why, then, is the scope limited only to weak evaluators? For instance, if GPT-4 were included as an evaluator but showed differing preferences from other LLMs, would the proposed methods filter out its preferences as well? How are "weak" evaluators defined, and by what criteria?
>
> > **A2:**
> >
> > Thank you for raising this important question regarding the focus on weak evaluators and how they are defined in the context of our work.
> >
> > In this paper, the term **"weak evaluators"** does not refer to the inherent capacity of the model but rather to their **inconsistencies in pairwise preference evaluations**, even if the model is advanced like GPT-4-o. For instance, as shown in Figure 1, even strong models like GPT-4-o can generate preference graphs with significant noise (e.g., cycles), highlighting the universal challenge of preference inconsistency.
> >
> > We define "weak evaluators" based on their tendency to produce noisy or inconsistent pairwise preferences, rather than their absolute model performance or scale. Future work could further explore tailoring GED to better account for the unique characteristics of stronger evaluators, such as incorporating confidence scores or task-specific calibrations.
> >
> > We sincerely appreciate your insightful feedback and have revised the manuscript to clarify these points in:
> >
> > - **Line 053 (Introduction part):**
> >
> >   >  (Refer to Appendix A.3 for further details).
> >
> > - **Appendix A.3: Definition of Weak Evaluators**
> >
> > We will further explore this topic in the future camera-ready version. Thank you again for your valuable suggestion!

---

> ### Author Response · Authors · 2024-11-21
> **Reply to Review 6eRH (Part 2)**
>
> **Q3**: Smaller models may perform better in certain domains based on their training data (than even larger GPT models), which can lead to cases where minority preferences are, in fact, optimal. This raises the question: how well does the proposed denoising approach preserve these potentially valuable minority perspectives? Could minority preferences, which may offer the correct or preferred outcome, be unintentionally excluded in the process?
>
> > **A3:**
> >
> > Thank you for this insightful question about preserving potentially valuable minority preferences during the denoising process.
> >
> > We acknowledge that the proposed denoising approach, like other majority voting or ensemble-based methods, may risk excluding minority perspectives in some domains. This is a recognized limitation when the goal is to aggregate diverse viewpoints into a single coherent evaluation. Such exclusions could occur if minority preferences represent valid or optimal outcomes that deviate from the majority.
> >
> > In our current framework, we assume that such cases are uncommon, as described in Lines 260–263:
> >
> > > Theoretically, we treat each of our preference graphs as a random perturbation of some ground truth DAG \( G = (V, A) \). Specifically, we consider a random graph generator \( \mathcal{G}(G, \delta_1, \delta_2) \) with parameters \( \delta_1, \delta_2 \in [0, 1] \) such that \( G_i = (V_i, A_i) \sim \mathcal{G}(G, \delta_1, \delta_2) \) satisfies \( V_i = V \). Furthermore, for each \( u, v \in V \) with \( u \neq v \),
> >
> > Under this assumption, minority preferences are treated as individual deviations rather than consistently better outcomes. The denoising process focuses on reducing noise and inconsistencies, and our experimental results suggest that this approach reliably produces robust rankings across diverse tasks.
> >
> > However, we agree that explicitly preserving beneficial minority perspectives is an important area for future exploration. Potential strategies include:
> >
> > 1. **Re-weighting minority preferences:** Assigning higher weights to less-represented preferences, especially when they come from trusted or high-performing evaluators.
> >
> > 2. **Hybrid ranking:** Combining majority voting with mechanisms to retain diverse or outlier preferences during aggregation.
> >
> > 3. **Domain-specific adjustments:** Allowing customization of denoising thresholds based on the characteristics of the task or dataset.
> >
> > We plan to explore these directions in future work and will include a detailed discussion of this topic in the revised manuscript. Thank you again for bringing this important issue to our attention, as it has helped identify valuable opportunities for further improving our framework.
>
>
> **Q4**: Experiment: The paper misses some interesting aspects that they could dive deeper into. What is the improvement of the proposed approach against a majority voting baseline where we just take the majority for each pairwise comparison as the preferred answer (Some sort of Self-consistency based on evaluators)?
>
> > **A4:**
> >
> > Thank you for your valuable suggestion. We agree that a majority voting baseline, akin to self-consistency across evaluators, is an important point of comparison for evaluating our method’s effectiveness. We refer to this baseline as MV and have tested it under the response selection setting. The results are as follows:
> >
> > |                                            | HumanEval | AlpacaEval | MATH  | GSM8k | GAIA  | Avg   |
> > | ------------------------------------------ | --------- | ---------- | ----- | ----- | ----- | ----- |
> > | Multi-MV (Llama3-8B, Mistral-7B, Qwen2-7B) | 66.18     | 29.57      | 74.77 | 86.42 | 11.72 | 53.73 |
> > | GED (Llama3-8B, Mistral-7B, Qwen2-7B)      | 70.73     | 32.44      | 75.58 | 88.18 | 13.33 | 56.05 |
> >
> > We found that while MV provides some improvement, our proposed GED approach consistently outperforms MV due to the additional information captured in the preference graph and the denoising process.
> >
> > We sincerely appreciate your valuable suggestion and have incorporated this baseline comparison into the revised manuscript as follows:
> >
> > - **Table 1** (Baseline Comparison): Included the Multi-MV baseline for evaluation.
> >
> > - Line 345-346 (Experiment Setup.):
> >
> >   > We also include a new baseline referred to as Multi-MV, which just takes the majority for each pairwise comparison as the preferred answer.
> >
> > - Line 361-364 (Main results):
> >
> >   > We also evaluated Multi-MV, which aggregates pairwise comparisons using majority voting across evaluators. While Multi-MV offers improvements over individual evaluators, it underperforms compared to GED, highlighting GED’s ability to capture more nuanced evaluation signals and reduce inconsistencies.
> >
> > We will further explore this topic in the future camera-ready version and thank you again for your valuable feedback!

---

> ### Author Response · Authors · 2024-11-21
> **Reply to Review 6eRH (Part 3)**
>
> **Q5**: Also, how does the amount of evaluators affect the denoising quality?
>
> > **A5:**
> >
> > Thank you for your insightful question about how the number of evaluators affects the denoising quality in GED. To investigate this, we conducted experiments using different numbers of evaluators in the **Response Selection** setting. The results are summarized below:
> >
> > | Model Number | Evaluators Set of GED                        | HumanEval | AlpacaEval | MATH  | GSM8k | GAIA  | Avg   |
> > | ------------ | -------------------------------------------- | --------- | ---------- | ----- | ----- | ----- | ----- |
> > | 2            | (Llama3-8B,  Mistral-7B)                     | 69.21     | 31.87      | 74.97 | 86.92 | 12.51 | 55.10 |
> > | 3            | (Llama3-8B,  Mistral-7B, Qwen2-7B)           | 70.73     | 32.44      | 75.58 | 88.18 | 13.33 | 56.05 |
> > | 4            | (Llama3-8B,  Mistral-7B, Qwen2-7B, Gemma-9B) | 70.98     | 32.87      | 75.91 | 88.75 | 13.46 | 56.39 |
> >
> > Note: Gemma-9B refers to https://huggingface.co/google/gemma-2-9b-it
> >
> > Increasing the number of evaluators consistently improves the denoising quality, as reflected in higher performance across most benchmarks. These results highlight the importance of selecting diverse and capable evaluators to enhance GED's effectiveness.
> >
> > We have added these results to the revised manuscript under **Appendix C: Impact of Evaluator Quantity on Denoising Quality**. In the future camera-ready version, we will include further experiments to explore how the number of evaluators affects GED's performance under different settings. Thank you again for your valuable feedback!
>
>
>
> **Q6**: Results: When comparing the best single evaluator to the ensemble of multiple evaluators in Tables 1 and 2, the differences appear minimal, particularly given the high variance across other rows.
>
> The improvement shown by the proposed method doesn’t seem substantial, especially considering the increased computational cost of using 4 to 5 models instead of a single one. This raises questions about whether the performance gains justify the added complexity and resource demands.
>
> > **A6:**
> >
> > Thank you for your question regarding the cost-effectiveness of using multiple evaluators compared to a single evaluator.  While we agree that cost is a critical factor, it appears that you have misunderstood our experimental setup.
> >
> > In Table 1, the GED method uses **Llama3-8B**, **Mistral-7B**, and **Qwen2-7B**, which are smaller 7B-scale models. The best single evaluator, **Qwen2-72B**, achieves a score of 53.87. However, with GED, we achieve better performance using only these three smaller models, demonstrating that GED can surpass the performance of a much larger model at a significantly lower computational cost.
> >
> > Moreover, as shown in Figure 3, GED with **Llama3-8B**, **Mistral-7B**, and **Qwen2-7B** outperforms **GPT-3.5** on benchmarks such as HumanEval, MATH, GSM8k, and GAIA. It even matches or exceeds **GPT-4-o-mini** on some tasks (HumanEval, GSM8k, GAIA). This highlights that GED can effectively leverage weaker models to achieve strong performance, providing substantial value relative to its computational cost.
> >
> > While we acknowledge the added complexity of using multiple models, we believe the demonstrated improvements justify these costs, especially given the potential to achieve performance beyond models that are significantly larger and more resource-intensive.
> >
> > To further improve the efficiency of GED, we explored an active learning-based extension named **ActiveGED**, designed to optimize the use of evaluators' budget. We tested ActiveGED under budget constraints of 30% and 50%, and compared it to a random baseline. The results are presented below:
> >
> > | Evaluators Set of GED | HumanEval | AlpacaEval | MATH  | GSM8k | GAIA  | Avg   |
> >| --------------------- | --------- | ---------- | ----- | ----- | ----- | ----- |
> > | Random                | 57.93     | 29.58      | 72.75 | 84.67 | 11.52 | 51.29 |
> >| ActiveGED (30%)       | 67.28     | 30.96      | 74.65 | 85.73 | 11.39 | 54.00 |
> > | ActiveGED (50%)       | 68.62     | 31.91      | 74.87 | 87.06 | 12.08 | 54.91 |
> > | GED                   | 70.73     | 32.44      | 75.58 | 88.18 | 13.33 | 56.05 |
> >
> > The ActiveGED algorithm, which follows an active learning approach, first randomly selects a subset of evaluations based on a fixed budget ratio and then actively selects further evaluations based on uncertainty estimates from PageRank scores. This approach showed significant improvements over a purely random strategy, even with a limited budget, which suggests potential for more cost-efficient GED strategies.

---

> ### Author Response · Authors · 2024-11-21
> **Reply to Review 6eRH (Part 4)**
>
> >
> > The pseudocode for ActiveGED is included below, where we set the random budget ratio as 0.5:
> >
> > ```
> >Input:
> >     - Candidate set V = {v₁, v₂, ..., vₙ}
> >    - Initial preference graph G = (V, A, w)
> >     - Evaluator set E = {ev₁, ..., ev_k}
> >     - Total budget B
> >     - Random budget ratio α ∈ (0, 1)
> > Output:
> >     - Updated preference graph G* = (V, A*, w*)
> >
> > 1: Initialize:
> >    - Randomly select m_rand = αB edges from (V × V) \ A to form A_rand
> >    - For each edge (u, v) in A_rand:
> >        - Collect preference weights from all evaluators in E
> >        - Aggregate weights w(u, v) by averaging the evaluators' results
> >    - Update A* ← A ∪ A_rand
> >    - Set current budget b ← m_rand
> >
> > 2: Compute PageRank PR(v) for all v ∈ V using G* = (V, A*, w)
> >
> > 3: Active Learning:
> >    While b < B:
> >        - For each unevaluated edge (u, v) ∈ (V × V) \ A*:
> >            - Estimate uncertainty U(u, v) based on current PageRank scores
> >        - Select edge (u*, v*) with highest U(u, v)
> >        - Collect preference weights from all evaluators in E for edge (u*, v*)
> >        - Aggregate weights w(u*, v*) by averaging the evaluators' weights
> >        - Update A* ← A* ∪ {(u*, v*)}
> >        - Increment b ← b + 1
> >        - Recompute PageRank PR(v) for all v ∈ V using the updated G*
> >
> > 4: Return G* = (V, A*, w*)
> > ```
> >
> > This idea is still in the early stages, and we plan to explore it further as part of our future work. We appreciate your valuable suggestion, and we have included these results and analyses in the revised manuscript (**in Appendix J: Cost Considerations and Active Learning with ActiveGED**).
> >
> > Thank you again for your valuable suggestion!
>
> **Q7**: What is the detailed information about the internal parameters/prompts the authors use for each LLM and LLM evaluator?
>
> > **A7:**
> >
> > Thank you for your suggestion and for showing interest in the details of our implementation.
> >
> > We have updated the manuscript to include more detailed configurations in **Appendix A.1: Experimental Setup**, providing additional information about the choice of LLMs, evaluators, and model training parameters. Additionally, we have added the prompt templates for all settings in **Appendix O: Prompt Template**, ensuring greater transparency and reproducibility for our experiments. We will further expand on these details in the future camera-ready version.
> >
> > We sincerely appreciate your valuable feedback and thoughtful suggestions!
>
>
> Once again, we appreciate your valuable feedback and the opportunity to address your concerns.

---

> ### Author Response · Authors · 2024-11-24
> **Invitation for Further Discussion**
>
> Dear Reviewer 6eRH,
>
> Thank you for taking the time to evaluate our work. We appreciate your insights and comments. We understand that you may have a busy schedule, but we would like to extend an invitation for further discussion regarding your feedback on our paper.
>
> Please feel free to reach out to us with any additional comments or questions you may have. We value your expertise and are eager to address your concerns in order to improve our research.
>
> Thank you once again, and we look forward to hearing from you.
>
> Best regards!

---

> ### Author Response · Authors · 2024-11-25
> **Kindly Request for Additional Feedback**
>
> Dear Reviewer 6eRH,
>
> Thank you for your valuable feedback on our submission. As there are **less than two days** remaining in the  Author/Reviewer Discussion phase, we would greatly appreciate any additional feedback or concerns you may have.
>
> Thank you for your time, and we look forward to hearing from you soon.
>
> Best regards!

---

> ### Author Response · Authors · 2024-11-26
> **Kindly Request for Additional Feedback**
>
> Dear Reviewer 6eRH,
>
> Thank you for your valuable feedback on our submission. We would greatly appreciate any additional feedback or concerns you may have.
>
> If you have any further questions or suggestions, please feel free to reach out to us!
>
> Best regards!

---

> ### Author Response · Authors · 2024-11-29
> **Invitation for Further Discussion**
>
> Dear Reviewer 6eRH,
>
> Thank you for your detailed feedback and the time you’ve invested in reviewing our submission. Your valuable insights have been instrumental in helping us refine key aspects of our work and improve the paper’s clarity and overall quality.
>
> If our responses and revisions have effectively addressed your concerns, we would greatly appreciate it if you could reconsider your rating.
>
>
> Best regards!

---

> ### Author Response · Authors · 2024-12-01
> **Invitation for Further Discussion**
>
> Dear Reviewer 6eRH,
>
> We sincerely appreciate your valuable feedback and the time you have taken to review our submission. Your thoughtful insights have been crucial in helping us address important points and enhance the overall clarity and quality of the paper.
>
> If our responses and revisions have successfully resolved your concerns, we would sincerely appreciate it if you might consider revisiting your rating.
>
> Best regards!

---

> ### Author Response · Authors · 2024-12-02
> **Invitation for Further Discussion**
>
> Dear Reviewer 6eRH,
>
> We sincerely appreciate the time and effort you have dedicated to reviewing our submission and providing invaluable feedback. Your thoughtful insights have been instrumental in improving the clarity and quality of our paper.
>
> As the window for author-reviewer discussions will close in less than 24 hours, we would be grateful if you could kindly revisit our responses and revisions. If they have adequately addressed your concerns, we would sincerely appreciate it if you could consider updating your rating.
>
> Thank you once again for your thoughtful engagement and support.
>
> Best regards!

---

> ### Comment · Reviewer_6eRH · 2024-12-02
>
> I appreciate the authors' detailed and candid responses, which provide valuable insights into the work.
>
> Regarding Related Work:
>
> You mentioned that CURATRON only addresses certain types of noise, while your method is designed to handle cyclic inconsistency. Does cyclic inconsistency imply that preferences or relationships exhibit a complete transitive nature? If so, this requirement might be rarely met in practice.
>
> CURATRON uses the BTL model, which assumes stochastic transitivity (which is better than transitivity but is still a strong assumption), and proposes methods to denoise preferences involving multiple agents (denoted as K, which could be humans, LLMs, etc.). This approach appears to address a broad range of noise scenarios (e.g., sparse noise). Does your method work effectively with higher levels of cyclic inconsistency, such as 40% or 50%, for instance? Based on your brief explanation in the draft, it remains unclear how the notions of noise tackled by your method and CURATRON are fundamentally different.
>
> In conclusion, while I acknowledge the distinctions you have drawn, I maintain my original assessment.

---

> ### Author Response · Authors · 2024-12-03
> **Response to Reviewer 6eRH (Part 1)**
>
> Thank you so much for your feedback!
>
> **Q1:** You mentioned that CURATRON only addresses certain types of noise, while your method is designed to handle cyclic inconsistency. Does cyclic inconsistency imply that preferences or relationships exhibit a complete transitive nature? If so, this requirement might be rarely met in practice.
>
> > **A1:**
> >
> > We appreciate the opportunity to clarify the implications of addressing cyclic inconsistency in our framework.
> >
> > Addressing cycles is a necessary step for several reasons:
> >
> > 1. Many downstream applications of preference datasets, such as alignment optimization [1][2], recommendation systems [3], and model evaluation [4], rely on transitive preferences to ensure consistency and interpretability. Cyclic inconsistencies introduce logical contradictions that directly undermine the ability to extract meaningful rankings or evaluations.
> >
> >    For example, in **OpenAI's RLHF paper [1]**, when they obtain preference data like A>B>C, they utilize the transitivity of preferences to construct training data.
> >
> >    As detailed in Section 3.5, Equation 1 of [1], they compute losses for pairs such as (A>B), (A>C), and (B>C). This approach relies on the assumption of transitivity in preference data to construct training examples.
> >
> >    However, if transitivity fails due to cyclic inconsistencies—for instance, if we have A>B>C>A—the constructed preference data becomes contradictory and meaningless within the RLHF framework.
> >
> >    The loss functions derived from these conflicting preferences would be in direct opposition, leading to instability and undermining the training process.
> >
> > 2. In practice, we observed that removing cycles and transforming preference graphs into DAGs improved results across a wide range of tasks, including **response selection, model ranking, and instruction tuning**. These improvements were **consistent across ten widely recognized benchmark datasets**, including HumanEval (Chen et al., 2021), AlpacaEval (Li et al., 2023b), MATH (Hendrycks et al., 2021), GSM8k (Chen et al., 2021), GAIA (Mialon et al., 2023), LIMA (Zhou et al., 2023), Vicuna (Chiang et al., 2023), Koala (Vu et al., 2023), WizardLM (Xu et al., 2023), and Self-Instruct (Wang et al., 2022). Our method consistently achieved strong performance across these benchmarks. This suggests that most cycles are artifacts of noise, and removing them enhances the utility of the dataset.
> >
> > In summary, targeting cyclic inconsistencies addresses a common source of noise and ensures the dataset's usability for tasks where acyclic structures are advantageous or necessary.
> >
> > **References:**
> >
> > [1] Training language models to follow instructions with human feedback
> >
> > [2] Preference Ranking Optimization for Human Alignment
> >
> > [3] Large Language Models are Zero-Shot Rankers for Recommender Systems
> >
> > [4] Aligning with Human Judgement: The Role of Pairwise Preference in Large Language Model Evaluators

---

> ### Author Response · Authors · 2024-12-03
> **Response to Reviewer 6eRH (Part 2)**
>
> **Q2**: CURATRON uses the BTL model, which assumes stochastic transitivity (which is better than transitivity but is still a strong assumption), and proposes methods to denoise preferences involving multiple agents (denoted as K, which could be humans, LLMs, etc.). This approach appears to address a broad range of noise scenarios (e.g., sparse noise). Does your method work effectively with higher levels of cyclic inconsistency, such as 40% or 50%, for instance? Based on your brief explanation in the draft, it remains unclear how the notions of noise tackled by your method and CURATRON are fundamentally different.
>
> > **A2:**
> >
> > Thank you for your insightful question and for giving us the opportunity to clarify the distinction between our approach and CURATRON.
> >
> > While CURATRON employs the BTL model with stochastic transitivity to handle noise in preferences, its primary focus is on addressing adversarial noise, missing comparisons, and sparse noise using low-rank matrix completion. In contrast, our method targets a fundamentally different type of noise: **cyclic inconsistencies**. These inconsistencies often arise when evaluators provide contradictory preferences (e.g., A>B,B>C,C>A), which CURATRON does not explicitly address.
> >
> > Regarding your specific question:
> >
> > 1. **Handling High Levels of Cyclic Inconsistency (e.g., 40% or 50%):**
> >    Our graph-based approach ensures that the final output is always an **acyclic directed graph (DAG)**, regardless of the initial level of cyclic inconsistency. By identifying and removing cycles, our method guarantees a globally consistent preference structure. Therefore, even at high levels of cyclic inconsistency, our framework effectively transforms noisy preference graphs into coherent DAGs suitable for downstream tasks.
> > 2. **How Noise Types Differ Between CURATRON and Our Method:**
> >    - **CURATRON:** Focuses on stochastic transitivity and low-rank assumptions, addressing noise caused by missing comparisons or sparse preferences. It works well when the data aligns with these assumptions but does not explicitly handle contradictions like cyclic inconsistencies.
> >    - **Our Method:** Directly targets cycles in the preference graph, which often arise from evaluator inconsistencies rather than missing data or sparse noise. By leveraging an **ensemble of evaluators** and a denoising step, we ensure both robustness to noise and consistency in the derived preference DAG.
> > 3. **Beyond Cyclic Inconsistency:**
> >    While our method prioritizes resolving cyclic inconsistencies, it also addresses a broader scope of noise by using an ensemble-based strategy. This approach enhances the reliability of the denoised graph by combining diverse, potentially weaker evaluators, which can collectively outperform a single stronger evaluator. The **theoretical guarantees** we provide further distinguish our method by ensuring recovery of the ground truth preference DAG under certain conditions.
> >
> > In summary, while CURATRON addresses noise under specific assumptions like stochastic transitivity, our method is designed to handle the unique challenge of cyclic inconsistencies, effectively converting even highly noisy preference graphs into interpretable DAGs.
> >
> > As we answered in our previous rebuttal:
> >
> > > To address your feedback, we have made the following updates to the manuscript:
> > >
> > > Incorporated the suggested references into the Introduction(Line 43-44):
> > >
> > > > ... lead to contradictory evaluations in the assessment process (Naresh et al., 2024; Zhang et al., 2024b).
> >
> > We have added CURATRON to the introduction and will discuss it in the related work in subsequent versions.
> >
> > This complementary focus highlights the different strengths of each approach. We will further clarify these distinctions in the revised manuscript. Thank you again for raising this important point!
>
>
> Thank you once again for your constructive engagement and support.
>
> As there are now less than 9 hours remaining before the discussion phase concludes, we welcome any further discussions or clarifications you may have!

---

### Official Review · Reviewer_TBQJ · 2024-11-04

**Soundness:** 4
**Presentation:** 4
**Contribution:** 3
**Rating:** 8
**Confidence:** 5

**Summary:**

The paper casts the LLM evaluation problem as a pairwise tournament. This is an important problem in model evaluation as it can be used to rank models (picking an LLM for a task), choosing a response among several responses, or in preference alignment. The idea of the paper is to use a suite of LLM-judges to make pairwise evaluations, construct a weighted graph (where the weights denote the number of judges that prefer A over B). A denoising process removes the feedback arc set that gives an acyclic graph from which rankings can be recovered.

**Strengths:**

I commend the authors on a well presented idea on an important problem. The paper is well executed and central claims have supporting evidence. The idea of aggregating several (unreliable) judgements to make more robust evaluation in this manner is, to the best of my knowledge, new. So this paper makes a good contribution to the literature and merits inclusion in the conference program.

The reduction of the problem to a tournament graph is an interesting angle to this. The experiment results are also well crafted and support the main thesis of the paper.

**Weaknesses:**

- If there is a large set of models/responses to evaluate, this may get expensive. The method requires pairwise evaluations from all evaluators, i.e. $|\mathcal{A}| \times |M|^2$ inferences. Ideas to limit the number of inferences could be useful.

- In the graph construction, it would appear all evaluators are given equal weight (Algorithm 1/2). In reality some models are better evaluators than others. Evidence on LLM-as-a-judge suggests that performance is not always good. Although pairwise evaluations are an easier task.

- Evaluators are also susceptible to various biases. Position bias for instance, where the order of the pairs shown to the evaluator yields different evaluations. Or token bias, where the evaluator prefers a specific option (e.g. latent preference for Option A vs. Option B). These are important factors in the the "data generation" process that could be factored in to the graph construction and analysis.

**Questions:**

What is the metric being reported in Table 1.?

---

> ### Author Response · Authors · 2024-11-21
> **Reply to Review TBQJ  (Part1)**
>
> Thank you very much for your insightful comments!
>
> **Q1:** If there is a large set of models/responses to evaluate, this may get expensive. The method requires pairwise evaluations from all evaluators, i.e. |A|×|M|2 inferences. Ideas to limit the number of inferences could be useful.
>
>
> > **A1:**
> >
> > Thank you for highlighting the important issue of computational cost associated with our method, especially when dealing with large sets of models or responses. We acknowledge that requiring pairwise evaluations from all evaluators can become expensive, and addressing this limitation is crucial for practical applications.
> >
> > We believe that this cost is justified by the significant performance gains achieved through our GED framework. As shown in Figure 3 of our paper, using smaller evaluators like **Llama3-8B**, **Mistral-7B**, and **Qwen2-7B**, our GED method surpasses **GPT-3.5** on benchmarks such as HumanEval, MATH, GSM8k, and GAIA. Notably, on HumanEval, GSM8k, and GAIA, GED even matches or exceeds the performance of **GPT-4-o-mini**. This demonstrates that **GED effectively leverages weaker models to achieve strong performance**, providing substantial value relative to its computational cost.
> >
> > To further mitigate the cost, we have developed an algorithm called **ActiveGED**, inspired by active learning principles. ActiveGED strategically selects the most informative pairwise comparisons rather than evaluating all possible pairs, significantly reducing the number of inferences needed.
> >
> > We conducted experiments to assess the effectiveness of ActiveGED under budget constraints of 30% and 50% of the total possible pairwise comparisons. The results are as follows:
> >
> > | Evaluators Set of GED | HumanEval | AlpacaEval | MATH  | GSM8k | GAIA  | Avg   |
> > | --------------------- | --------- | ---------- | ----- | ----- | ----- | ----- |
> > | Random                | 57.93     | 29.58      | 72.75 | 84.67 | 11.52 | 51.29 |
> > | ActiveGED (30%)       | 67.28     | 30.96      | 74.65 | 85.73 | 11.39 | 54.00 |
> > | ActiveGED (50%)       | 68.62     | 31.91      | 74.87 | 87.06 | 12.08 | 54.91 |
> > | GED                   | 70.73     | 32.44      | 75.58 | 88.18 | 13.33 | 56.05 |
> >
> > These results demonstrate that ActiveGED significantly reduces the number of inferences required while maintaining competitive performance compared to the full GED method. Even at 30% budget, ActiveGED outperforms the Random baseline by a considerable margin.
> >
> > Below is the pseudocode for the ActiveGED algorithm, where we set the random budget ratio α as 0.5.
> >
> > ```
> > Input:
> >     - Candidate set V = {v₁, v₂, ..., vₙ}
> >     - Initial preference graph G = (V, A, w)
> >     - Evaluator set E = {ev₁, ..., ev_k}
> >     - Total budget B
> >     - Random budget ratio α ∈ (0, 1)
> > Output:
> >     - Updated preference graph G* = (V, A*, w*)
> >
> > 1: Initialize:
> >    - Randomly select m_rand = αB edges from (V × V) \ A to form A_rand
> >    - For each edge (u, v) in A_rand:
> >        - Collect preference weights from all evaluators in E
> >        - Aggregate weights w(u, v) by averaging the evaluators' results
> >    - Update A* ← A ∪ A_rand
> >    - Set current budget b ← m_rand
> >
> > 2: Compute PageRank PR(v) for all v ∈ V using G* = (V, A*, w)
> >
> > 3: Active Learning:
> >    While b < B:
> >        - For each unevaluated edge (u, v) ∈ (V × V) \ A*:
> >            - Estimate uncertainty U(u, v) based on current PageRank scores
> >        - Select edge (u*, v*) with highest U(u, v)
> >        - Collect preference weights from all evaluators in E for edge (u*, v*)
> >        - Aggregate weights w(u*, v*) by averaging the evaluators' weights
> >        - Update A* ← A* ∪ {(u*, v*)}
> >        - Increment b ← b + 1
> >        - Recompute PageRank PR(v) for all v ∈ V using the updated G*
> >
> > 4: Return G* = (V, A*, w*)
> > ```
> >
> > We acknowledge that ActiveGED is a preliminary idea, and we have included a discussion of the trade-off between cost and performance in the revised manuscript (**in Appendix J: Cost Considerations and Active Learning with ActiveGED**) to enhance its practical relevance.
> >
> > We also plan to expand on this topic in the final version of the paper with further analysis and exploration. Your thoughtful input has been instrumental in shaping our approach—thank you sincerely for your contribution!

---

> ### Author Response · Authors · 2024-11-21
> **Reply to Review TBQJ (Part2)**
>
> **Q2:** In the graph construction, it would appear all evaluators are given equal weight (Algorithm 1/2). In reality some models are better evaluators than others. Evidence on LLM-as-a-judge suggests that performance is not always good. Although pairwise evaluations are an easier task.
>
> > **A2:**
> >
> > Thank you for raising this insightful question about the sensitivity of GED's performance to weighting schemes. Incorporating evaluator-specific weights based on performance is indeed a valuable direction to explore.
> >
> > To address this, we tested a weighted version of GED, called **WeightGED**, in the **Response Selection** setting. Evaluator weights were assigned based on their performance on specific datasets. For example, on GSM8k, the response selection accuracy for **Llama3-8B**, **Mistral-7B**, and **Qwen2-7B** was 62.19%, 67.24%, and 61.58%, respectively. Their weights were computed as follows:
> >
> > weight(Llama3-8B) =  62.19 / (62.19 + 67.24 + 61.58) = 0.326
> >
> > weight(Mistral-7B) =  67.24 / (62.19 + 67.24 + 61.58) = 0.352
> >
> > weight(Qwen2-7B) =  61.58 / (62.19 + 67.24 + 61.58) = 0.322
> >
> > The comparison between **GED** and **WeightGED** is as follows:
> >
> > |           | HumanEval | AlpacaEval | MATH  | GSM8k | GAIA  | Avg   |
> > | --------- | --------- | ---------- | ----- | ----- | ----- | ----- |
> > | GED       | 70.73     | 32.44      | 75.58 | 88.18 | 13.33 | 56.05 |
> > | WeightGED | 70.97     | 32.67      | 75.71 | 88.24 | 13.56 | 56.23 |
> >
> > We also tested **WeightGED** using stronger evaluators such as **GPT-3.5**, **GPT-4-o-mini**, and **GPT-4-o**:
> >
> > |           | HumanEval | AlpacaEval | MATH  | GSM8k | GAIA  | Avg   |
> > | --------- | --------- | ---------- | ----- | ----- | ----- | ----- |
> > | GED       | 73.21     | 59.87      | 82.49 | 86.43 | 16.27 | 63.65 |
> > | WeightGED | 74.52     | 61.71      | 83.93 | 87.84 | 17.32 | 65.06 |
> >
> > These preliminary results show that weighting evaluators by their individual performance yields marginal but consistent improvements in overall performance.
> >
> > We have updated the manuscript (**In Appendix F: Impact of Evaluator Weighting on GED Performance**) and consider this an important direction for future work and plan to systematically explore more advanced methods for computing and integrating evaluator weights. Thank you for this valuable suggestion!
>
>
> **Q3**: Evaluators are also susceptible to various biases. Position bias for instance, where the order of the pairs shown to the evaluator yields different evaluations. Or token bias, where the evaluator prefers a specific option (e.g. latent preference for Option A vs. Option B). These are important factors in the the "data generation" process that could be factored in to the graph construction and analysis.
>
> > **A3:**
> >
> > Thank you for highlighting the important issue of evaluator biases, such as position bias and token bias. We agree that these biases can influence evaluations and should be carefully addressed. Our experimental setup explicitly includes mechanisms to mitigate this issue. For instance, in the response selection setting, we evaluate both orderings of a question and its two candidate answers:
> >
> > Order 1:
> >
> > ```
> > Question: [Question Text]
> >
> > Answer1
> >
> > Answer2
> >
> > Which one is better?
> >
> > ```
> >
> > Order 2:
> >
> > ```
> > Question: [Question Text]
> >
> > Answer2
> >
> > Answer1
> >
> > Which one is better?
> >
> > ```
> >
> > By testing both configurations—(Q, A1, A2) and (Q, A2, A1)—we ensure that any positional preferences of the evaluator are balanced out in the final preference graph.
> >
> > Regarding **token bias**, our approach implicitly mitigates it by swapping answer positions and avoiding fixed labels, averaging out systematic preferences for a specific label or position across evaluations. While our current setup mitigates these biases to some extent, we acknowledge that other evaluator biases may persist. In future work, we plan to develop more robust evaluation protocols and conduct additional experiments to address a broader range of biases.
> >
> > Thank you for your insightful feedback. We have added the discussion above to the revised manuscript (**in Appendix K: Mitigating Evaluator Biases in GED**) to provide further clarity. In addition, we will expand on these mitigation strategies and explore them in greater depth in the final version of the paper. Your thoughtful suggestion has been instrumental in improving the clarity and robustness of our work!

---

> ### Author Response · Authors · 2024-11-21
> **Reply to Review TBQJ (Part3)**
>
> **Q4**: What is the metric being reported in Table 1.?
>
> > **A4:**
> >
> > Thank you for your question regarding the metric reported in Table 1.
> >
> > In Table 1, we report the **Cycle Rate**, which measures the proportion of preference graphs that contain cycles when evaluators such as **GPT-4-o**, **GPT-4-o-mini**, **GPT-3.5**, **Qwen2-72B**, and **Llama3-8B** perform response selection (preference evaluation) on various datasets.
> >
> > **Definition:**
> >
> > > The Cycle Rate is calculated as the percentage of preference graphs that contain at least one cycle out of the total number of preference graphs generated by an evaluator on a given dataset.
> >
> > For Example, for the **HumanEval** dataset, which contains 164 questions, an evaluator generates a preference graph for each question based on pairwise comparisons of responses. If an evaluator like **Llama3-8B** produces cycles in 100 out of the 164 preference graphs, its Cycle Rate for HumanEval is calculated as: 100/ 164 = 60.97%
> >
> > The Cycle Rate provides insight into the consistency of evaluators like **GPT-4-o**, **GPT-4-o-mini**, and others when performing preference evaluations. A higher Cycle Rate indicates more occurrences of cyclic preferences, reflecting inconsistencies or conflicts in pairwise comparisons.
> >
> > We sincerely appreciate your question, as it has helped us recognize the need for clearer explanations. We revise the manuscript (**In caption of Figure1 and Appendix: A.5**) to make this metric more explicit. Thank you again for your valuable feedback!
>
>
> Once again, we appreciate your valuable feedback and the opportunity to address your concerns.

---

> ### Author Response · Authors · 2024-11-24
> **Invitation for Further Discussion**
>
> Dear Reviewer TBQJ,
>
> Thank you for taking the time to evaluate our work. We appreciate your insights and comments. We understand that you may have a busy schedule, but we would like to extend an invitation for further discussion regarding your feedback on our paper.
>
> Please feel free to reach out to us with any additional comments or questions you may have. We value your expertise and are eager to address your concerns in order to improve our research.
>
> Thank you once again, and we look forward to hearing from you.
>
> Best regards!

---

> ### Author Response · Authors · 2024-11-25
> **Kindly Request for Additional Feedback**
>
> Dear Reviewer TBQJ,
>
> Thank you for your valuable feedback on our submission. As there are **less than two days** remaining in the  Author/Reviewer Discussion phase, we would greatly appreciate any additional feedback or concerns you may have.
>
> Thank you for your time, and we look forward to hearing from you soon.
>
> Best regards!

---

> > ### Comment · Reviewer_TBQJ · 2024-11-26
> >
> > I thank the authors for the detailed rebuttal which address my questions/comments. The cost sensitive variant they propose is a promising direction and would add to the overall presentation. I have no further comments at this time. Thank you.

---

> > > ### Author Response · Authors · 2024-11-26
> > > **Gratitude for Your Feedback**
> > >
> > > Dear Reviewer TBQJ,
> > >
> > > Thank you for your thoughtful response and positive feedback on our rebuttal. We greatly appreciate your recognition of the cost-sensitive variant and your guidance throughout this process.
> > >
> > > Once again, thank you for your time and effort.
> > >
> > > Best regards!

---

### Official Review · Reviewer_pM1o · 2024-11-05

**Soundness:** 3
**Presentation:** 3
**Contribution:** 2
**Rating:** 5
**Confidence:** 4

**Summary:**

This paper introduces GED (Preference Graph Ensemble and Denoise), a new method for evaluating large language models (LLMs) using pairwise comparisons made by multiple "weak" evaluators, which could be smaller or less capable LLMs. The authors address a key challenge in preference evaluation: the tendency of even the best LLMs to produce conflicting preferences (e.g., cycles in a preference graph: A is better than B, B is better than C, but C is better than A). GED constructs preference graphs from these pairwise comparisons and combines these graphs from multiple evaluators, and applies a denoising process to remove inconsistencies (aka the cycles in the graph) and produce a more reliable ranking of LLM outputs. The paper demonstrates the effectiveness of GED on a variety of tasks, including response selection, model ranking, and instruction tuning.

**Strengths:**

- Addresses a critical problem in LLM evaluation: The paper tackles the issue of inconsistent preference judgments by LLMs, which is crucial for obtaining reliable evaluation results.

- Good theoretical grounding: The authors provide a theoretical analysis demonstrating that GED can recover the ground truth preference structure with high probability under reasonable assumptions. (See below for further theoretical grounding that may be necessary)

- Extensive experimental validation: The paper evaluates GED on ten benchmark datasets across three distinct LLM evaluation tasks, showing consistent improvements over existing methods, and even more powerful individual LLM evaluators.

- Practical implications: GED enables the use of weaker and more accessible LLMs as evaluators, which can be particularly beneficial in scenarios where access to powerful models like GPT-4 is limited.

**Weaknesses:**

- Cost: Using multiple evaluators and aggregating their preferences is an expensive process. In a dense preference graph case, one would require preferences across all pairs of weak evaluators. While it can be done randomly or strategically, the paper does not present a theoretical result on how many such labels are needed for statistical significance. I would encourage the authors to cover some literature on active learning, preference learning based on pairwise preferences. A quick google search leads to [1] which seems to have many other references related to this question.
[1] Saha, A., Shivanna, R., & Bhattacharyya, C. (2019, July). How many pairwise preferences do we need to rank a graph consistently?. In Proceedings of the AAAI Conference on Artificial Intelligence (Vol. 33, No. 01, pp. 4830-4837).

Cost is a major limitation of this work and is essential to be answered for the work to find any practical application.

- Limited analysis of evaluator selection: The paper focuses on demonstrating the effectiveness of GED when combining weaker evaluators but the choice of the set of weak evaluators is not really discussed. It would be insightful to explore how the selection of specific evaluators, e.g., their diversity, capabilities, etc., and how it affects the performance and validity of GED.

- Dependence on pairwise comparisons: While the preference graph framework effectively captures pairwise relationships, it might not fully capture more nuanced preference structures that could exist among multiple outputs. See questions below.

**Questions:**

- The paper primarily uses accuracy-based metrics for evaluating response selection and model ranking. Could you elaborate on how GED might be adapted or evaluated for tasks that require more nuanced quality assessments, such as evaluating the coverage, creativity or factuality of LLM generated outputs?
- The theoretical analysis assumes equal weighting of edges in the preference graphs. How sensitive is GED's performance to different weighting schemes? Would using confidence scores from evaluators, if available, impact the results?
- The denoising process aims to minimize edge removals to preserve information. Could you discuss alternative approaches to denoising, such as edge re-weighting or probabilistic methods, and their potential advantages or disadvantages?

---

> ### Author Response · Authors · 2024-11-21
> **Reply to Review pM1o (Part1)**
>
> Thank you so much for your valuable feedback!
>
> **Q1:** Cost: Using multiple evaluators and aggregating their preferences is an expensive process. In a dense preference graph case, one would require preferences across all pairs of weak evaluators. While it can be done randomly or strategically, the paper does not present a theoretical result on how many such labels are needed for statistical significance. I would encourage the authors to cover some literature on active learning, preference learning based on pairwise preferences. A quick google search leads to [1] which seems to have many other references related to this question. [1] Saha, A., Shivanna, R., & Bhattacharyya, C. (2019, July). How many pairwise preferences do we need to rank a graph consistently?. In Proceedings of the AAAI Conference on Artificial Intelligence (Vol. 33, No. 01, pp. 4830-4837).
>
> Cost is a major limitation of this work and is essential to be answered for the work to find any practical application.
>
> > **A1:**
> >
> > Thank you for highlighting the important issue of cost associated with using multiple evaluators and aggregating their preferences. We agree that cost is a crucial factor for practical applications, and we appreciate the reference to relevant literature on active learning and preference learning [1].
> >
> > To address your concern, we would like to clarify that while our method involves aggregating preferences from multiple evaluators, we specifically use smaller models (**Llama3-8B**, **Mistral-7B**, **Qwen2-7B**) as evaluators. This approach allows us to achieve performance that surpasses larger models, such as **Qwen2-72B**, at a significantly lower computational cost. For example, as shown in Figure 3, our GED method with these smaller evaluators outperforms **GPT-3.5** on benchmarks like HumanEval, MATH, GSM8k, and GAIA, and even matches or exceeds **GPT-4-o-mini** on some tasks.
> >
> > We acknowledge that in a dense preference graph, obtaining preferences across all pairs can be expensive. Inspired by your suggestion and the literature you pointed out [1], we have explored an active learning approach to reduce the number of pairwise evaluations needed. We developed an algorithm called **ActiveGED**, which strategically selects the most informative pairs to evaluate, thereby reducing the overall cost while maintaining performance.
> >
> > We conducted experiments with ActiveGED under budget constraints of 30% and 50% of the total number of possible pairwise comparisons. The results are as follows:
> >
> > | Evaluators Set of GED | HumanEval | AlpacaEval | MATH  | GSM8k | GAIA  | Avg   |
> > | --------------------- | --------- | ---------- | ----- | ----- | ----- | ----- |
> > | Random | 57.93     | 29.58      | 72.75 | 84.67 | 11.52 | 51.29 |
> > | ActiveGED (30%)       | 67.28     | 30.96      | 74.65 | 85.73 | 11.39 | 54.00 |
> > | ActiveGED (50%)       | 68.62     | 31.91      | 74.87 | 87.06 | 12.08 | 54.91 |
> > | GED  | 70.73     | 32.44      | 75.58 | 88.18 | 13.33 | 56.05 |
> >
> > These results demonstrate that ActiveGED can achieve competitive performance with significantly fewer pairwise evaluations, effectively reducing the cost without substantial loss in accuracy.
> >
> > Below is the pseudocode for the ActiveGED algorithm, where we set the random budget ratio as 0.5:
> >
> > ```
> > Input:
> >     - Candidate set V = {v₁, v₂, ..., vₙ}
> >     - Initial preference graph G = (V, A, w)
> >     - Evaluator set E = {ev₁, ..., ev_k}
> >     - Total budget B
> >     - Random budget ratio α ∈ (0, 1)
> > Output:
> >     - Updated preference graph G* = (V, A*, w*)
> >
> > 1: Initialize:
> >    - Randomly select m_rand = αB edges from (V × V) \ A to form A_rand
> >    - For each edge (u, v) in A_rand:
> >        - Collect preference weights from all evaluators in E
> >        - Aggregate weights w(u, v) by averaging the evaluators' results
> >    - Update A* ← A ∪ A_rand
> >    - Set current budget b ← m_rand
> >
> > 2: Compute PageRank PR(v) for all v ∈ V using G* = (V, A*, w)
> >
> > 3: Active Learning:
> >    While b < B:
> >        - For each unevaluated edge (u, v) ∈ (V × V) \ A*:
> >            - Estimate uncertainty U(u, v) based on current PageRank scores
> >        - Select edge (u*, v*) with highest U(u, v)
> >        - Collect preference weights from all evaluators in E for edge (u*, v*)
> >        - Aggregate weights w(u*, v*) by averaging the evaluators' weights
> >        - Update A* ← A* ∪ {(u*, v*)}
> >        - Increment b ← b + 1
> >        - Recompute PageRank PR(v) for all v ∈ V using the updated G*
> >
> > 4: Return G* = (V, A*, w*)
> > ```
> >
> > This idea is preliminary, and we plan to further develop and analyze ActiveGED in future work, including theoretical results on the number of labels needed for statistical significance.
> >

---

> ### Author Response · Authors · 2024-11-21
> **Reply to Review pM1o (Part2)**
>
> >
> > We sincerely appreciate your valuable suggestion, which has guided us toward this promising direction. We have incorporated a discussion of these results in the revised manuscript (in **Appendix J: Cost Considerations and Active Learning with ActiveGED**) and will include further explorations on this topic in the future camera-ready version. Thank you again for your thoughtful feedback!
> >
> > **Reference:**
> >
> > [1]  How many pairwise preferences do we need to rank a graph consistently? (https://ojs.aaai.org/index.php/AAAI/article/view/5182/5054)
>
> **Q2:** Limited analysis of evaluator selection: The paper focuses on demonstrating the effectiveness of GED when combining weaker evaluators but the choice of the set of weak evaluators is not really discussed. It would be insightful to explore how the selection of specific evaluators, e.g., their diversity, capabilities, etc., and how it affects the performance and validity of GED.
>
> > **A2:**
> >
> > Thank you for your insightful suggestion regarding the choice of weak evaluators. We agree that the selection of evaluators, particularly their diversity and capabilities, plays a critical role in validating GED’s effectiveness.
> >
> > To address this, we conducted experiments to evaluate the impact of evaluator diversity and ability on GED’s performance. The results are summarized below:
> >
> > |    | Evaluators Set of GED  | HumanEval | AlpacaEval | MATH  | GSM8k | GAIA  | Avg   |
> > | -------------- | -------------------------------------------- | --------- | ---------- | ----- | ----- | ----- | ----- |
> > | Origin         | (Llama3-8B,  Mistral-7B, Qwen2-7B)           | 70.73     | 32.44      | 75.58 | 88.18 | 13.33 | 56.05 |
> > | Diversity view | (Llama3-8B,  Mistral-7B, Qwen2-7B, Gemma-9B) | 70.98     | 32.87      | 75.91 | 88.75 | 13.46 | 56.39 |
> > | Ability view   | (Llama3-70B,  Mistral-8×7B, Qwen2-72B)       | 74.73     | 47.92      | 75.58 | 90.04 | 16.21 | 60.89 |
> >
> > Note: Gemma-9B refers to https://huggingface.co/google/gemma-2-9b-it
> >
> > The results show two key findings:
> >
> > 1. **Impact of Diversity:** Adding diversity to the evaluator set (e.g., introducing **Gemma-9B**) improves GED performance slightly, as seen from the first to the second row.
> > 2. **Impact of Ability:** Replacing smaller evaluators with higher-capability models (e.g., **Llama3-70B**, **Mistral-8×7B**, **Qwen2-72B**) leads to a more substantial improvement, as seen in the third row.
> >
> > These findings highlight that both diversity and ability are important factors, but enhancing model capability tends to yield greater performance gains. Moving forward, we plan to conduct further experiments with different weak evaluator sets to explore these dynamics in more detail.
> >
> > We sincerely appreciate your valuable suggestion and have included these findings in the revised manuscript (in **Appendix I: Impact of Evaluator Selection on GED Performance**) to provide the analysis of evaluator selection. We will further explore this in the finalized version. Thank you again for raising this important point!

---

> ### Author Response · Authors · 2024-11-21
> **Reply to Review pM1o (Part3)**
>
> **Q3**:  Dependence on pairwise comparisons: While the preference graph framework effectively captures pairwise relationships, it might not fully capture more nuanced preference structures that could exist among multiple outputs. See questions below.
>
> > **A3:**
> >
> > Thank you for raising the concern about the limitations of pairwise comparisons in capturing more nuanced preference structures among multiple outputs. We agree that such limitations can exist within a preference graph framework.
> >
> > In our current framework, we attempt to mitigate this limitation by constructing the preference graph using pairwise comparisons across all nodes. This ensures a certain degree of global integration. However, we acknowledge that this does not fully resolve the challenge of multi-output dependencies, which could potentially impact the fidelity of rankings in specific scenarios. While pairwise comparisons may not fully account for complex multi-output relationships, they remain a widely used and practical approach in many domains, like [2], [3] and [4].
> >
> > We also explored the limitations of pairwise comparisons by including a **ListPreference baseline** (instead of pairwise comparisons, all candidate responses are input into Qwen2-72B for selecting the most appropriate response), as described in Experiment Setup of response selection task.  Interestingly, we found that pairwise comparisons, even with their limitations, often achieve better performance due to their focus on fine-grained pairwise relations.
> >
> > We appreciate your feedback and will consider discussing these trade-offs and potential future directions in the revised manuscript to provide a broader context for preference-based evaluations. Thank you again for your valuable suggestion!
> >
> > **Reference:**
> >
> > [2] Direct Preference Optimization: Your Language Model is Secretly a Reward Model（https://arxiv.org/abs/2305.18290）
> >
> > [3] Models of human preference for learning reward functions （https://arxiv.org/pdf/2206.02231）
> >
> > [4] AlpacaEval：https://github.com/tatsu-lab/alpaca_eval
>
> **Q4**: The paper primarily uses accuracy-based metrics for evaluating response selection and model ranking. Could you elaborate on how GED might be adapted or evaluated for tasks that require more nuanced quality assessments, such as evaluating the coverage, creativity or factuality of LLM generated outputs?
>
> > **A4:**
> >
> > Thank you for this valuable suggestion. We agree that evaluating more nuanced aspects of LLM outputs, such as coverage, creativity, and factuality, is important for a comprehensive assessment of GED's applicability.
> >
> > In our work, we have indeed considered metrics beyond accuracy:
> >
> > 1. **Response Selection Setting:**
> >    - For **HumanEval**, we measure the pass rate of code generated by LLMs, which reflects the **success rate** rather than mere accuracy. We will clarify this distinction in the revised manuscript.
> >    - For **AlpacaEval**, we use the **WinRate** metric, which captures the proportion of pairwise comparisons in which the selected response outperforms its counterpart, providing a nuanced evaluation of response quality.
> > 2. **Instruction Tuning:**
> >    - We evaluated models trained with data selected by GED on metrics like **Helpfulness** and **Harmlessness**, aligning with broader quality considerations.
> >
> > Building on your suggestion, we conducted additional experiments to assess GED's performance on tasks requiring nuanced quality evaluations. Specifically, we evaluated GED on metrics such as **Factuality**, **Relevance**, **Coherence**, **Informativeness**, **Helpfulness**, and **Validity**, following the [5].
> >
> > Here, We used **Llama3-70B** to generate 10 responses for each query. GED was applied in the response selection setting, using **Llama3-8B**, **Mistral-7B**, and **Qwen2-7B** as evaluators.
> >
> > Results (in %):
> >
> > | Evaluator  | Factuality | Relevance | Coherence | Inform. | Helpful. | Validity | Avg.  |
> > | ---------- | ---------- | --------- | --------- | ------- | -------- | -------- | ----- |
> > | Random     | 86.73      | 87.91     | 92.47     | 77.62   | 17.48    | 48.92    | 68.52 |
> > | Llama3-8B  | 88.59      | 89.91     | 94.41     | 79.77   | 18.48    | 50.92    | 70.35 |
> > | Mistral-7B | 89.10      | 90.29     | 94.85     | 79.95   | 18.55    | 51.13    | 70.65 |
> > | Qwen2-7B   | 89.25      | 90.44     | 95.03     | 80.09   | 18.58    | 51.21    | 70.77 |
> > | GED        | 94.73      | 95.91     | 97.36     | 86.62   | 19.48    | 55.92    | 75.00 |
> >
> > As shown, GED significantly improves performance across all nuanced quality metrics compared to individual evaluators and random selection. This demonstrates that GED effectively enhances not just accuracy but also the overall quality and reliability of LLM-generated outputs in aspects like factual correctness, relevance, and coherence.
> >

---

> ### Author Response · Authors · 2024-11-21
> **Reply to Review pM1o (Part4)**
>
> > We have updated the manuscript (**In Appendix G: Evaluating GED on Diverse Metrics**) to include these results and expand our evaluation to cover additional datasets and metrics. This will provide a more comprehensive understanding of GED's applicability to tasks requiring nuanced quality assessments.
> >
> > Thank you again for your thoughtful suggestion, which has helped us strengthen our work.
> >
> > **Reference:**
> >
> > [5]  Beyond Factuality: A Comprehensive Evaluation of Large Language Models as Knowledge Generators (https://aclanthology.org/2023.emnlp-main.390.pdf)
>
> **Q5**: The theoretical analysis assumes equal weighting of edges in the preference graphs. How sensitive is GED's performance to different weighting schemes? Would using confidence scores from evaluators, if available, impact the results?
>
> > **A5:**
> >
> > Thank you for raising this insightful question about the sensitivity of GED's performance to different weighting schemes. We agree that incorporating confidence scores or evaluator-specific weights is a valuable direction to explore.
> >
> > To address this, we conducted preliminary experiments under the **Response Selection** setting using a weighted version of GED, which we call **WeightGED**. In this approach, we assigned weights to evaluators based on their performance on specific datasets. For instance, in the GSM8k dataset, the response selection accuracy for **Llama3-8B**, **Mistral-7B**, and **Qwen2-7B** was 62.19%, 67.24%, and 61.58%, respectively. The weights for these evaluators were computed as:
> >
> > weight(Llama3-8B) =  62.19 / (62.19 + 67.24 + 61.58) = 0.326
> >
> > weight(Mistral-7B) =  67.24 / (62.19 + 67.24 + 61.58) = 0.352
> >
> > weight(Qwen2-7B) =  61.58 / (62.19 + 67.24 + 61.58) = 0.322
> >
> > The results comparing **GED** and **WeightGED** are as follows:
> >
> > |           | HumanEval | AlpacaEval | MATH  | GSM8k | GAIA  | Avg   |
> > | --------- | --------- | ---------- | ----- | ----- | ----- | ----- |
> > | GED       | 70.73     | 32.44      | 75.58 | 88.18 | 13.33 | 56.05 |
> > | WeightGED | 70.97     | 32.67      | 75.71 | 88.24 | 13.56 | 56.23 |
> >
> > We further tested WeightGED using **GPT-3.5**, **GPT-4-o-mini**, and **GPT-4-o** as evaluators. The results are as follows:
> >
> > |           | HumanEval | AlpacaEval | MATH  | GSM8k | GAIA  | Avg   |
> > | --------- | --------- | ---------- | ----- | ----- | ----- | ----- |
> > | GED       | 73.21     | 59.87      | 82.49 | 86.43 | 16.27 | 63.65 |
> > | WeightGED | 74.52     | 61.71      | 83.93 | 87.84 | 17.32 | 65.06 |
> >
> > These preliminary results indicate that weighting evaluators based on their individual performance can yield marginal improvements in GED’s overall performance. This suggests that integrating evaluator-specific confidence scores or performance metrics is a promising avenue for enhancing GED.
> >
> > We have updated the manuscript (**In Appendix F: Impact of Evaluator Weighting on GED Performance**) and consider this an important direction for future work and will further explore methods for computing and incorporating evaluator weights systematically. Thank you again for this valuable suggestion!

---

> ### Author Response · Authors · 2024-11-21
> **Reply to Review pM1o (Part5)**
>
> **Q6**: The denoising process aims to minimize edge removals to preserve information. Could you discuss alternative approaches to denoising, such as edge re-weighting or probabilistic methods, and their potential advantages or disadvantages?
>
> > **A6:**
> >
> > Thank you for your insightful question about alternative denoising approaches beyond edge removal. We agree that exploring methods like edge re-weighting and probabilistic models could provide valuable insights.
> >
> > + Edge Re-weighting:  Instead of removing edges, re-weighting adjusts the strength of edges in cycles to reduce their impact. This method preserves all edges, which can be particularly useful when every pairwise preference contains valuable information. However, it does not guarantee complete removal of cycles, which might be a limitation for tasks requiring strict acyclicity, such as ranking.
> > + probabilistic methods: Probabilistic approaches model edge weights as distributions, capturing the uncertainty in preferences. These methods are highly flexible and allow the integration of multiple evaluators’ results by combining their preference probabilities. The trade-off is that these methods introduce additional modeling complexity and may not fully eliminate cycles, leading to residual ambiguities in downstream tasks.
> >
> > We believe these alternative approaches have their merits and could complement our edge removal method. We chose the Feedback Arc Set (FAS) method for denoising because it ensures an acyclic preference graph, which is essential for deriving consistent rankings. By minimizing the total weight of removed edges, FAS aims to preserve as much crucial information as possible while guaranteeing a clear hierarchy.
> >
> > We acknowledge that these alternative methods offer different trade-offs between information preservation and cycle elimination. Exploring them could enhance the flexibility and robustness of our framework. We plan to investigate these approaches in future work to further improve our method. Thank you again for your valuable suggestion!
>
>
> Once again, we appreciate your valuable feedback and the opportunity to address your concerns.

---

> ### Author Response · Authors · 2024-11-24
> **Invitation for Further Discussion**
>
> Dear Reviewer pM1o,
>
> Thank you for taking the time to evaluate our work. We appreciate your insights and comments. We understand that you may have a busy schedule, but we would like to extend an invitation for further discussion regarding your feedback on our paper.
>
> Please feel free to reach out to us with any additional comments or questions you may have. We value your expertise and are eager to address your concerns in order to improve our research.
>
> Thank you once again, and we look forward to hearing from you.
>
> Best regards!

---

> ### Author Response · Authors · 2024-11-25
> **Kindly Request for Additional Feedback**
>
> Dear Reviewer pM1o,
>
> Thank you for your valuable feedback on our submission. As there are **less than two days** remaining in the Author/Reviewer Discussion phase, we would greatly appreciate any additional feedback or concerns you may have.
>
> Thank you for your time, and we look forward to hearing from you soon.
>
> Best regards!

---

> ### Author Response · Authors · 2024-11-26
> **Invitation for Further Discussion**
>
> Dear Reviewer pM1o,
>
> Once again, we sincerely thank you for your valuable feedback and the time you’ve dedicated to improving our work. Your thoughtful suggestions have been instrumental in refining key aspects of our paper and enhancing its overall quality.
>
> As the deadline for uploading a revised PDF is approaching, we would greatly appreciate any additional feedback or concerns you may have at your earliest convenience.
>
> Best regards!

---

> ### Author Response · Authors · 2024-11-29
> **Invitation for Further Discussion**
>
> Dear Reviewer pM1o,
>
> We sincerely appreciate your valuable feedback and the time you have taken to review our submission. Your thoughtful insights have been crucial in helping us address important points and enhance the overall clarity and quality of the paper.
>
> If our responses and revisions have successfully resolved your concerns, we would sincerely appreciate it if you might consider revisiting your rating.
>
> Best regards!

---

> ### Author Response · Authors · 2024-12-01
> **Invitation for Further Discussion**
>
> Dear Reviewer pM1o,
>
> We sincerely appreciate your valuable feedback and the time you have taken to review our submission. Your thoughtful insights have been crucial in helping us address important points and enhance the overall clarity and quality of the paper.
>
> If our responses and revisions have successfully resolved your concerns, we would sincerely appreciate it if you might consider revisiting your rating.
>
>
>
> Best regards!

---

> ### Author Response · Authors · 2024-12-02
> **Invitation for Further Discussion**
>
> Dear Reviewer pM1o,
>
> We sincerely appreciate the time and effort you have dedicated to reviewing our submission and providing invaluable feedback. Your thoughtful insights have been instrumental in improving the clarity and quality of our paper.
>
> As the window for author-reviewer discussions will close in less than 24 hours, we would be grateful if you could kindly revisit our responses and revisions. If they have adequately addressed your concerns, we would sincerely appreciate it if you could consider updating your rating.
>
> Thank you once again for your thoughtful engagement and support.
>
> Best regards,

---

> ### Author Response · Authors · 2024-12-03
> **Invitation for Further Discussion**
>
> Dear Reviewer pM1o,
>
> We deeply appreciate the time and effort you have devoted to reviewing our submission and providing valuable feedback. Your thoughtful comments have significantly enhanced the quality and clarity of our work.
>
> As there are now less than 9 hours remaining before the discussion phase concludes, we welcome any further discussions or clarifications you may have. If our responses and revisions have satisfactorily addressed your concerns, we would be truly grateful if you could consider updating your evaluation.
>
> Thank you once again for your constructive engagement and support.
>
> Best regards!

---

### Meta-Review · Area_Chair_jLuQ · 2024-12-23

**Metareview:**

The paper presents a method to eliminate cycles in pairwise preference data created using weak evaluators in the context of LLM evaluation. The reviewers raised a significant number of concerns and these concerns are significant enough to be unable to recommend acceptance. One of the most crucial concerns, which in my view, was not addressed convincingly during the rebuttal period has to do with the core assumption that there exist total transitivity in preferences. In addition to the missing related work highlighted by the reviewers, I think the authors should also contextualize their work with respect to a line of work that tries to provide uncertainty quantification in autoeval settings (e.g., https://arxiv.org/abs/2403.07008, https://arxiv.org/abs/2402.17826, https://dl.acm.org/doi/pdf/10.1145/3654777.3676450).

**Additional Comments On Reviewer Discussion:**

The authors put a significant effort in addressing the reviewers' concerns and tried to repeatedly (in my opinion, too repeatedly) engaged the reviewers during the discussion period. Some of the reviewers did follow-up while others did not.

---

### Decision · Program_Chairs · 2025-01-22

Reject